# CompleteP for RL:
# Maintaining Feature Learning When Scaling Deep Reinforcement Learning

Adam Lee[*][1]  M Ganesh Kumar[*][1]  Blake Bordelon[1]  Cengiz Pehlevan[1][2]

## Abstract

The maximal update parameterization ($\mu P$) provides an approach to scale parameters while preserving feature learning in deep neural networks. In fixed data distribution settings it has been found to yield more stable learning dynamics, more consistent learned features, and enable optimal hyperparameters such as learning rate to transfer from small models to larger ones. However, it is unclear if these benefits readily transfer to reinforcement learning problems, where learning dynamics are coupled to the non-stationary data distribution induced by an agent's own actions. We empirically study how two regimes, the "rich" CompleteP and "lazy" Neural Tangent Kernel (NTK) parameterizations affect hyperparameter transfer, feature and policy consistency, and learning behavior as we scale reinforcement learning agents. Ultimately, we show that agents trained using CompleteP consequentially improves compute and reward efficiency compared to the NTK parameterization across multiple control tasks and variants.

## 1. Introduction

Reinforcement learning (RL) provides a general framework to train agents to solve an environment by learning to maximize rewards. Similar to supervised learning (Hestness et al., 2017) and self-supervised learning (Kaplan et al., 2020; Hoffmann et al., 2022), RL has shown to also benefit from increasing the total parameters in a model (Hilton et al., 2023). However, efficiently scaling up neural networks in a stable manner in this setting remains an open challenge. Unlike supervised and unsupervised learning, RL agents

*Equal contribution [1]School of Engineering and Applied Sciences, Harvard University [2]Kempner Institute, Harvard University. Correspondence to: Adam Lee <adamlee@g.harvard.edu>, M Ganesh Kumar <m_ganeshkumar@u.nus.edu>, Cengiz Pehlevan <cpehlevan@seas.harvard.edu>.

*Proceedings of the 43rd International Conference on Machine Learning*, Seoul, South Korea. PMLR 306, 2026. Copyright 2026 by the author(s).

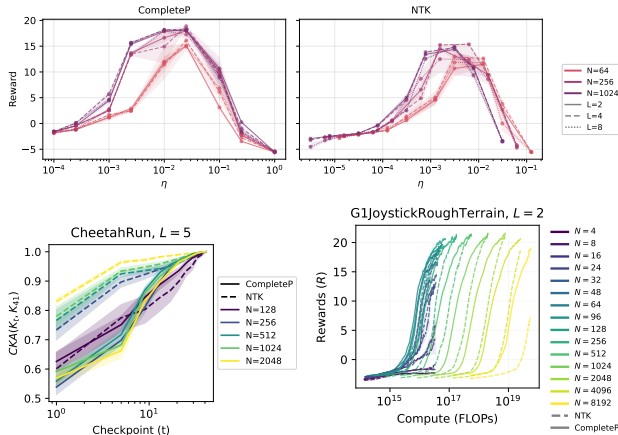

*Figure 1.* **CompleteP parameterization affords learning rate transfer (Left) and consistent rate of feature learning across scales (Middle) to improve learning efficiency (Right).**

face additional difficulties: (1) training data is nonstationary since the distribution of sampled trajectories depends on the evolving policy, (2) large-scale hyperparameter sweeps are prohibitively expensive due to the sample inefficiency of RL and costly environment interactions. Consequently, RL agents are often trained with relatively small networks (Andrychowicz et al., 2020b), limiting systematic scaling studies.

Two learning regimes have been proposed for understanding the scaling behavior of neural networks: the "lazy" regime, arising from parameterizations such as the Neural Tangent Kernel parameterization (Yang et al., 2024a;b), where feature learning is suppressed at large widths, reducing the network to a static kernel method (Jacot et al., 2018; Lee et al., 2019), and the "rich" or feature learning regime, where parameterizations such as mean-field and maximal update ($\mu P$) preserve the evolution of hidden representations even as width grows (Mei et al., 2018; Yang & Hu, 2020; Bordelon & Pehlevan, 2022; Vyas et al., 2023). CompleteP is $\mu P$ with depth-dependent residual scaling (Bordelon et al., 2024b;a; Dey et al., 2026; Yang et al., 2024b) enabling learning rate transfer and improved generalization across width and depth scaling (Yang et al., 2021; Bordelon et al., 2024b; Yang et al., 2024b).

While these advances have transformed supervised and self-supervised learning, their benefits have not been systematically studied in RL, where optimization dynamics and data distributions differ substantially from supervised contexts. Thus, it is not clear whether the benefits of rich feature learning regime also translates to RL for improved efficiency, consistency and generalization capabilities as the network scales. This motivates the core question of our work:

**Does rich feature learning improve a RL agent's policy consistency and learning efficiency as networks scale?**

To address this, we study two scaling parameterizations (NTK and CompleteP) using feedforward residual networks (ResNets) on continuous control RL tasks, where NTK suppresses feature learning at large widths whereas CompleteP maintains feature learning at large widths. Our key contributions are as follows, with links to relevant figures and appendices (containing more figures) for each:

- We demonstrate that the learning rate hyperparameter $\eta_0$ transfers across width and depth when using CompleteP, reducing hyperparameter sweep overhead for PPO (Figures 1, 3, Appendix F) and SAC agents (Appendix F.3).

- We analyze inconsistencies in learning curves (Figure 7a, Appendices O, Q), representations (Figures 4, Appendix L), and policy outputs (Figures 6, Appendix M) across width and random seeds; CompleteP achieves more consistent feature learning for sufficiently wide networks compared to SP and NTK.

- We show that CompleteP significantly improves compute and reward efficiency on 10 continuous control tasks with 6 variants, and 1 discrete task compared to NTK parameterization (Figure 7d, Appendix R).

## 2. Related Works

Proximal Policy Optimization (PPO) (Schulman et al., 2017) and related algorithms have achieved strong RL performance across locomotion, robotics, and vision-based tasks (Zakka et al., 2025; Tassa et al., 2018; Lin et al., 2023; Huang et al., 2022), often surpassing supervised or heuristic methods (Ouyang et al., 2022). However, most RL agents remain limited in size (widths $< 256$, depths $< 4$) due to the expensive nature of environment interactions (Andrychowicz et al., 2020a), though GPU-accelerated simulation is beginning to ease these constraints (Freeman et al., 2021).

**Scaling in Reinforcement Learning.** Recent research shows that increasing model width (Hilton et al., 2023), depth (Wang et al., 2025), or the updates-to-data ratio (Fu et al., 2026; Rybkin et al., 2025) consistently improves RL performance, resulting in higher rewards. Such advances

are particularly vital for robotics, where closing the sim2real gap requires agents with strong generalization and robustness. However, the influence of model parameterization on feature learning remains poorly understood. While wider models can sometimes perform well even in the "lazy" learning regime (Kumar et al., 2022; 2024), theoretical work suggests that mean-field scaling could lead to richer feature learning and improved optimization, though these results have only been established for simplified value learning (Yamamoto et al., 2024) and policy learning settings (Yaghoubi et al., 2026). Hence, empirical validation of rich parameterizations (e.g. $\mu P$) within challenging RL domains remains limited. Our work systematically investigates how scaling model size and parameterization impacts the consistency and generalization of RL agents on the relatively demanding robotics applications.

**Transferring from supervised to reinforcement context.** Scaling laws, parameterization strategies, and feature learning dynamics are well investigated in supervised settings, where techniques such as mean-field or maximal update parameterizations allow hidden representations to evolve and support better generalization (Yang et al., 2021; Bordelon & Pehlevan, 2022). However, transferring these advances to RL is nontrivial (Yamamoto et al., 2024) due to additional challenges unique to RL, such as stochastic action sampling, nonstationary data distributions, catastrophic forgetting, loss of network plasticity and non-convex Temporal Difference error based loss dynamics (Sokar et al., 2023; Dohare et al., 2024; Rybkin et al., 2025; Bordelon et al., 2023). These issues complicate stable scaling and often lead to inconsistent learning trajectories and final performance. Whether continual feature learning, made possible with parameterizations like $\mu P$ (Graldi et al., 2025), can mitigate these RL-specific challenges and lead to improved policy consistency remains an open research question.

**Techniques to mitigate inconsistencies.** A number of modern strategies have emerged to address inconsistency and instability in RL beyond simply increasing scale. Dormant neuron reinitialization (Lee et al., 2024; Dohare et al., 2024) help restore plasticity by reactivating underutilized network units, combating the staleness that can arise during prolonged training. Layer and orthogonal normalization techniques (Lee et al., 2025) have proven effective for stabilizing optimization and reducing variance across different seed runs. Moreover, explicit regularization e.g. weight decay, remains important for controlling capacity and improving generalization. The "BRO" (Bigger, Regularized, Optimistic) approach (Nauman et al., 2024) combines larger models, weight regularization, and mechanisms to balance exploration and exploitation to further reduce performance variability and support consistent learning at scale.

## 3. Methodology

Our goal is to compare how NTK and CompleteP parameterizations affect learning rate consistency, feature evolution, and efficiency in RL agents as we scale networks. In this section, we describe the agent architecture, parameterization schemes, training tasks and diagnostic analysis setup.

### 3.1. Residual Networks for Reinforcement Learning

We leverage on two separate networks $h^\pi$ and $h^v$ to learn the policy and value function respectively, although a single deep network with shared features that splits into the actor and critic layers is possible using the same parameterization (Fig. 2, gray arrow). The actor head is parameterized by $W^L$ and $\boldsymbol{b}^L_\pi$ to output the action vector $\boldsymbol{a}_t$, while the critic head uses parameters $\boldsymbol{w}^L$ and $b^L_v$ to output the value estimate $v_t$. Notably, $\Omega$ exists as a knob to smoothly control the level of "feature learning" in the model. We note that this parameter is often referred to as $\gamma$ in other works (Graldi et al., 2025; Atanasov et al., 2025), but we chose $\Omega$ as to avoid confusion with the discount factor in the reward prediction error.

Figure 2 shows the feedforward residual network architecture we used to study the influence of network parameterization during reinforcement learning. The agent receives an observation vector $\boldsymbol{o}_t$ from the environment, which is first processed by an input layer parameterized by weights $W^0$ with a scaling factor $1/\sqrt{D}$ and bias $\boldsymbol{b}^0$. This produces a preactivation vector, $\boldsymbol{h}^1_t$ which is fed through a stack of $L$ residual blocks (ResNet $\times L$). $L = 0$ means there is no residual layer but a single feature layer and an actor/critic layer. Each residual layer applies a Rectified Linear Unit (ReLU) nonlinearity $\phi(\cdot)$, followed by weight matrix $W^\ell$ and bias $\boldsymbol{b}^\ell$, and two scaling factors ($N$ and $L$). A skip connection adds the input preactivation $\boldsymbol{h}^\ell_t$ to the output of the layer to form $\boldsymbol{h}^{\ell+1}_t$. After passing through the final residual layer, the penultimate preactivation is processed by a nonlinearity and then to the actor layer which has a normalizing scale factor governed by $\Omega N$.

$$\boldsymbol{h}^1_t = \frac{1}{D} \sum_j^D W^0_{ij} \boldsymbol{o}_t + \boldsymbol{b}^0 \,,$$

$$\boldsymbol{h}^{\ell+1}_t = \boldsymbol{h}^\ell_t + \frac{1}{L^\alpha}\left[\frac{1}{N}\sum_j^N W^l_{ij}\phi(\boldsymbol{h}^\ell_t) + \boldsymbol{b}^\ell\right], \quad (1)$$

$$v_t = \frac{1}{\Omega N}\boldsymbol{w}^L_V \cdot \phi(\boldsymbol{h}^{L,v}_t) + b^L_v \,,$$

$$\boldsymbol{a}_t = \frac{1}{\Omega N}\boldsymbol{W}^L_a\phi(\boldsymbol{h}^{L,\pi}_t) + \boldsymbol{b}^L_\pi \,.$$

### 3.2. NTK and CompleteP Parameterizations

A key choice in scaling deep networks is the parameterization scheme, which are rules for initializing parameters

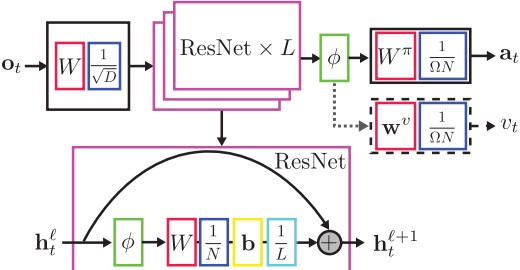

*Figure 2.* **Residual agent architecture.** Policy and value function can be learned using separate or shared (grey) ResNets.

| Network Initialization | Input, Output | Hidden |
|---|---|---|
| Weight variance ($\sigma^2$) | 1 | $N$ |
| Bias variance ($\sigma^2_b$) | 1 | 1 |

| Hyperparameter scaling | NTK | CompleteP |
|---|---|---|
| Output Scaling ($\Omega$) | $\Omega_0/\sqrt{N}$ | $\Omega_0$ |
| Depth Scaling ($\alpha$) | 0 | 1 |
| Learning Rate ($\eta$) | $\eta_0\Omega_0/\sqrt{N}$ | $\eta_0\Omega_0$ |

*Table 1.* Network initialization and hyperparameter scaling rules when using the Adam optimizer. Refer to Appendix A for SGD optimizer. $\Omega_0$ and $\eta_0$ are free hyperparameters. $\Omega$, the output scaling, depends on the base $\Omega_0$ and the width in the case of NTK scalings.

and scaling hyperparameters across width ($N$) and depth ($L$). These schemes play a big role in the dynamics of learning, the preservation of meaningful features, and the transferability of hyperparameters. Our goal is to ensure desirable properties ("desiderata") as networks are scaled, especially in reinforcement learning. Refer to Appendix A for the desiderata. The NTK parameterization is designed such that as $N \to \infty$, the learned features change minimally: "lazy" learning with a rate of $\Theta(1/\sqrt{N})$. Conversely, maximal update ($\mu P$) modifies the scaling rules so feature learning capacity is retained even as width and depth grows: the change in intermediate representations remains $\Theta(1) \sim \boldsymbol{h}(t) - \boldsymbol{h}(0)$ (Yang & Hu, 2020). These two parameterizations cleanly isolate the extremes of feature learning while the Standard Parameterization (SP) does not enforce either limit and therefore mixes width-dependent effects from both regimes, making it harder to attribute observed behavior to a specific scaling principle. Additionally, introducing a $1/L$ scaling in the residual branch affords consistent feature evolution with depth scaling. Normalizing for both width and depth scaling is the CompleteP parameterization (Dey et al., 2026) allows all three desiderata to be simultaneously satisfied.

We set $\Omega_0 = 1$ for most experiments, except in Appendix G to determine if there was an optimal value as seen in (Graldi et al., 2025; Atanasov et al., 2025). For each environment an optimal learning rate $\eta^*_0$ was determined for a specific

width and depth configuration. Performing hyperparmeter sweeps for each configuration is computationally expensive. For NTK agents, we scaled the learning rate by a factor of $1/\sqrt{N}$ according to the learning rate scaling rule in Table 1 and used by Hilton et al. (2023). This heuristic ensures learning rates for NTK agents are close to optimal without needing additional hyperparmeter sweeps when scaling. For CompleteP agents, learning rate was a constant. Note that the scaling rules in Table 1 are specifically for the ADAM optimizer which was used in all of our experiments. Though the scaling rule derivations are simpler for SGD optimizer, training agents using SGD resulted in lower maximum reward and slower learning performance. Hence, we focused on developing the scaling rules for ADAM optimizer instead as that is the current SOTA optimizer for RL research. Refer to Appendix A for SGD optimizer rules. We set the ADAM stabilization constant to a sufficiently small value $\epsilon = 10^{-8}$ which did not dominate the update dynamics at the model widths we study (Dey et al., 2026; Everett et al., 2024) although scaling $\epsilon$ with width is theoretically more principled.

### 3.3. Training on Continuous Control Tasks

We evaluated the effects of network parameterization on learning and representation by training agents on ten continuous control environments, ranging from standard Deep-Mind Control Suite tasks (Tassa et al., 2018) to more difficult manipulation in MuJoCo Playground (Zakka et al., 2025) and humanoid maze navigation problems (Wang et al., 2025; Bortkiewicz et al., 2025). In each setting, agents observe high-dimensional sensory inputs and learn to modulate joint torques to maximize cumulative reward. Most tasks use dense rewards to encourage learning at every step (reward shaping), but we also considered task variants with sparse rewards (Appendix O.4) given only upon successful completion, providing a greater test of exploration and learning capabilities.

Most experiments used Proximal Policy Optimization (PPO) with the ADAM optimizer (Freeman et al., 2021). This is an online learning method that iteratively gathers trajectories with the current policy, updates the network, and then collects new data under the updated policy. We chose to primarily study an on-policy learning algorithm because it reflects the most extreme case of data distribution shift in reinforcement learning algorithms: Changes in the networks from learning are immediately realized in the new collection of trajectories as data for the next update. This is in contrast to off-policy algorithms such as Soft Actor-Critic (SAC), which slow data distributional shift through experience replay. Nevertheless, we performed additional scaling experiments using SAC to determine the generality of the effects of network parameterizations. Additional details on the effects of policy updates and data distribution can be

found in the Appendix I. Models with varying hyperparameters, widths and depths were trained on a mix of A100s and H100s.

### 3.4. Kernel and Action Distribution Analysis

For feature and policy analyses, we constructed a synthetic evaluation dataset by varying each state dimension individually across a standardized range while holding all other dimensions at zero. Since our PPO implementation (Freeman et al., 2021) employs observation normalization, a majority of observation values are processed between the range of -3 and 3, with more extreme values taking larger magnitudes. As such, for each of the $D$ observation dimensions, we sampled $S$ linearly spaced values from $-3$ to 3, resulting in $DS$ total samples—each with only a single dimension of nonzero values. (See Appendix B for an example.) This range of values ensures that we cover input values across different levels of extremity. Ultimately the purpose of this synthetic dataset is to enable consistent and interpretable comparison of feature kernels and policy outputs across training, network configurations, and random seeds, similar to kernel analyses in supervised learning (Vyas et al., 2023).

## 4. Learning Rate Transfers Across Scales

Before scaling agent parameters, we want to reduce the need to sweep for the optimal learning rate hyperparameter, especially when training agents on highly challenging tasks like Humanoid Maze. An ideal setting is when we only need to perform a single learning rate sweep with a small width (e.g. $N = 32$) and depth (e.g. $L = 2$) model and transfer it to train a larger parameter model. We will call this the optimal learning rate $\eta_0^*$. In this section, we demonstrate that agents initialized in the CompleteP parameterization demonstrate improved learning rate hyperparameter transfer across width and depth compared to the NTK parameterization. Refer to Fig. 8d and Appendix F.2 for learning rate transfer curves using Standard Parameterization.

Figure 3 demonstrates how the final reward achieved by the agent (after the last gradient update) changes based on the learning rate ($\eta_0$) hyperparameter across different depths and widths for the NTK and CompleteP parametrized agents. For both HalfCheetah and HumanoidMaze, we see that increasing either the model width and depth leads to a general increase in overall performance, supporting the trend that scaling agent parameters improves learning performance. However, naively increasing the depth in NTK agents without learning rate optimization causes a general decrease in the final rewards achieved. NTK agents with deeper models require a lower learning rate to achieve similar performance. Instead, CompleteP agents demonstrate improved consistency where a single learning rate ($\eta_0^*$) transfers to deeper and wider models for reliable increase in learning

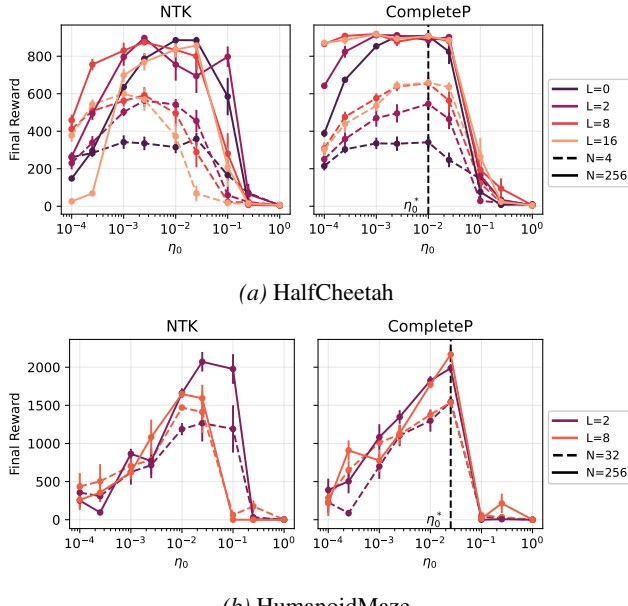

*(a)* HalfCheetah

*(b)* HumanoidMaze

*Figure 3.* **Learning rate hyperparameter transfers across width and depth more consistently with CompleteP parameterization.** Across environments, NTK agents require a general reduction in the learning rate with scaling, though this is heuristically done using the $\Omega$ hyperparameter. Comparatively, the same learning rate $\eta_0^*$ is optimal across different scales of CompleteP agents, and showing improvements in final rewards. Refer to Appendix F for other environments. Error bars are standard error across 10 and 5 seeds for HalfCheetah and HumanoidMaze respectively.

performance. Refer to Appendix F for learning rate transfer experiments for CartpoleSwingup, AcrobotSwingup, PandaPickCube, G1Rough, and HalfCheetah with Standard Parameterization. We also include results demonstrating similar learning rate transfer properties with SAC, a popular off-policy algorithm, in Appendix F.3.

Hence, the transferability of learning rates across different model widths and depths applies even to the reinforcement learning context. While NTK agents can achieve similar final reward as CompleteP agents, only the latter reliably achieves our goal to reduce compute costs incurred for hyperparameter optimization. Additionally, while we do see learning rate transferring in NTK agents for some environments, CompleteP agents demonstrate a more robust transfer. We can now use this transferability to perform a single learning rate sweep for smaller width and depth models to determine the optimal learning rate to train reinforcement learning agents to solve significantly harder environments.

We additionally performed batch-size scaling experiments on CheetahRun varying the minibatch size $B = 128-32768$ and number of minibatches accordingly to make sure the same amount of data was ingested. We observe a critical batch size beyond which performance degrades. CompleteP

continues to preserve learning-rate transfer across scale even under large-batch training while NTK's optimal learning rate diverges sharply (Appendix W).

## 5. Inconsistencies in Representation Learning

We have seen that the CompleteP parameterization robustly improves learning performance when we scale width and depth. However, reinforcement learning is notorious for having high variance among seeds. Hence, improving the consistency of feature learning could reduce sources of variance during the noisy training process. We will now analyze how features and logits evolve during training with scale.

We obtained a feature matrix $H \in R^{DS \times N}$ by feeding in the synthetic dataset with $DS$ samples (refer to section 3.4) to agents at different checkpoints of training ($t$). A kernel (defined as the Gram matrix $HH^\top$) of the penultimate layer activations ($h^L$) is then constructed at different checkpoints. Each kernel's alignment with the the kernel at the last checkpoint ($t$) is visualized in Figure 4a. NTK agents demonstrate a gradual increase in alignment with the final kernel. However, this alignment is width-dependent. Larger width NTK agents demonstrate a smaller rate of change across training compared to smaller width agents. This is expected as NTK agents demonstrate slower feature evolution with width scaling (Refer to Appendix K) in the order of $\Theta(1/\sqrt{N})$. Hence, the difference between the final and intermediate kernels will be persistently high due to a lack of feature learning. Whereas smaller width NTK agents demonstrate a slightly faster feature evolution. Refer to Appendix P for feature consistency in other environments.

Conversely, CompleteP agents demonstrate width-invariant feature evolution where agents with different widths demonstrate similar rates of change in feature evolution (also see Appendix K) due to our feature learning parameterization. The example kernels demonstrate the difference in feature evolution with training by the NTK and CompleteP agents (Fig. 4b). Additionally, features in CompleteP agents initialized with different seeds demonstrate increasing consistency with width scaling compared to NTK agents (Fig. 4c & 26), suggesting that CompleteP might be able to reduce the variance in feature learning, even in the reinforcement learning framework. It is expected for random feature kernels (gray) to align at large widths.

Additionally, Figure 5 shows NTK agents of different widths and depths demonstrate highly variable eigenspectrum in the early onset of training, but gradually learn low-dimensional representations. Conversely, since the early onset of training, CompleteP agents with different number of parameters demonstrate consistent eigenspectrum, and maintain this low-dimensional representation throughout training. Note that checkpoint 0 indicates the first model training check-

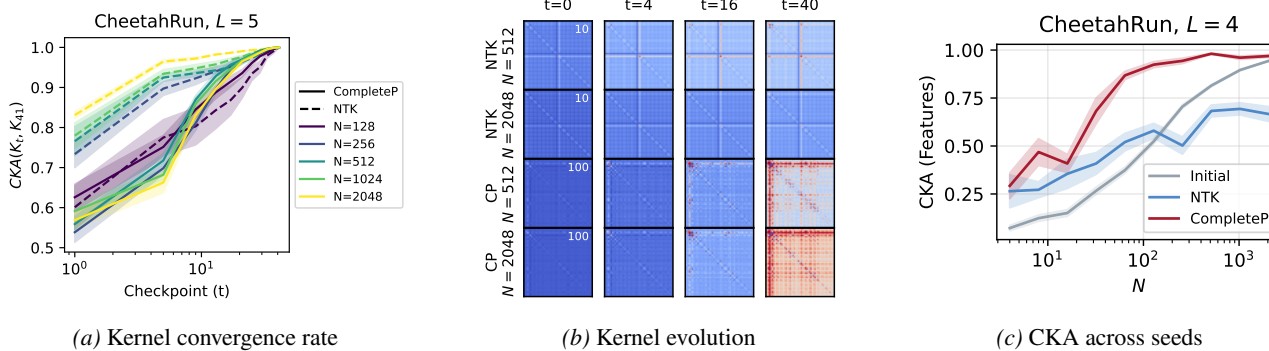

*(a)* Kernel convergence rate      *(b)* Kernel evolution      *(c)* CKA across seeds

*Figure 4.* **CompleteP maintains consistent speed of feature learning across width scaling.** **(a)** As width increases, NTK agents show gradual and width-dependent feature evolution; larger widths evolve slower and hence more aligned with the feature kernel at the end of training while smaller width models show a faster rate of change in alignment. Conversely, CompleteP agents demonstrate width-consistent feature evolution; features of larger width agents at each checkpoint aligns with the feature kernel at the final checkpoint at the same or faster speed than smaller width agents. **(b)** Example NTK and CompleteP feature kernels evolving at width-dependent and width-invariant rates. **(c)** Feature kernels across different seeds are more aligned for CompleteP agents compared to NTK agents with scale. Shaded area is standard error over 10 seeds. Refer to Appendix L for other tasks.

point after random network initialization.

One of the key differences between the supervised and reinforcement learning framework is that the data distribution that is used to update a network's parameters is always changing in the latter (Fig. 16). The data distribution stops changing once policy learning has concluded. Non-stationary data distributions accelerate the loss of plasticity phenomenon where networks lose the ability to learn the changing distribution (Dohare et al., 2024). Since NTK agents have reduced feature learning capabilities with scale, they can maintain a stable proportion of dormant neurons though they are not plastic. Correspondingly, CompleteP agents maintain $\Theta(1)$ feature evolution with scale (Fig. 19), and demonstrate a similar proportion of dormant neurons to NTK agents (Fig. 17), especially at larger widths. Whether CompleteP agents can continually adapt to changing data distributions needs further analysis. Still, the ability to maintain plasticity even with maximal feature updates might be a possible mechanism for continual reinforcement learning.

## 6. Inconsistencies in Policy Learning

To determine how logits evolve, we plot the difference in logits generated (mean squared error) between each model checkpoint ($t$) and the final logit distribution using the synthetic dataset. Both NTK and CompleteP agents demonstrate width invariant logit evolution (Figure 6a). As learning proceeded, we observe a similar rate of logit evolution between the largest width and smaller width agents. This systematic divergence across scale is observed in both NTK and CompleteP agents (Fig. 6b), and across different environments (Fig. 27). This is expected since there are no restrictions placed on logit output evolution for NTK agents

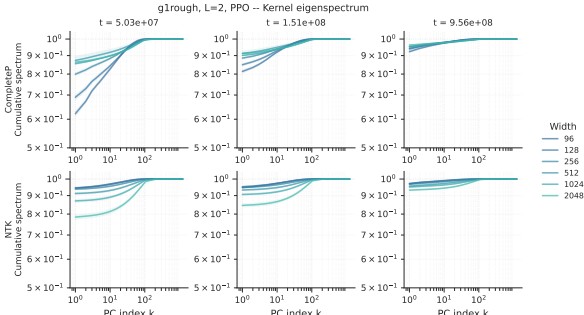

*Figure 5.* **Consistent eigenspectrum at large width across training for CompleteP.** Eigenspectrum of the penultimate layer of the policy network across checkpoints for G1Rough using synthetic dataset. The shape of the spectra from NTK features spreads out more with larger widths, whereas the shape of the spectra from CompleteP features becomes more consistent with larger widths. Both spectra begin converging with training time.

and they can evolve at $\Theta(1)$ (Fig. 21) based on our desiderata 2 (Appendix A). However, the speed of logit evolution is environment specific, with CompleteP agents showing a faster rate of convergence (Fig. 27).

However, Figure 6c shows that NTK agents with different seed initialization converge to policy kernels that are more distinct (Fig. 28), causing a lower kernel alignment despite width scaling. Comparatively, CompleteP agents demonstrate a significantly higher consistency in seed specific policy kernels with width scaling. This is interesting in the reinforcement learning framework as two different sequences of actions that produces the same reward magnitude are equally likely to be reinforced during policy learning due to the reward dependent objective, and there are no constraints to the action sequence taken. Furthermore, the

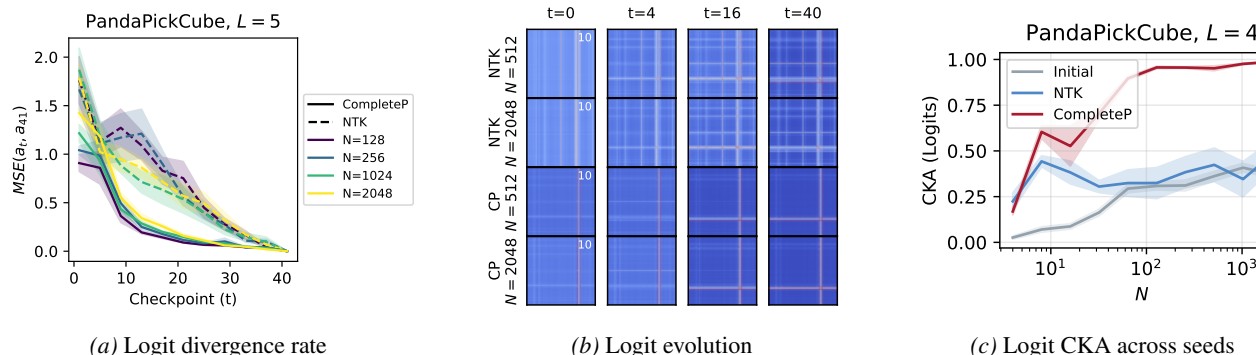

*(a)* Logit divergence rate    *(b)* Logit evolution    *(c)* Logit CKA across seeds

*Figure 6.* **Both NTK and CompleteP show consistent logit evolution, with CompleteP showing seed-specific consistency. (a)** MSE of policy logits between the last and intermediate checkpoints. **(b)** Consistent logit kernel differences across widths. **(c)** CompleteP shows improved kernel consistency across seeds. Shaded area is standard error over 10 seeds.

stochasticity in the action sampling process is not controlled by the seed initialization. Despite these sources of stochasticity, CompleteP agents tend to demonstrate a higher tendency to converge to the same policy, which could be driven by increased feature consistency.

# 7. Improving Reward and Compute Efficiency

Lastly, we want to study how scaling reinforcement learning agents across width and depth using either the NTK or CompleteP parameterizations influence reward maximizing performance. Figure 7a shows that as the agent's width is increased from $N = 4$ to $N = 64$, reward maximization performance increases monotonically for both NTK and CompleteP agents, with a faster increase in performance observed in the latter. However, as the width is further increased from $N = 64$ to $N = 2048$, NTK agents demonstrate the expected model collapse or slower learning performance as in supervised learning framework with the need for additional training steps to achieve similar performance as smaller width (Bordelon & Pehlevan, 2022; Vyas et al., 2023; Noci et al., 2024). Conversely, CompleteP agents demonstrate faster policy convergence as the width is increased, and agents with widths between $N = 256$ and $N = 2048$ show consistent reward maximization behavior, replicating the learning consistency phenomenon also seen in the supervised learning context. We observe the same behavior in the reward maximizing dynamics between the NTK and CompleteP agents across different depths and environments (Refer to Appendix O), with CompleteP agents achieving state-of-the-art performance reported prior (Zakka et al., 2025). Instead, the speed of policy convergence in NTK agents approaches the limiting behavior seen in supervised learning frameworks (Noci et al., 2024; Vyas et al., 2023; Bordelon & Pehlevan, 2022).

Compute is proportional to the number of model parameters multiplied by the number of gradient steps and environment

steps. Since all the other hyperparameters such as the same number of parallel environments, mini-batch size, and trajectory rollout lengths were fixed (Refer to Appendix C), this is a sufficient metric to determine whether the CompleteP parameterization could improve compute efficiency for reinforcement learning agents. Figure 7b shows the average rewards agents of different widths between $N = 4$ to $N = 2048$ achieve on the Humanoid Maze task with different compute budgets. Importantly, the Humanoid Maze task cannot be easily solved by a network with a small number of parameters (Bortkiewicz et al., 2025), as the agent needs to learn to balance, move and reach a target. Hence, scaling parameters matters in this task, and we see that increasing the widths of CompleteP agents improves the maximum rewards attained while NTK agents demonstrate a slower onset of learning with the same compute budget. Refer to Fig. 50 for learning performance against wallclock runtime.

In Figure 7c, we visualize the pareto optimal compute frontier for both CompleteP (pink) and NTK (cyan) agents. We then fit an isotonic regression line to the pareto frontier points. We define compute efficiency as the amount of compute flops needed to achieve 95% of the maximum reward achieved either by the NTK or CompleteP agent. We also define reward efficiency as the amount of rewards attained based on 95% of compute used to train the models. In this task, we observe CompleteP agents requiring a significantly lower amount of compute budget ($\Delta C = -31\%$) to achieve the same amount of reward achieved by NTK agents, and a significantly higher amount of reward ($\Delta R = 33\%$) for the same compute budget, demonstrating the benefits of feature learning in reinforcement learning. Furthermore, inspection of the compute-reward curves across multiple environments demonstrates the compute and reward efficiency of CompleteP over NTK agents. Refer to Appendix O for compute frontiers for other environments and variants.

Initializing network parameters using orthogonal initialization instead of Gaussian distribution improved NTK agent's

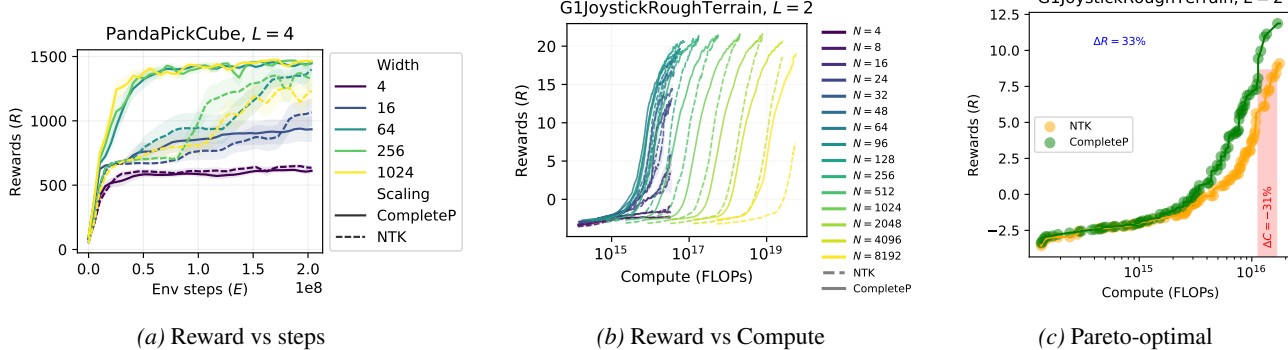

*(a)* Reward vs steps       *(b)* Reward vs Compute       *(c)* Pareto-optimal

*Figure 7.* **CompleteP improves speed of policy convergence and compute efficiency when scaling across different environments. (a)** With NTK width scaling, agents require more interactions and fail to reach peak rewards compared to CompleteP; CompleteP performance is stable even at large widths. **(b)** For a given reward, NTK agents need substantially more FLOPs than CompleteP. **(c)** Pareto frontiers diverge at larger FLOPs (comparison till $10^{16}$ FLOPs), letting CompleteP reach higher reward with the same compute (reward efficiency) or require less compute for the same reward (compute efficiency).

learning performance for some environments (Appendix O.2). Additionally, layer normalization has been shown to improve reward learning performance (Lee et al., 2025). We noticed a mixture of improvements in NTK agents in terms of maximal reward that can be attained at lower compute budgets (Appendix O.3). Despite these techniques, CompleteP agents with orthogonal or layer normalization techniques consistently demonstrated improved reward and compute efficiencies, suggesting that CompleteP parameterization can work synergistically with other methods to improve reinforcement learning performance. Additionally, CompleteP agents demonstrate significantly higher efficiencies in environments with sparse rewards while NTK agents failed to learn the task (Appendix O). Hence, CompleteP parameterization improves compute and reward efficiency as seen in supervised learning contexts (Yang & Hu, 2020; Bordelon & Pehlevan, 2022; Vyas et al., 2023; Noci et al., 2024).

## 8. Discussion

Our results show that scaling deep reinforcement learning agents reveals systematic inconsistencies in representation and policy learning that depend critically on how feature learning is parameterized. We emphasize that the impact of feature learning in the contexts of non-stationary data distributions and feedback loops typically faced in RL problems has not been systematically studied. By contrasting the Neural Tangent Kernel (NTK) and CompleteP parameterizations in actor–critic networks, we isolate how suppressing versus preserving feature learning affects scaling behavior in continuous control reinforcement learning.

A practical finding is that maintaining feature learning enables reliable learning rate hyperparameter transfer across scale. Under CompleteP, learning rates tuned at smaller widths transfer robustly to larger and deeper models, while yielding improved learning performance without costly retuning (Figure 3; Appendix F). In contrast, NTK-parameterized agents exhibit learning curve collapse with scale (Figure 7a; Appendices O, Q). While similar effects have been reported in supervised learning (Yang & Hu, 2020; Bordelon & Pehlevan, 2022; Vyas et al., 2023; Noci et al., 2024), their emergence in reinforcement learning is nontrivial given the policy-dependent data distribution. Our results show that hyperparameter transfer, assumed infeasible in RL, becomes attainable when feature learning is preserved.

CompleteP also enforces a more consistent rate of feature learning across scale. As models grow, NTK parameterization increasingly suppresses representation change, leading to growing discrepancies in learned kernels (Figure 4; Appendix L). By maintaining feature evolution, CompleteP improves alignment of learned representations and action distributions across scales and seeds (Figure 6; Appendix M). These improvements are partial rather than absolute: while CompleteP reduces scale-induced divergence, it does not eliminate inconsistencies intrinsic to reinforcement learning, which arise from exploration noise, stochastic policies, and evolving data distributions (Menache et al., 2005). Accordingly, we do not claim super-consistency (Noci et al., 2024), but rather a mitigation of RL-specific sources of variability.

These representational benefits translate into improved compute and reward efficiency. For a fixed compute budget, CompleteP agents achieve higher returns, whereas NTK agents exhibit diminishing returns to scale (Figure 7d; Appendix R). The gap is largest in high-dimensional tasks such as humanoid locomotion, where NTK agents fail to achieve comparable performance despite equal compute, illustrating that preserving feature learning is a practical requirement for effective RL scaling.

Our study has several limitations that suggest future directions. We focus on residual architectures and on-policy learning; extending this analysis to convolutional architectures for vision-based control (Hilton et al., 2023; Lin et al., 2025). A deeper study of the hyperparameters towards off-policy learning in SAC from Appendix F would inform the relation of these parameterizations between on-policy and off-policy RL. RL-specific hyperparameters, including rollout length and environment parallelism, may further interact with feature-learning parameterizations. More principled scaling analyses could be enabled by converting sigmoidal learning curves into intrinsic performance metrics (Hilton et al., 2023), while multi-agent or non-stationary environments would stress-test consistency limits (Chiappa et al., 2022; Dohare et al., 2024). Finally, comparisons to biological representations may clarify whether similar feature-learning regimes exist in natural systems (Kumar et al., 2024).

Overall, these results position feature-learning–preserving parameterizations such as CompleteP as a key mechanism for enabling hyperparameter transfer, stabilizing feature learning, and improving compute efficiency in scaled reinforcement learning, while clarifying the intrinsic limits imposed by non-stationarity and closed-loop learning.

## Impact Statement

This paper presents work whose goal is to advance the field of machine learning. There are many potential societal consequences of our work, none of which we feel must be specifically highlighted here.

### Ethics and Reproducibility Statement

We have used Large Language Models (LLMs) for assistance in proofreading, writing, and coding.

### Acknowledgments

This work was supported by an NSF GRFP award, Spencer Foundation award number 202600086, NSF CAREER Award (IIS-2239780), DARPA grants DIAL-FP-038 and AIQ-HR00112520041, the Simons Collaboration on the Physics of Learning and Neural Computation, and the William F. Milton Fund from Harvard University. This work has been made possible in part by a gift from the Chan Zuckerberg Initiative Foundation to establish the Kempner Institute for the Study of Natural and Artificial Intelligence.

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

# A. Details of NTK and $\mu P$ Parameterization

## A.1. Primer on Parameterizations (SP, NTK, $\mu P$) and Large width limits for Supervised Learning

Before describing how we apply $\mu P$ for RL training, we first describe how it is implemented in the more classic setting of supervised learning. We will first illustrate the width scaling in multilayer perceptron (MLPs) following arguments similar to those in Bordelon & Pehlevan (2022); Yang & Hu (2020); Vyas et al. (2023).

**Width Scaling**    Consider a depth $L$ multi-layer perceptron $f(x) : \mathbb{R}^D \to \mathbb{R}$ maps inputs $x \in \mathbb{R}^D$ to output predictions $f$ is defined recursively as

$$f(x) = \frac{1}{N^{a_L}} \sum_{i=1}^{N} w_i^L \phi(h_i^L) \, , \; h_i^{\ell+1} = \frac{1}{N^{a_\ell}} \sum_{j=1}^{N} W_{ij}^\ell \phi(h_j^\ell) \, , \; h^1 = \frac{1}{N^{a_0}} \sum_{j=1}^{D} W_{ij}^0 x_j \tag{2}$$

The parameters $\theta = \{w_i^L, \{W_{ij}^\ell\}_{\ell=1}^{L-1}, W^0\}$ are randomly initialized with variances

$$w_i^L \sim \mathcal{N}(0, N^{-2b_L}) \tag{3}$$

$$W_{ij}^\ell \sim \mathcal{N}(0, N^{-2b_\ell}) \, , \; \ell \in \{1, ..., L-1\} \tag{4}$$

$$W_{ij}^0 \sim \mathcal{N}(0, N^{-2b_0}) \tag{5}$$

For a given optimizer (SGD, Adam, etc), we must also (potentially) specify a rule for how to scale the learning rate for each hidden layer with $N$. The parameters $\theta$ are updated with a stochastic gradient of the loss $\mathcal{L}(\theta) = \mathbb{E}_x \ell(\theta, x)$ each step on a minibatch $\mathfrak{B}_t$ drawn from the data distribution. The function $\ell(\theta, x)$ represents the loss associated with data point $x$. The updates take a general form

$$\theta(t+1) = \theta(t) - \mathsf{OPT}\left( \{\eta^\ell\}, \left\{ \frac{\partial}{\partial \theta} \ell(\theta(t), x) \right\}_{x \in \mathfrak{B}_t} \right) \tag{6}$$

where $\mathsf{OPT}$ is a transformation of the gradients $\frac{\partial}{\partial \theta} \ell(\theta, x)$ that is specific to the optimizer (SGD, Adam, etc) and depends on the set of learning rates $\{\eta^\ell\}$. For example, the SGD update is simply

$$W_{ij}^\ell(t+1) = W_{ij}^\ell(t) - \eta_\ell \mathbb{E}_{x \in \mathfrak{B}_t} \frac{\partial}{\partial W_{ij}^\ell} \ell(\theta(t), x). \tag{7}$$

The last scaling criterion is choosing how the learning rates should be updated

$$\eta_\ell = \eta_0 N^{c_\ell} \tag{8}$$

A parameterization is a collection of choices for the scaling exponents for the parameters and learning rates $\{(a_\ell, b_\ell, c_\ell)\}$ and $(d_1, d_2)$ that enable the limit of width $N \to \infty$ with the input dimension $D$, fixed batch size, and fixed number of training steps under the optimizer of choice.

To derive a stable *feature-learning* parameterization, we desire the following properties (consistent with the desiderata of Yang & Hu (2020)) as the width $N$ goes to infinity

**Desiderata:** For reliable and efficient scaling, we require:

1. **Stable signal:** Pre-activation variances $\rho(h^\ell)$ remain $\Theta(1)$ regardless of width $N$ and depth $L$. These can be estimated using simple computation of variances for each hidden variable at initialization.

2. **Stable output learning:** The magnitude of output changes during learning $(\boldsymbol{a}(t) - \boldsymbol{a}(0))$ is $\Theta(1)$, ensuring functional changes are well-behaved as the network scales.

3. **Stable feature learning:** The change in intermediate representations $(\boldsymbol{h}(t) - \boldsymbol{h}(0))$ is separated from the choice of width and depth, and can be controlled from the hyperparameter $\Omega_0$.

| **Params** | $(a_L, b_L, c_L)$ | $(a_\ell, b_\ell, c_\ell)$ | $(a_0, b_0, c_0)$ | **Stable Signal** | **Function Learning** | **Feature Learning** |
|---|---|---|---|---|---|---|
| Standard | $(0, \frac{1}{2}, 0)$ | $(0, \frac{1}{2}, 0)$ | $(0, 0, 0)$ | Yes | No | No |
| NTK | $(\frac{1}{2}, 0, 0)$ | $(\frac{1}{2}, 0, 0)$ | $(0, 0, 0)$ | Yes | Yes | No |
| MF | $(1, 0, 1)$ | $(\frac{1}{2}, 0, 1)$ | $(0, 0, 1)$ | Yes | Yes | Yes |
| $\mu$P | $(\frac{1}{2}, \frac{1}{2}, 0)$ | $(0, \frac{1}{2}, 1)$ | $(0, 0, 1)$ | Yes | Yes | Yes |

*Table 2.* The rules for scaling the model under the **SGD optimizer**, $\mathsf{OPT} = \mathsf{SGD}$. The function learning and feature learning columns indicate that the function (and resp. feature) updates are stable and $\Theta(1)$ as $N \to \infty$. The mean field (MF) and $\mu$P parameterizations are equivalent in their training dynamics.

| Param. | $(a_L, b_L, c_L)$ | $(a_\ell, b_\ell, c_\ell)$ | Stable Signal | Function Learning | Feature Learning |
|---|---|---|---|---|---|
| Standard | $(0, \frac{1}{2}, 0)$ | $(0, \frac{1}{2}, 0)$ | Yes | No | No |
| NTK | $(\frac{1}{2}, 0, -\frac{1}{2})$ | $(\frac{1}{2}, 0, -1)$ | Yes | Yes | No |
| MF/$\mu$P | $(1, 0, 0)$ | $(1, -\frac{1}{2}, 0)$ | Yes | Yes | Yes |

*Table 3.* The rules for scaling the model under the **Adam optimizer**, $\mathsf{OPT} = \mathsf{Adam}$. The function learning and feature learning columns indicate that the function (and resp. feature) updates are stable and $\Theta(1)$ as $N \to \infty$. The mean field (MF) and $\mu P$ parameterizations are equivalent in their training dynamics. Note that $a_0, b_0, c_0$ is similar to $a_L, b_L, c_L$.

We compare various parameterizations and their properties under SGD in Table 2. We repeat this exercise for Adam in Table 3.

We note that the main change in our derivation for accomplishing this with the Adam optimizer comes from modeling the update as a simplified version of Adam without the exponential moving average, SignSGD (Dey et al., 2026):

$$W_{ij}^\ell(t+1) = W_{ij}^\ell(t) + \eta^\ell \frac{1}{Z_{ij}^\ell} g_i^{\ell+1} \phi(h_j^\ell), \tag{9}$$

$$Z_{ij}^\ell = \sqrt{\left(g_i^{\ell+1}\right)^2 \phi\left(h_j^\ell\right)^2}, \tag{10}$$

$$\mathbf{g}^\ell \equiv \frac{\partial \mathcal{L}^\ell}{\partial \mathbf{h}^\ell}. \tag{11}$$

This optimization rule yields the correct scaling rules for Adam, especially Adam at small $\epsilon$ parameter. We note that the $\epsilon$ parameter can also be scaled with $N, L$ as well (Dey et al., 2026).

**Scale of Updates for Normalized GD/Sign-GD/Adam** Under this sign-SGD, the entry-wise weight updates are thus $\Theta(1)$. These updates to the weights induce the following scale updates to the hidden features

$$\Delta \mathbf{h}^{\ell+1} = \eta^\ell N^{-a_\ell} \left( \frac{1}{\mathbf{Z}^\ell} \odot \mathbf{g}^{\ell+1} \left(\boldsymbol{\phi}^\ell\right)^\top \right) \boldsymbol{\phi}^\ell + N^{-a_\ell} \mathbf{W}^\ell \Delta \boldsymbol{\phi}^\ell \sim \Theta\left(\eta^\ell N^{1-a_\ell}\right) \tag{12}$$

where the last expression should be understood as the scale of each entry in the vector. If we want the features to have non-negligible entry-wise evolution, then we must demand that $\eta^\ell \sim N^{a_\ell-1}$. This is achieved in the $\mu$P scaling where $a_\ell = 1$ and $\eta_\ell = \Theta(1)$. We note that in the NTK scaling, that $a_\ell = \frac{1}{2}$ and $\eta_\ell = \Theta(N^{-1})$ leading to a $\Delta h = \Theta(N^{-1/2})$. This indicates that internal features become static in the NTK scaling as $N \to \infty$.

## A.2. Output Rescaling as an Equivalent Route to NTK from MF/$\mu$P

In the previous section, we described how different parameterizations can be implemented with SGD or Adam optimizers. In this section, we connect our analysis to the implementation we provided in Table 1. To do so, we introduce an output scale multiplier $\Omega$ to the mean-field/$\mu$P parameterization for Adam

$$f = \frac{1}{\Omega N} \sum_{i=1}^N w_i^L \phi(h_i^L), \quad h_i^{\ell+1} = \frac{1}{N} \sum_{j=1}^N W_{ij}^\ell \phi(h_j^\ell), \quad h_i^1 = \frac{1}{D} \sum_{j=1}^D W_{ij}^0 x_j. \tag{13}$$

The rule for Adam $\mu$P/MF scaling is

$$\text{Mean Field / } \mu\text{P:} \quad \Omega, \eta^\ell = \Theta(1), \tag{14}$$

recovering the usual $\mu$P scaling for Adam (Yang & Hu, 2020; Dey et al., 2026). This parameterization achieves constant scale output and feature updates $\Delta f = \Theta(1)$ and $\Delta h_i^\ell = \Theta(1)$. However, we can alter the relative rate of internal weight updates and network outputs by decreasing $\Omega$ and simultaneously decreasing the learning rate. For Adam, the rescaling rule that maintains constant scale changes in $f$ at the first step of training is

$$\text{Learning Rate Under Output Rescaling (Adam):} \quad \eta = \Omega \times \eta_0 \,, \ \Omega \to 0 \tag{15}$$

where $\eta_0$ is a constant. To see this, consider

$$\Delta f = \frac{1}{\Omega N} \Delta \boldsymbol{w}^L \boldsymbol{\phi}^L + \frac{1}{\Omega N} \boldsymbol{w}^L \Delta \boldsymbol{\phi}^L = \Theta\left(\frac{\eta}{\Omega}\right) = \Theta(1) \tag{16}$$

$$\Delta \boldsymbol{h}^\ell = \Theta(\Omega) \tag{17}$$

In the proper $\Omega \to 0$ limit, the internal features $\boldsymbol{h}^\ell$ are static while the network outputs still have $\Theta(1)$ movement. We recover NTK parameterization (up to symmetries in the definition of the parameterization (Yang & Hu, 2020; Dey et al., 2026))

$$\text{NTK Parameterization:} \quad \Omega = \Omega_0 N^{-1/2} \,, \ \eta = \eta_0 \Omega_0 N^{-1/2} \tag{18}$$

This is equivalent to the NTK parameterization in the previous Appendix as it has identical output and feature scale dynamics

$$\text{NTK Output and Feature Changes:} \quad \Delta f = \Theta(1) \,, \ \Delta h_i^\ell = \Theta(N^{-1/2}) \tag{19}$$

This is equivalent to the output and feature changes achieved using the $(a_\ell, b_\ell, c_\ell) = (\frac{1}{2}, 0, -1)$ since it induces equivalent scale changes to the outputs and the internal features.

## A.3. Depth Scaling

Following Bordelon et al. (2024b); Yang et al. (2024b), we study depth scaling of deep residual networks

$$h_i^{\ell+1} = h_i^\ell + L^{-\alpha} N^{-a_\ell} \sum_{j=1}^N W_{ij}^\ell \phi(h_j^\ell) \tag{20}$$

Now we can ask which multipliers $\alpha$ and learning rate scalings $\eta_\ell$ allow for stable signal propagation, stable function updates, and stable feature updates.

**Learning Rate Depth Scalings for Adam** We will focus on the Adam optimizer where $a_\ell = 1$ for both NTK and $\mu$P scalings. The key desideratum is that the update to each branch of the residual stream is $\Theta(L^{-1})$ under each step of the optimizer. To achieve this, we set

$$\eta^\ell = \Theta\left(L^{-1+\alpha}\right), \tag{21}$$

so that the updates accumulate a total $\Theta(1)$ change along the entire residual stream.

## A.4. Training with Bias Parameters

Here, we consider the effect of trainable bias parameters

$$h_i^{\ell+1} = h_i^\ell + \frac{1}{L^\alpha} \left[ \frac{1}{N^{a_\ell}} \sum_{j=1}^N W_{ij}^\ell \phi(h_j^\ell) + b_i^\ell \right] \tag{22}$$

The bias parameters are initialized with

$$\text{Var}(b_i^\ell) = \Theta_{N,L}(1). \tag{23}$$

The biases need to move by $\Delta b_i^\ell = \Theta_{N,L}(L^{-1+\alpha})$. We summarize the learning rates needed to achieve this change in Table 4.

| Param. | Bias Learning Rate SGD | Bias Learning Rate Adam |
|--------|------------------------|-------------------------|
| NTK | $\Theta(L^{-1+2\alpha})$ | $\Theta(L^{-1+\alpha})$ |
| MF | $\Theta(L^{-1+2\alpha}N)$ | $\Theta(L^{-1+\alpha})$ |

*Table 4.* How bias parameters are scaled for SGD and Adam.

## A.5. Standard Parameterization, LeCun Scalings

Following standard practice (Klambauer et al., 2017) and removing activation scaling by width and depth, we get the following architecture:

$$
\begin{aligned}
h_\mu^{(0)} &= W^{(\mathrm{in})}x_\mu + b^{(\mathrm{in})}, \\
h_\mu^{(\ell+1)} &= h_\mu^{(\ell)} + W^{(\ell)}\phi\big(h_\mu^{(\ell)}\big) + b^{(\ell)}, \\
h_\mu^{(\mathrm{out})} &= W^{(\mathrm{out})}\phi\big(h_\mu^{(L)}\big) + b^{(\mathrm{out})}, \\
f_\mu &= h_\mu^{(\mathrm{out})}.
\end{aligned}
\tag{24}
$$

With LeCun normal scalings, the variance of the weights are drawn based on their "fan in" or input dimensionality. In our architecture that translates to the following:

$$
\begin{aligned}
W_{ij}^{\mathrm{in}} &\sim \mathcal{N}\big(0, \tfrac{1}{D}\sigma_W^2\big), \\
W_{ij}^{\ell}, W_{ij}^{\mathrm{out}} &\sim \mathcal{N}\big(0, \tfrac{1}{N}\sigma_W^2\big), \\
b_i^{\mathrm{in}}, b_i^{\ell}, b_i^{\mathrm{out}} &= 0.
\end{aligned}
\tag{25}
$$

With LeCun uniform, the only change is that the distribution sampled from is uniform. The bounds are setup such that the variance still corresponds to $\frac{1}{\mathrm{fan\_in}}\sigma_W^2$. Biases are still initialized from zero.

$$
\begin{aligned}
W_{ij}^{\mathrm{in}} &\sim \mathcal{U}\left(-\sigma_W\sqrt{\tfrac{3}{D}},\, \sigma_W\sqrt{\tfrac{3}{D}}\right), \\
W_{ij}^{\ell}, W_{ij}^{\mathrm{out}} &\sim \mathcal{U}\left(-\sigma_W\sqrt{\tfrac{3}{N}},\, \sigma_W\sqrt{\tfrac{3}{N}}\right).
\end{aligned}
\tag{26}
$$

## B. Synthetic Observation Dataset for Feature Analysis

For feature and policy analysis, we constructed a synthetic evaluation dataset by varying each observation dimension individually across a consistent range while fixing all other dimensions at zero. Specifically, for an environment with $D$ observation dimensions, we select $S$ linearly spaced values between $-3$ and $3$ for each feature. For each dimension, $S$ samples are generated where only that dimension is varied and all others are set to zero, producing a total of $SD$ samples.

This approach matches the input range encountered by agents during training, as we utilize running observation normalization (Freeman et al., 2021). The synthetic dataset provides a controlled basis for evaluating feature and kernel responses along single input axes, and supports consistent comparisons across network width, depth, and seeds.

**Example:** Suppose the observation has $D = 2$ dimensions and we choose $S = 7$ samples per dimension, the varied values are $-3$, $-2$, $-1$, $0$, $1$, $2$, and $3$. The resulting dataset contains $14$ samples in total. Each sample is a column in the table below corresponding to one synthetic input vector.

*Table 5.* Synthetic evaluation dataset for two features ($D = 2$), with each feature dimension varied over 7 values ($S = 7$); other features set to zero. **Each column represents one sample (input vector).**

| | Vary Feature 1 | | | | | | | Vary Feature 2 | | | | | | |
|---|---|---|---|---|---|---|---|---|---|---|---|---|---|---|
| Sample # | 1 | 2 | 3 | 4 | 5 | 6 | 7 | 8 | 9 | 10 | 11 | 12 | 13 | 14 |
| Feature 1 | $-3$ | $-2$ | $-1$ | 0 | 1 | 2 | 3 | 0 | 0 | 0 | 0 | 0 | 0 | 0 |
| Feature 2 | 0 | 0 | 0 | 0 | 0 | 0 | 0 | $-3$ | $-2$ | $-1$ | 0 | 1 | 2 | 3 |

The full dataset is created by concatenating these samples across all $D$ dimensions. This enables direct and interpretable comparison of how each network responds to changes along individual input axes across all experiments.

# C. Reinforcement Learning Environment Details

We provide additional details of the control environments used. For environment configurations we use the ones provided by (Zakka et al., 2025). We also utilize domain randomization for G1 Humanoid, and access to a privileged state.

| Environment Name | Action Dim | Policy Obs Dim | Value Obs Dim |
|---|---|---|---|
| CartpoleSwingup | 1 | 5 | 5 |
| CartpoleSwingupSparse | 1 | 5 | 5 |
| PandaPickCube | 8 | 66 | 66 |
| G1JoystickRoughTerrain | 29 | 103 | 216 |
| CheetahRun | 6 | 17 | 17 |
| SwimmerSwimmer6 | 5 | 25 | 25 |
| AcrobotSwingup | 1 | 6 | 6 |
| AcrobotSwingupSparse | 1 | 6 | 6 |
| humanoid | 17 | 271 | 271 |
| humanoid_u_maze | 17 | 271 | 271 |
| Craftax-Symbolic-v1 | 17 (discrete) | 1345 | 1345 |

*Table 6.* Dimensionalities for actions and observations across environments.

*Table 7.* Base PPO hyperparameters for key environments

| Parameter | CartpoleSwingup | PandaPickCube | G1JoystickRoughTerrain | Craftax-Symbolic-v1 |
|---|---|---|---|---|
| env_name | CartpoleSwingup | PandaPickCube | G1JoystickRoughTerrain | Craftax-Symbolic-v1 |
| policy_obs_key | state | state | state | state |
| value_obs_key | state | state | privileged_state | state |
| $\Omega$ | 1.0 | 1.0 | 1.0 | 1.0 |
| activation | relu | relu | relu | relu |
| num timesteps | 100,000,000 | 200,000,000 | 2,000,000,000 | 100,000,000 |
| num evals | 21 | 41 | 41 | 21 |
| num envs | 2048 | 2048 | 32,768 | 1024 |
| num minibatches | 32 | 32 | 32 | 4 |
| update epochs | 16 | 8 | 5 | 4 |
| unroll length | 30 | 10 | 32 | 16 |
| minibatch size | 1024 | 512 | 1024 | 1024 |

*Table 8.* Base SAC Hyperparameters for key environments

| Parameter | CartpoleSwingup | CheetahRun | AcrobotSwingup | Craftax-Symbolic-v1 |
|---|---|---|---|---|
| env_name | CartpoleSwingup | CheetahRun | AcrobotSwingup | Craftax-Symbolic-v1 |
| policy_obs_key | state | state | state | state |
| q_obs_key | state | state | state | state |
| $\Omega$ | 1.0 | 1.0 | 1.0 | 1.0 |
| activation | relu | relu | relu | relu |
| num timesteps | 50,000,000 | 100,000,000 | 500,000,000 | 50,000,000 |
| num evals | 50 | 50 | 50 | 21 |
| num envs | 128 | 128 | 128 | 128 |
| batch size | 512 | 512 | 512 | 2048 |
| grad updates per step | 8 | 8 | 8 | 1 |
| min replay size | 8,192 | 8,192 | 8,192 | 8,192 |
| max replay size | 4,194,304 | 4,194,304 | 4,194,304 | 300,000 |
| $\tau$ | 0.005 | 0.005 | 0.005 | 0.005 |
| discounting | 0.99 | 0.99 | 0.99 | 0.99 |
| reward scaling | 1.0 | 1.0 | 1.0 | 1.0 |
| q_network_layer_norm | true | true | true | true |

# D. PPO Implementation Details

For all PPO experiments, we used the PPO implementation provided by (Freeman et al., 2021). While this implementation follows the original PPO objective and update rules of (Schulman et al., 2017), it has a few modifications to maximize throughput on GPU accelerators. One key difference is the collection of environment trajectories in fixed-size chunks (as specified by the unroll length hyperparameter) across each parallel environment. This enables the computation of the Generalized Advantage Estimation within each parallel chunk. Among other tricks, this allows for the compilation of the reinforcement learning algorithms in tandem with the simulator. These differences do not modify the PPO clipped objective, policy update rule, or value function learning.

# E. FLOPs Derivation

We calculate our FLOPs based on the PPO implementation in Freeman et al. (2021) and following convention from (Hilton et al., 2023). We describe our quantities below.

For the resnet style architecture, we estimate FLOPs per forward pass based on the parameter count.

To count parameters we count the main quantity $LN^2$. Biases are neglible in size.

**Definitions**

`minibatch_size` Number of environment steps in each minibatch fed to the optimizer

`num_minibatches` How many minibatches are processed per batch collection.

`unroll_length` (Additional) trajectory length collected for the generalized advantage estimation

`num_timesteps` Total environment steps gathered over training

`num_evals` Number of checkpoints for evaluation (including the initial one).

`update_epochs` Times the same batch is reused inside a PPO update (e.g. policy, value, and entropy losses).

`flops_per_forward_pass` FLOPs required for one network forward pass.

**Total model cost, PPO**

$$\text{env\_steps\_per\_train\_step} = \text{minibatch\_size num\_minibatches}$$
$$\cdot \text{unroll\_length}, \tag{27}$$

$$\text{num\_train\_steps\_per\_epoch} = \left\lceil \frac{\text{num\_timesteps}}{(\text{num\_evals} - 1)\,\text{env\_steps\_per\_train\_step}} \right\rceil, \tag{28}$$

$$\text{unique\_env\_steps} = \text{num\_train\_steps\_per\_epoch env\_steps\_per\_train\_step}, \tag{29}$$

$$\text{total\_model\_flops} = 3\,\text{flops\_per\_forward\_pass update\_epochs}$$
$$\cdot \text{unique\_env\_steps}. \tag{30}$$

Equation (30) gives the FLOP budget for one training epoch; summing over epochs yields the overall training cost.

# F. Learning Rate Hyperparameter Transfer

Scaling deep reinforcement learning—especially for large, high-dimensional embodied intelligence tasks (Gupta et al., 2021; Duan et al., 2022; Zador et al., 2023)—is often bottlenecked by the need to sweep learning rate hyperparameters for each new model width, depth, and environment. For NTK and Standard parameterizations, the optimal learning rate changes with scale, requiring repeated and expensive searches that multiply total compute cost.

In contrast, our results (Figure 3) show that CompleteP parameterization enables consistent transfer of a single optimally-tuned learning rate $\eta_0^*$ from small to large models, across both simple and challenging tasks. This removes the need for repeated tuning and provides stable performance gains as models scale.

While NTK and Standard Parameterization agents can match CompleteP performance, they require exhaustive grid search for every width, depth and learning rate configuration, which is impractical for large-scale, generalist RL. CompleteP, by reducing this overhead, offers a practical and scalable solution in line with scaling laws that predict continued gains with increased capacity (Hilton et al., 2023; Rybkin et al., 2025; Fu et al., 2026).

In summary, the ability to reliably transfer learning rate hyperparameters across scales and environments is a distinct practical advantage of CompleteP, streamlining RL research and supporting the efficient training of ever-larger agents.

## F.1. Learning Rate Transfer over Width and Depth

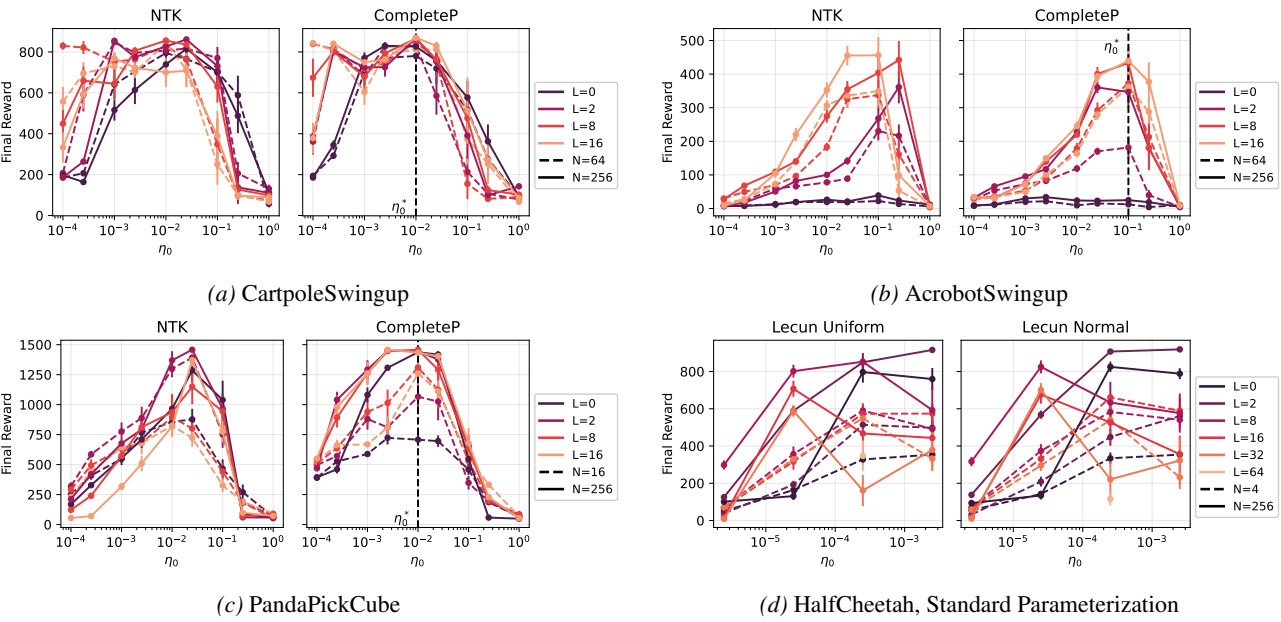

*Figure 8.* **Learning rate transfers more consistently across width and depth with CompleteP.** Agents scaled using the NTK, CompleteP parameterizations were evaluated to determine if a single learning rate could transfer across agents with different depths ($L$) and widths ($N$). Increasing the depth required lowering the learning rate for NTK agents. Instead, CompleteP agents show a tighter consistency in learning rates across depth and width, with deeper networks showing improved final reward performance, suggesting that the learning rate hyperparameter could transfer across depth with the proposed parameterization. Additionally, we also observe improved stability across learning rates with CompleteP. We see a weaker consistency in learning rate transfer when we initialize our residual agents using standard parameterizations such as Lecun Uniform and Lecun Normal. Error bars indicate standard error across 10 seeds.

## F.2. Learning Rate Transfer Across Width with Standard Parameterization

We report PPO learning rate sweeps across width $N$ at fixed depth $L = 2$ for CompleteP, NTK, and Standard Parameterization (SP) on three environments. We report both the base learning rate $\eta_0$ (Figure 9) and the raw learning rate $\eta$ (Figure 10).

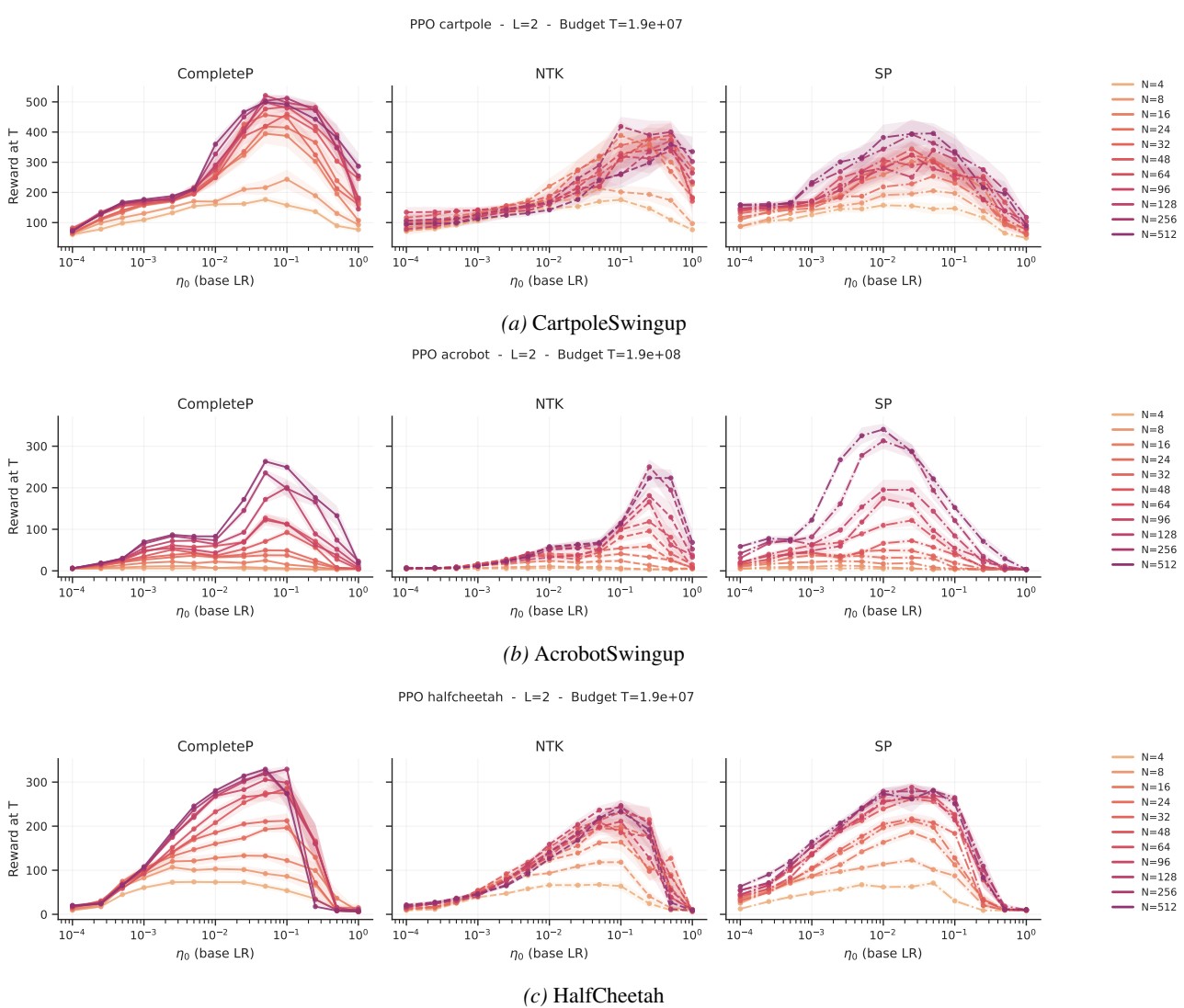

*(a)* CartpoleSwingup

*(b)* AcrobotSwingup

*(c)* HalfCheetah

*Figure 9.* **PPO learning rate transfer across width under CompleteP, NTK, and Standard Parameterization (SP).** Final reward at the end of training as a function of the base learning rate $\eta_0$, for widths $N \in \{4, \dots, 512\}$ at fixed depth $L = 2$.

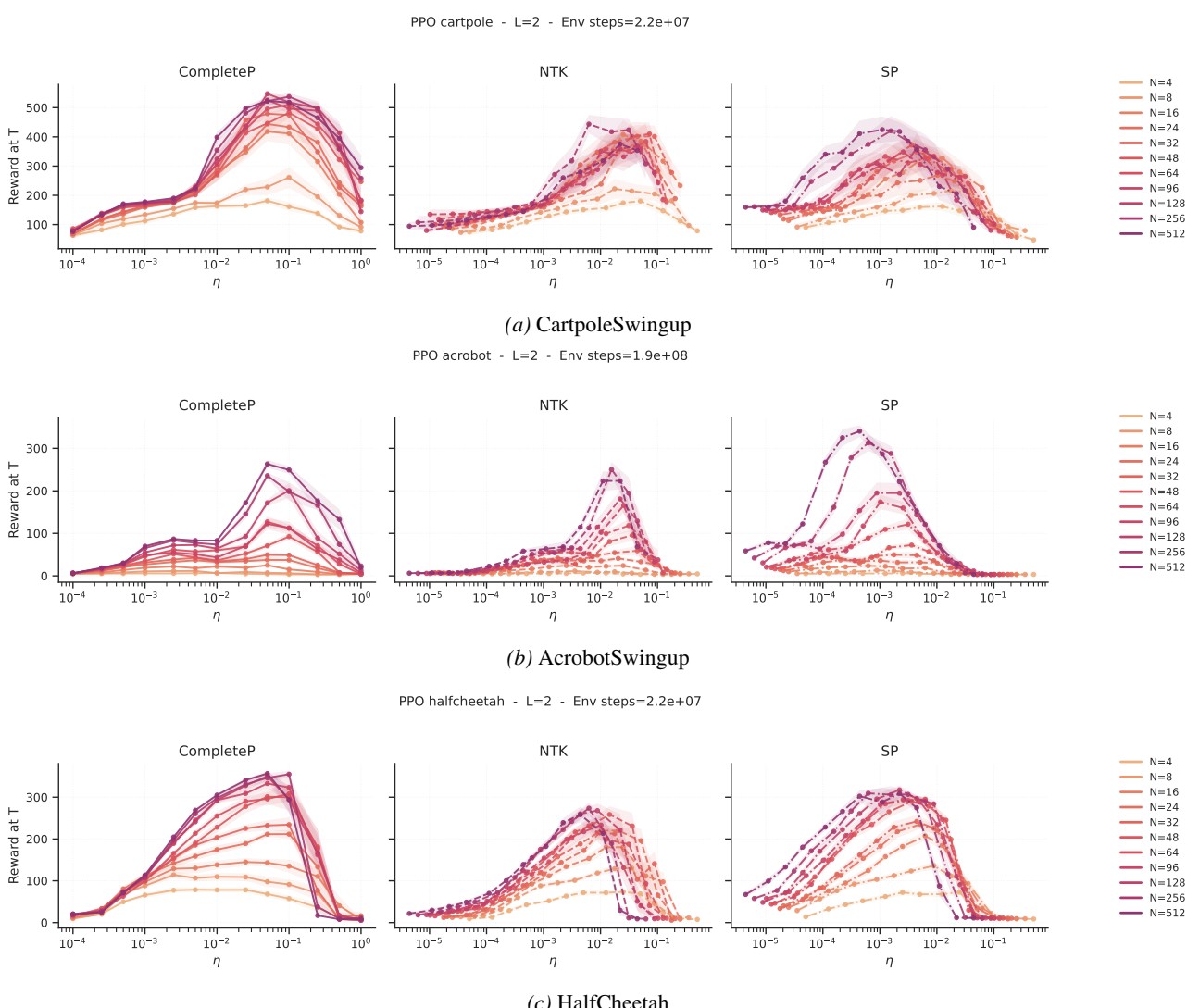

*Figure 10.* **PPO learning rate transfer across width under CompleteP, NTK, and Standard Parameterization (SP), shown against the raw learning rate** $\eta$**.** Final reward at the end of training as a function of the raw learning rate $\eta$, for widths $N \in \{4, \ldots, 512\}$ at fixed depth $L = 2$.

## F.3. Raw Learning Rate Transfer Across Width with SAC

We additionally evaluate whether the learning rate transfer pattern, established for the on-policy PPO setting, persists for off-policy Soft Actor-Critic (SAC). Experience replay in SAC smooths the rate of data distribution shift relative to PPO, providing a complementary test of whether the transfer phenomenon is specific to on-policy dynamics. We sweep the base learning rate $\eta_0$ across widths $N$ at fixed depth $L = 2$ for CompleteP, NTK, and Standard Parameterization (SP) on three environments. We report this sweep against both the base learning rate $\eta_0$ (Figure 11) and the raw learning rate $\eta$ (Figure 12).

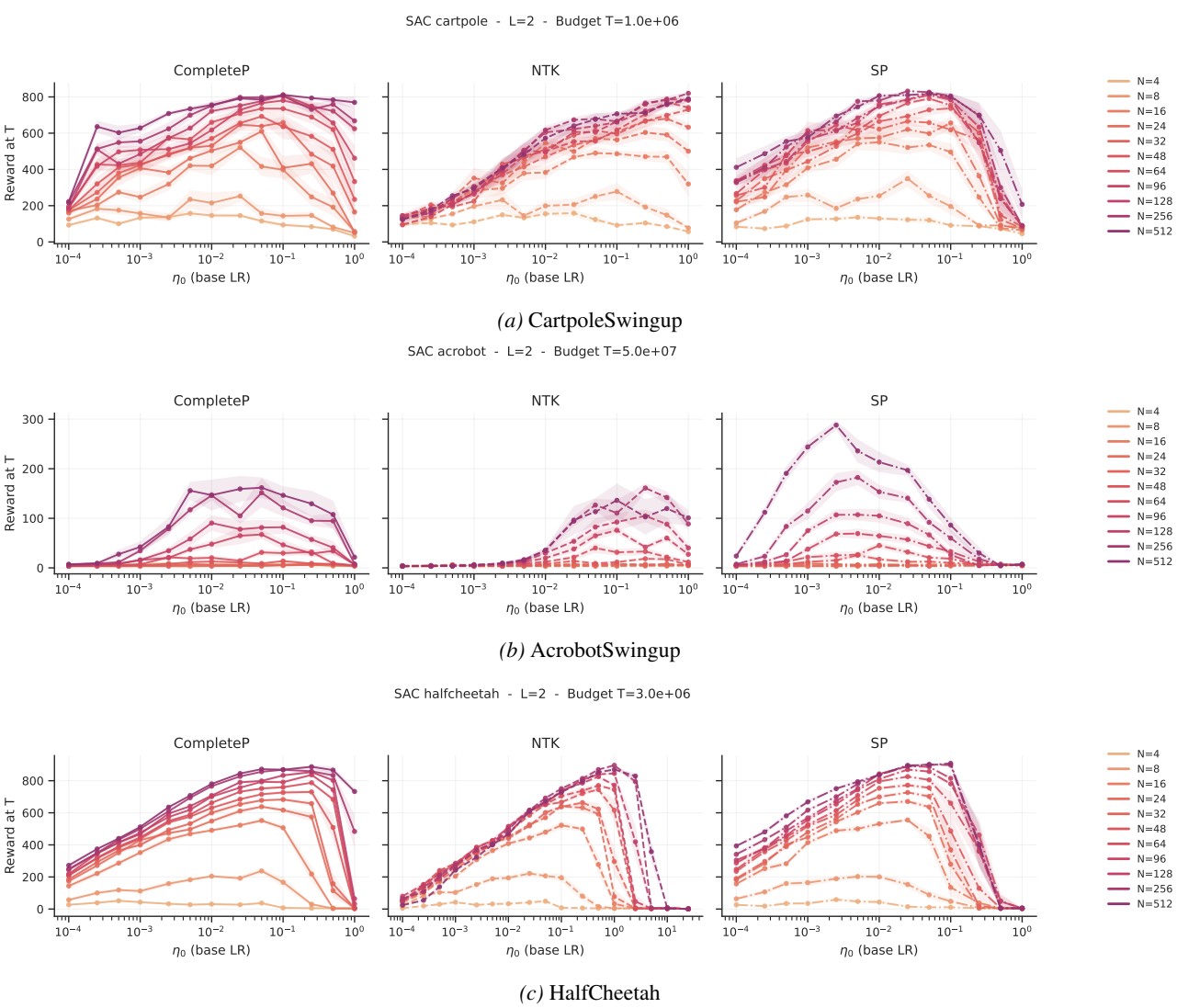

*(a)* CartpoleSwingup

*(b)* AcrobotSwingup

*(c)* HalfCheetah

*Figure 11.* **SAC learning rate transfer across width under CompleteP, NTK, and Standard Parameterization (SP).** Final reward at the end of training as a function of the base learning rate $\eta_0$, for widths $N$ at fixed depth $L = 2$ under off-policy SAC. Shaded area is standard error over 10 seeds.

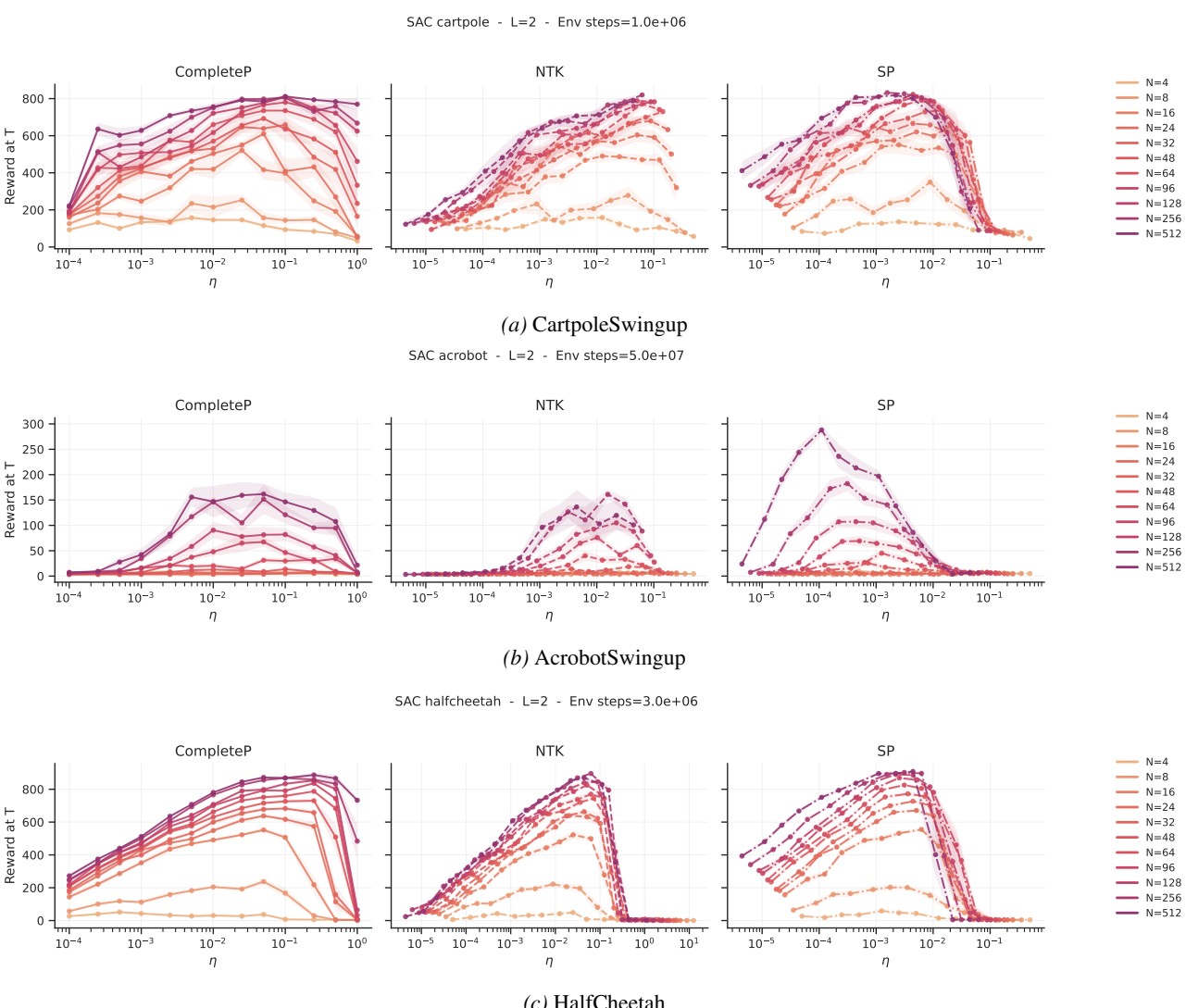

*Figure 12.* **SAC learning rate transfer across width under CompleteP, NTK, and Standard Parameterization (SP), shown against the raw learning rate** $\eta$**.** Final reward at the end of training as a function of the raw learning rate $\eta$, for widths $N$ at fixed depth $L = 2$ under off-policy SAC. Shaded area is standard error over 10 seeds.

# G. Optimal Feature Learning Hyperparameter for RL

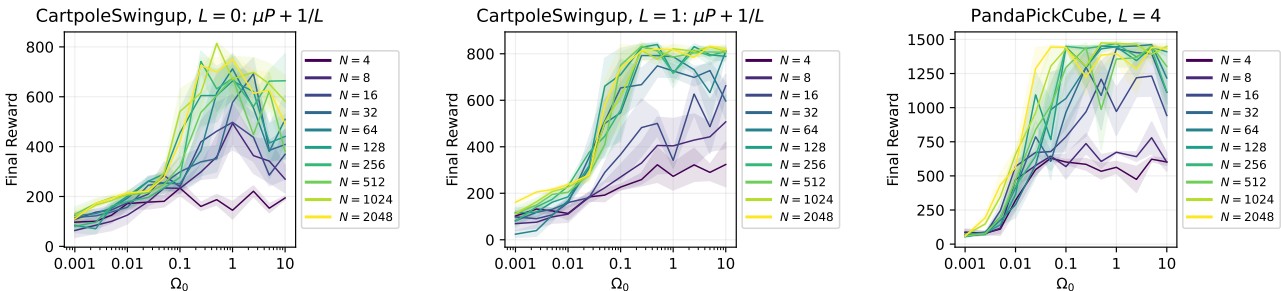

*Figure 13.* **RL performance as a function of the feature learning hyperparameter** $\Omega_0$**.** (**Left**) For shallow networks ($L = 0$), large $\Omega_0$ ($> 5$) leads to frequent model collapse and degraded performance, especially for large width. (**Middle**) Increasing the network depth to $L = 1$ markedly stabilizes learning for larger $\Omega_0$, enabling agents to maintain high final reward. (**Right**) In PandaPickCube with deeper networks ($L = 4$), performance improves with increasing $\Omega_0$ for wide agents with some signs of model collapse when $\Omega_0 > 5$. Together, these results suggest that while rich feature learning generally benefits RL, its stability depends on task, depth, and width.

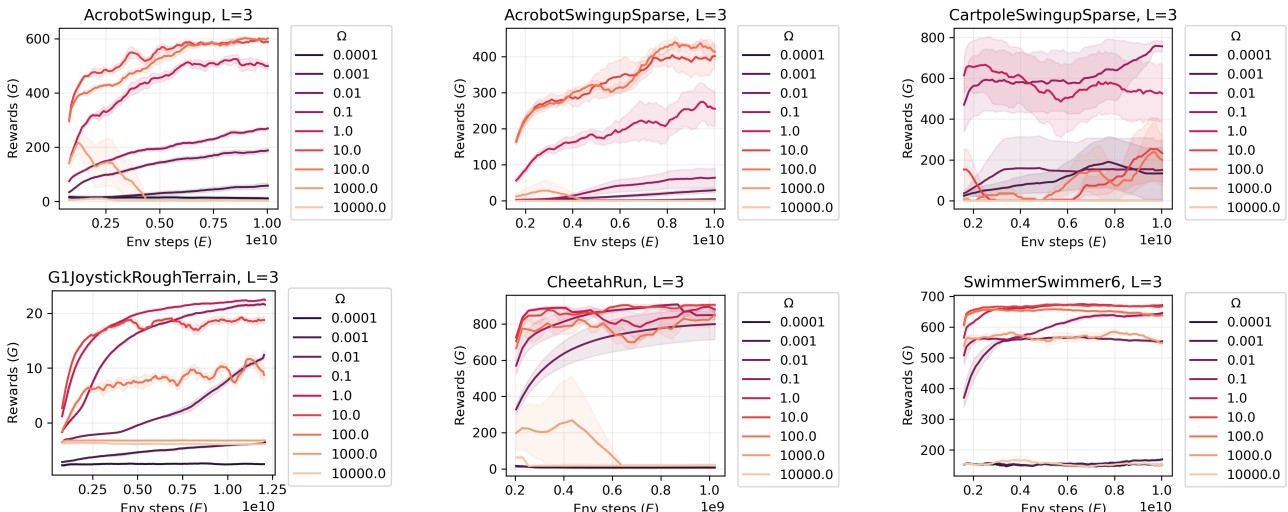

*Figure 14.* **RL performance across environment steps as a function of feature learning hyperparameter** $\Omega_0$**.** Each panel shows the $\Omega_0$ sweep for a different environment. Different environments vary in their range of "optimal" feature learning.

Prior work ([Graldi et al.](#), [2025](#)) highlighted that rich feature learning, facilitated by large values of the richness hyperparameter $\Omega_0$, may not always benefit continual learning; in fact, increased feature plasticity can accelerate catastrophic forgetting of useful structure. Motivated by these findings, we investigated whether promoting richer feature learning (by increasing $\Omega_0$) improves or hinders reinforcement learning performance.

Figure 13 summarizes the relationship between $\Omega_0$, network width, depth, and final agent reward on two tasks. When the network has minimal depth ($L = 0$), setting $\Omega_0 > 5$ leads to frequent model collapse and substantially reduced final performance, suggesting that unchecked feature plasticity can be harmful without sufficient network depth. Interestingly, even a single additional residual layer ($L = 1$) mitigates collapse, making training larger widths more robust to larger $\Omega_0$ and leading to a performance plateau for $\Omega_0 \geq 1$. This behavior is consistent for the harder PandaPickCube task.

These observations lead to several key insights. First, richer feature learning is generally advantageous for RL agents on both simple and complex environments, provided the network has sufficient depth to absorb the increased feature evolution. Second, model collapse at shallow depths with large $\Omega_0$ highlights the nuanced interaction between feature plasticity and representational stability; rigorous characterization of when and why this occurs is an important direction for future work. Third, the optimal degree of feature richness, as controlled by $\Omega_0$, is highly dependent on both task and architecture: neither excessive rigidity nor excessive plasticity is universally optimal.

## H. Reducing Policy Shift Constraint for Learning

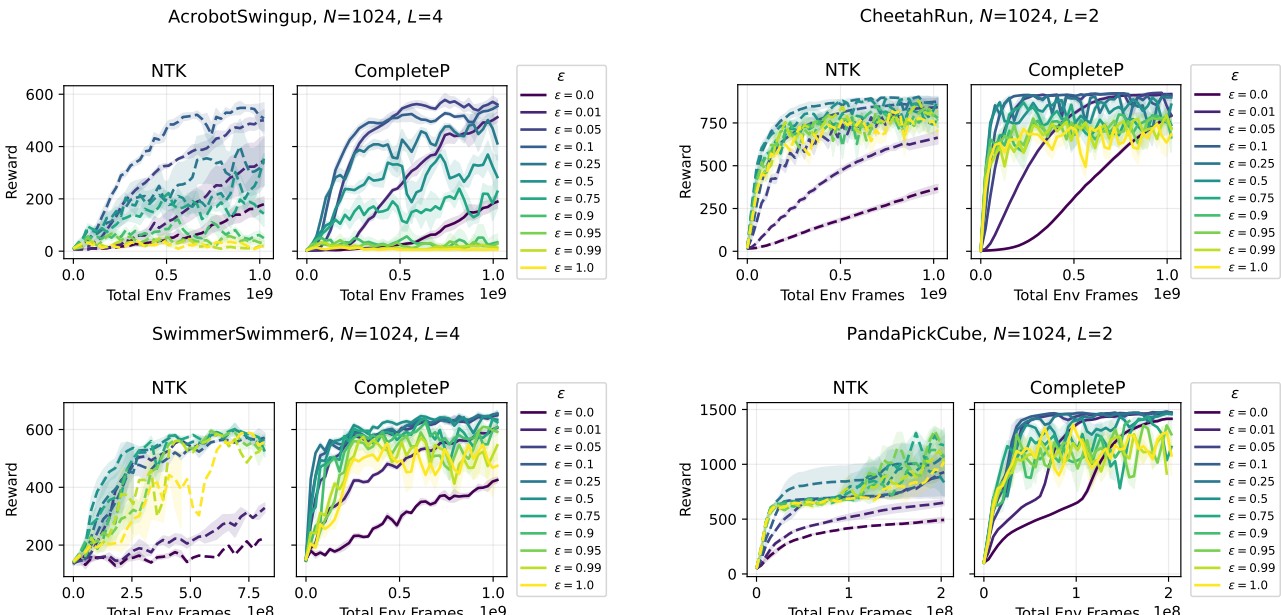

*Figure 15.* **RL performance as a function of the epsilon clipping hyperparameter $\epsilon$.** Larger $\epsilon$ values allow a greater magnitude of change in model parameters, causing the policy to change to a greater extent after each gradient update.

Here we study the role that the clipping parameter in PPO plays in the process, and whether it changes our results regarding the presence of feature learning. One can view smaller values of clipping as a way to constrain the change in the policy between optimization steps. From observing 15 we argue that across both NTK and CompleteP agents the modulation of $\varepsilon$ has roughly the same effect on both NTK and CompleteP performance.

# I. Non-Stationary Data Distributions in Reinforcement Learning

A central challenge unique to reinforcement learning, as compared to supervised learning, is the coupling between policy and environment. The policy not only determines the agent's actions, but directly influences what data trajectories are sampled from the environment. Every policy update therefore changes the future data distribution, which in turn affects future learning. This tight policy-environment loop leads to non-stationary—and initially unpredictable—data distributions throughout training, complicating both analysis and optimization.

To characterize and visualize this effect, Figure 16 depicts the changes in the statistical moments of observed data distributions as training progresses. For each model checkpoint $t$, we roll out 30 trajectories and compute the first four moments (mean, variance, skewness, and excess kurtosis) of the observations collected. We then measure the change in each moment between consecutive checkpoints $t$ and $t + 1$. This set of observations are not the synthetically generated dataset. This analysis allows us to quantify the rate and nature of data distribution shifts induced by policy updates.

The results show that, for both CheetahRun and PandaPickCube tasks, the rate of change in the mean and variance of the observation distribution is highest at the beginning of training, but gradually decreases over time. This trend reflects the learning process: initially, an untrained random policy generates highly diverse, failure-prone trajectories. As training proceeds and the policy improves, the distribution of sampled trajectories incorporates a growing fraction of successful behaviors. In later stages, as the agent approaches and maintains a near-optimal policy, the sampled data becomes more stereotyped—the mean and variance stabilize, indicating less dramatic distributional change.

The evolution of skewness and excess kurtosis provides additional insights. Large changes in these higher moments early in training suggest the distribution of observations is highly non-Gaussian with significant outliers, corresponding to rare events or failure states. As learning progresses, these statistics also generally decrease, indicating a reduction in the frequency and severity of such outliers and a move toward a more regular, lower-entropy observation distribution as the agent consistently succeeds.

Interestingly, this overall convergence trend is observed across all tested model widths, but appears somewhat faster for larger networks. This could indicate that wider models more rapidly converge on stable, successful policies, thus reducing the non-stationarity of the data distribution they induce at the later stages of learning.

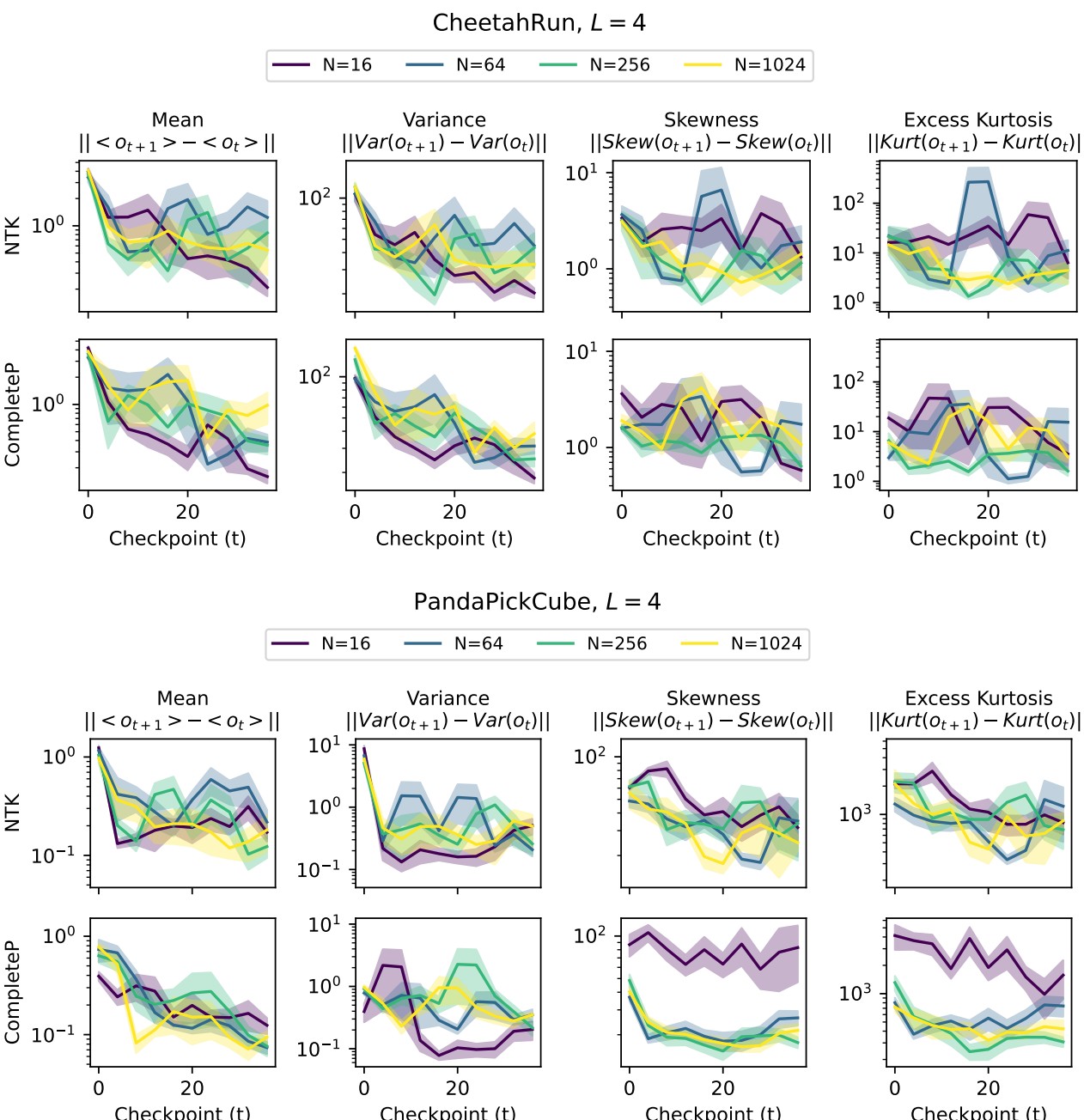

*Figure 16.* **Observation data distribution changes with policy evolution.** We plot the checkpoint-to-checkpoint difference in the first four moments (mean, variance, skewness, excess kurtosis) of the normalized observation data for two tasks and four model widths. Both NTK and CompleteP parameterizations show rapid changes early in training and stabilize as the policy converges. Larger models display slightly faster convergence toward a stable data distribution.

# J. Loss of Plasticity and Dormant Neurons in RL Agents

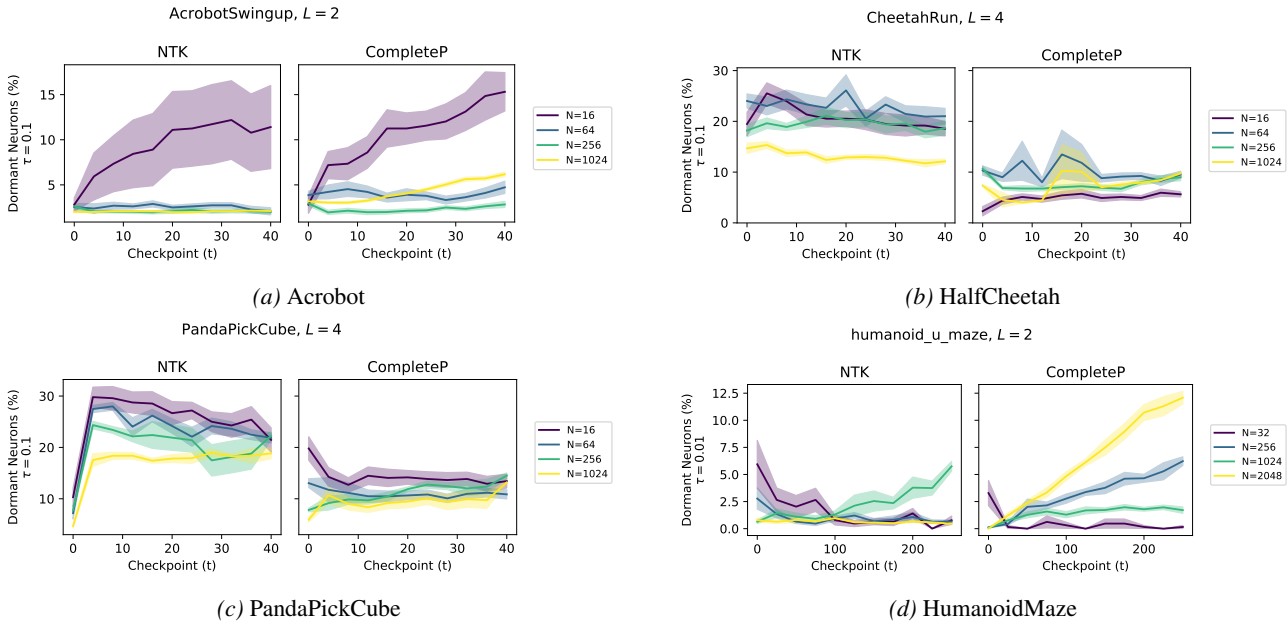

*Figure 17.* **Dormant neurons increase in CompleteP.** Both NTK and CompleteP agents display fewer dormant neurons as width increases. However, note that larger width NTK agents have a reduced extent of feature learning and hence will be less prone to the dormant neurons phenomenon whereas feature learning is consistent in CompleteP agents.

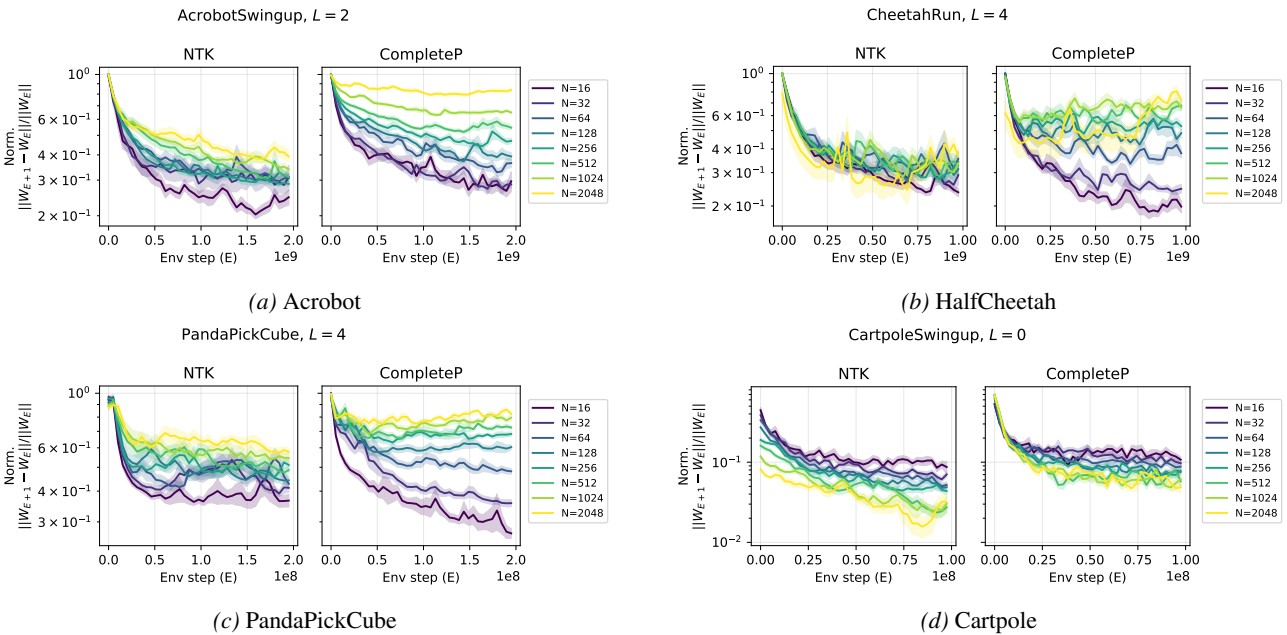

*Figure 18.* **Weight change magnitude across learning.** In NTK, the weight change magnitude drops with width and across training, indicating loss of plasticity. By contrast, muP (CompleteP) yields stable weight dynamics for wider models, supporting ongoing adaptability.

A challenge in training neural networks, particularly in reinforcement and continual learning tasks, is the increase of activation sparsity over time. During learning, some neurons become "dormant": They fail to activate, with values shrinking toward zero and contributing neglible gradients during backpropagation (Sokar et al., 2023). This loss of plasticity can be

particularly problematic in reinforcement learning, where continual adaptation to non-stationary data is required. When a significant proportion of neurons become dormant, the capacity of the model to adapt to new data distributions diminishes, often leading to learning collapse. Prior works have attempted to mitigate this effect through periodic reinitialization of dormant neurons (Dohare et al., 2024).

Here, we compare NTK and CompleteP parameterized agents in terms of the proportion of dormant neurons and the magnitude of weight changes throughout training. Figure 17 shows that CompleteP agents display a lower proportion of dormant neurons relative to NTK agents, particularly when the network width is sufficiently large ($N > 16$). Both parameterizations exhibit reduced dormancy as model width increases; however, in NTK, this arises because feature learning diminishes rapidly as width grows, effectively "freezing" representations and thus reducing exposure to the dormant neuron problem. By contrast, CompleteP agents preserve active, continually evolving features at $O(1)$ scale even for large widths. Refer to Appendix K for the rate of feature evolution for both NTK and CompleteP agents.

Weight change dynamics, depicted in Figure 18, further highlight differences. In NTK agents, the average magnitude of weight updates declines steadily during training and across width, indicating a sharp loss of plasticity with scale. For CompleteP, weight change magnitude decreases initially with width but plateaus for wide models, reflecting stable, consistent plasticity throughout learning. Larger CompleteP networks are thus better equipped to maintain adaptability, a property especially useful for continual learning and challenging RL environments (Graldi et al., 2025).

Together, these results suggest an underexplored benefit of wide, richly-parameterized RL agents: they may avoid catastrophic loss of plasticity, enabling both continual adaptation and retention of prior knowledge. This capacity could be crucial for scaling embodied control and lifelong learning systems far beyond current empirical limits.

## K. Coordinate Check for Feature and Logit Evolution

### K.1. Width Scaling Feature Coordinate Checks

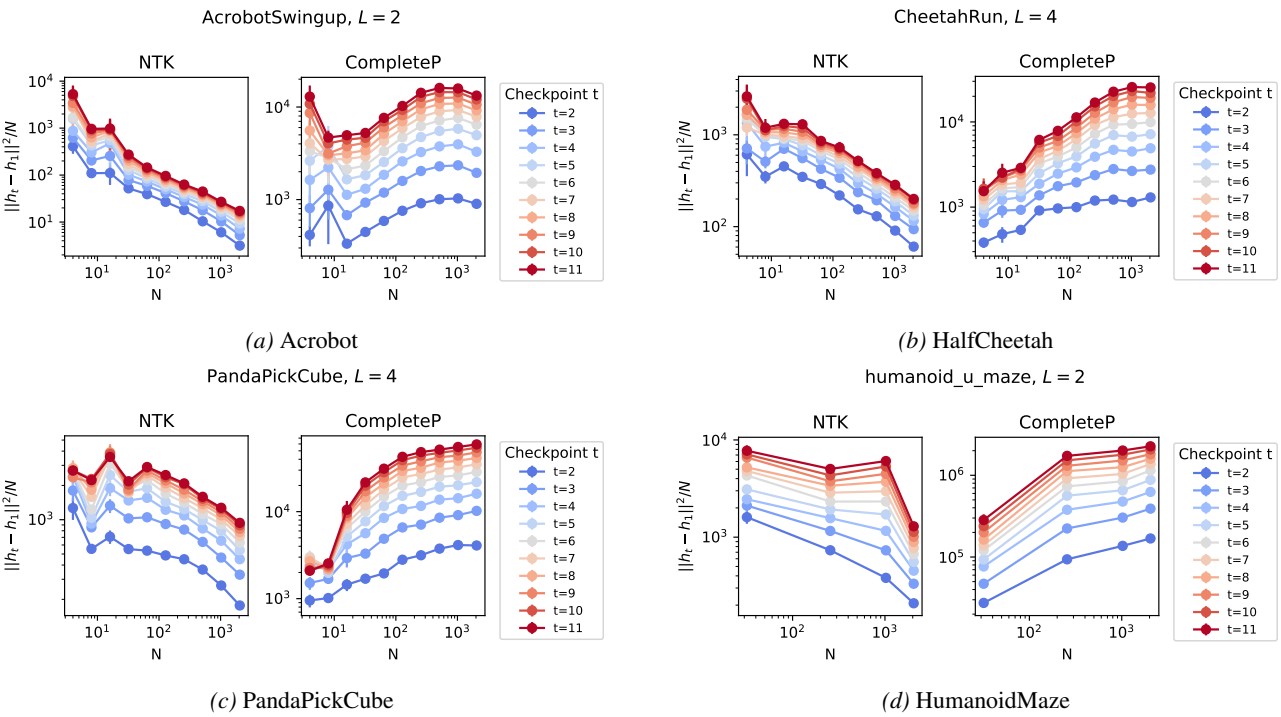

*Figure 19.* **Feature evolution.** NTK shows decreasing feature evolution with width scaling, confirming that feature learning diminishes in wide networks. CompleteP shows increasing feature evolution that plateaus for $N > 512$, supporting robust, width-invariant feature learning in large networks.

To empirically check that our NTK and CompleteP agents achieve the theoretical desiderata for network parameterization, especially concerning their feature and output function evolution across width scaling, we follow the methodology outlined in (Yang et al., 2022; Dey et al., 2026). Feature evolution is quantified as the L2 norm $\|h_t - h_1\|$ with $h$ measured on the synthetic dataset, where $t$ denotes the training checkpoint and $h_1$ the feature vector after the first model checkpoint. This metric tracks how much the network's internal representations deviate from those at the start of training, and thus captures the degree of feature learning achieved (our desideratum 3).

As seen in Figure 19, NTK agents display a clear trend: feature evolution *decreases* as width increases, consistent with theory that NTK parameterization causes feature learning to diminish in wide networks. For CompleteP agents, we observe that feature evolution *increases* with width up to $N \approx 512$, after which it plateaus—suggesting $\mathcal{O}(1)$ feature change in the large width regime. This agrees with our desiderata for rich, width-invariant feature learning, but with one caveat: for smaller widths, the observed feature evolution is less consistent, possibly due to low reward signals associated with smaller network capacity agents that suppress effective gradient updates. As width grows, agents have the capacity to learn to maximize rewards and maintain evolving features, as predicted by theory and recent empirical work (Dey et al., 2026).

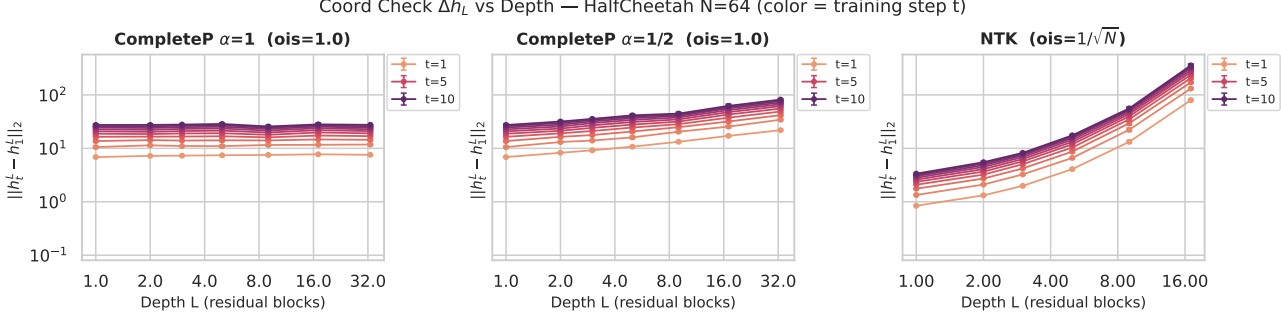

*Figure 20.* **Depth scaling and feature evolution.** NTK exhibits increasing $\Delta h_L$ with depth, reflecting accumulated residual drift and limited feature evolution. CompleteP with $\alpha = 1$ maintains approximately depth-invariant $\Delta h_L$, while $\alpha = 0.5$ shows partial stabilization with a mild upward trend.

## K.2. Depth Scaling Feature Coordinate Check

To complement the width-scaling analysis, we evaluate how NTK and CompleteP behave under *depth scaling*. Following the same coordinate-check methodology used in prior work (Yang et al., 2022; Dey et al., 2026), we compute the L2 deviation $\Delta h_L(t) = \|h_t^{(L)} - h_t^{(1)}\|_2$ where $h_t^{(L)}$ denotes the feature vector at depth $L$ and training step $t$, and $h_t^{(1)}$ is the corresponding representation at depth 1. This diagnostic measures how sensitively intermediate representations drift as residual blocks are stacked, and thus captures the stability of feature evolution across depth.

As shown in Figure 20, NTK agents display a clear increase in $\Delta h_L$ as depth grows, consistent with theory that NTK suppresses feature learning and accumulates residual drift in deep networks. CompleteP with $\alpha = 1$ maintains an approximately depth-invariant $\Delta h_L$ across training steps, matching the stable-depth behavior predicted by theory and observed in recent empirical studies (Dey et al., 2026). For $\alpha = 0.5$, we observe intermediate behavior: feature drift is substantially reduced relative to NTK but still exhibits a mild upward trend with depth, indicating partial—but not complete—normalization of depth scaling. This aligns with observations in (Mlodozeniec et al., 2025) that $\alpha < 1$ may not fully correct depth-dependent amplification.

Overall, these results support the conclusion that CompleteP with $\alpha = 1$ provides the most robust depth-invariant feature evolution, while $\alpha = 0.5$ offers partial stabilization. We include these results to address concerns about the sensitivity of our findings to the depth-scaling coefficient.

## K.3. Logit Coordinate Checks

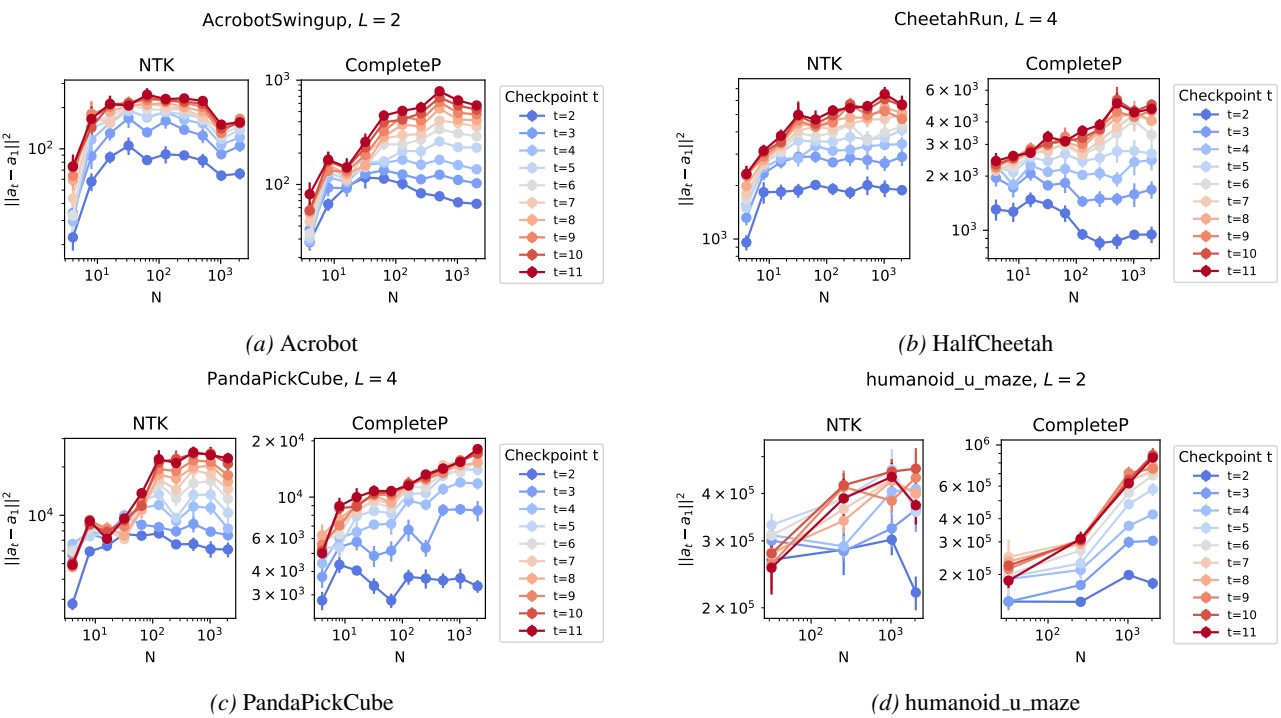

*(a)* Acrobot

*(b)* HalfCheetah

*(c)* PandaPickCube

*(d)* humanoid_u_maze

*Figure 21.* **Logit evolution.** Both NTK and CompleteP agents achieve stable, width-invariant logit evolution for $N > 512$. Smaller widths show greater variability, likely due to reward-dependent gradient suppression.

Logit (function) evolution is measured as $\|a_t - a_1\|$, where $a_t$ is the action (logit) output at checkpoint $t$, again evaluated on the synthetic dataset. Both NTK and CompleteP agents demonstrate *stable logit evolution* for sufficiently large width: as shown in Figure 21, function learning becomes width-invariant for $N > 512$, even as smaller widths display more variability.

Overall, we confirm that both NTK and CompleteP agents achieve desiderata 2 (stable function learning) and 3 (controllable feature learning) for sufficiently large widths. The instability in feature and function evolution at small widths likely arises from the reward-dependent nature of policy gradients, where low rewards limit effective weight (and thus feature) updates.

# L. Feature Consistency with Width Scaling

CompleteP parameterization robustly improves learning performance and plasticity at scale. In this section, we analyze how feature learning and representational consistency evolve with network width, focusing on both the structure and alignment of learned feature kernels.

Figure 22 visualizes the feature kernels (the similarity matrix $HH^\top$ of penultimate-layer activations) at initialization and after training under NTK and CompleteP parameterizations, for varying network widths in AcrobotSwingup, CheetahRun, and PandaPickCube. At initialization, kernels are highly similar and preactivation densities $\rho(h^L)$ are consistent across widths, verifying stable signal propagation (desideratum 1). After training, NTK kernels increasingly resemble their initial state and grow more diffuse, while CompleteP retains consistent kernel structure and sharper, low-dimensional features for larger widths ($N > 256$).

Figure 23 shows the cumulative variance explained by the top five eigenvectors of the feature kernel, highlighting feature dimensionality. As width increases, initial feature spectra decays rapidly, indicating increasingly high-dimensional, disordered features at scale. NTK shows significant spectra decay in simpler control tasks i.e. Acrobot but the decay slows down as the task becomes progressively harder. In contrast, CompleteP exhibits stable, strongly concentrated spectra, indicating the emergence of low-dimensional, task-relevant representations even as width grows.

Figure 25 examines feature kernel alignment across seeds using CKA similarity. Kernel alignment between initial and NTK increases rapidly with width, suggesting that NTK kernels become more similar to random kernels as width grows. In comparison, CompleteP alignment increases more slowly, reflecting the preservation of task-relevant diversity and less convergence to randomness.

Figure 26 shows direct visualizations of feature kernel variability across seeds. Both NTK and muP show inconsistent features and action logits with different seeds, but CompleteP exhibits greater CKA similarity, supporting improved robustness and representation consistency, especially at width $N$ 256.

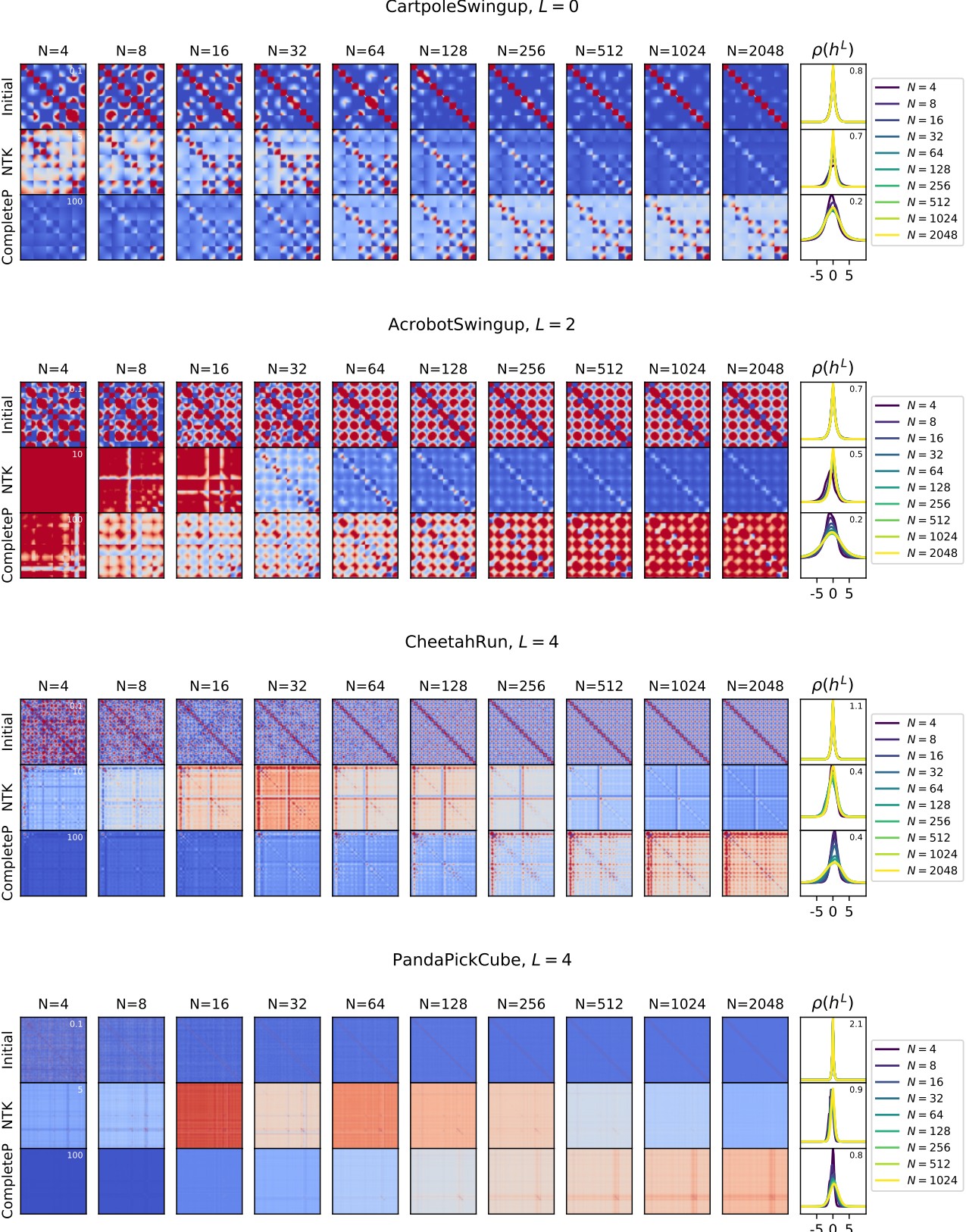

*Figure 22.* **Consistent features across width.** At initialization, both NTK and CompleteP show similar feature kernels and preactivation densities. After training, NTK feature kernels become increasingly diffuse and similar to the initial state, indicating reduced feature learning at large widths. CompleteP retains consistent and structured kernels with sufficiently large widths ($N > 256$), demonstrating $\Theta(1)$ feature learning.

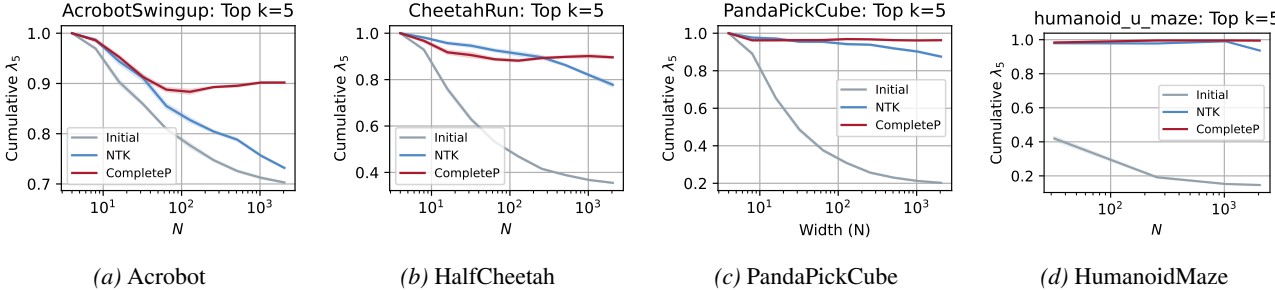

*Figure 23.* **CompleteP learns low-dimensional features with width scaling.** For random initializations and NTK features, the cumulative variance explained by the top 5 eigenvalues decays with width, indicating increasingly diffuse features. However, NTK's spectral decay seems to be task dependent, where the decay is slower for harder tasks. By contrast, CompleteP preserves concentrated, low-dimensional structure across widths, supporting robust representation learning.

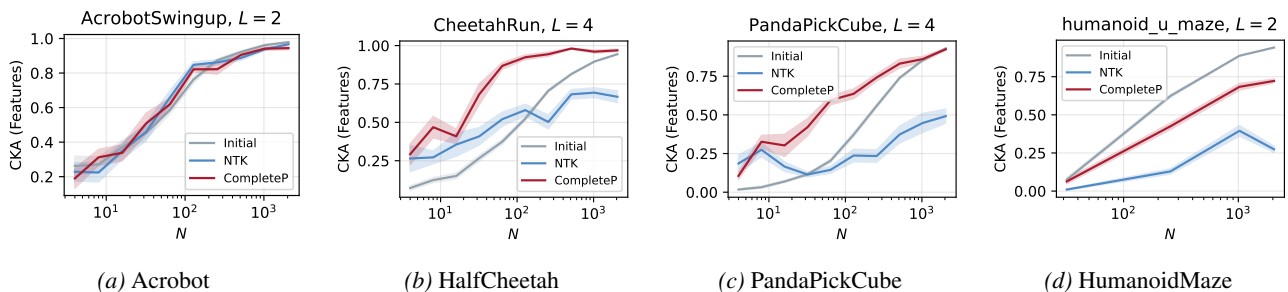

*Figure 24.* **Seed specific kernel alignment increases more rapidly for CompleteP.** CKA similarity for CompleteP between different seed initializations rises quickly with width, indicating CompleteP kernels converge toward similar structured states.

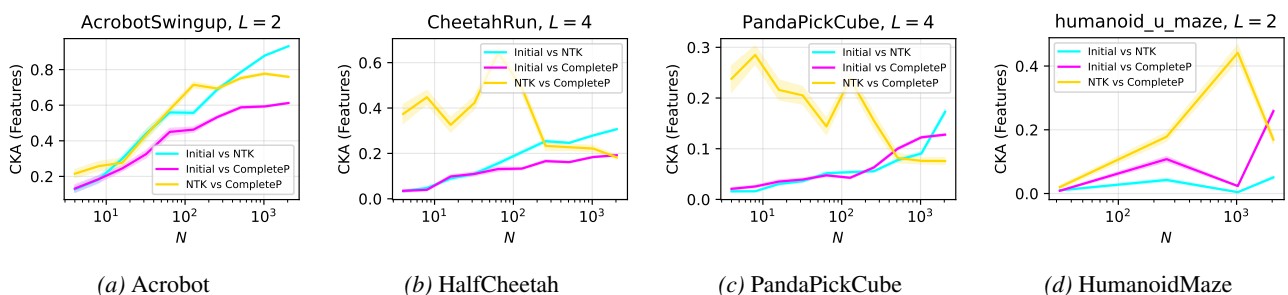

*Figure 25.* **Kernel alignment increases more rapidly for NTK with initial features.** CKA similarity between initial and NTK kernels rises quickly with width, indicating NTK kernels converge toward the structureless random initial state. For CompleteP, alignment rises more slowly, indicating greater diversity and stability of learned features across seeds and widths. However, this rate of increase seems to be task dependent.

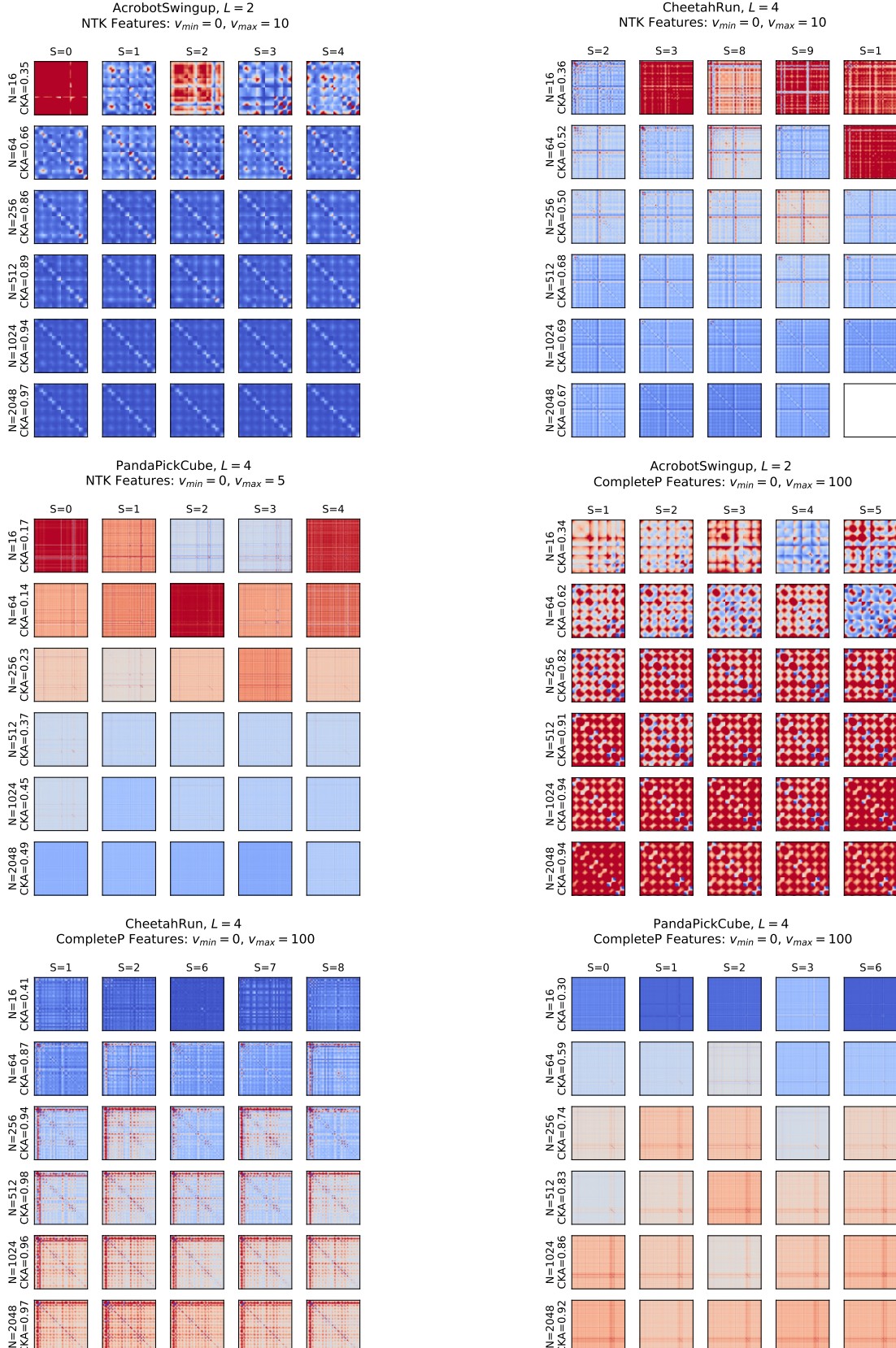

*Figure 26.* **Feature consistency improves across seeds with width scaling.** Both NTK and muP show inconsistent action logits and features across seeds, but muP exhibits slightly higher kernel alignment, indicating enhanced representation robustness across initializations.

# M. Policy Consistency with Width Scaling

Network parameterization also affects the consistency and reproducibility of learned policies. Here, we analyze how the action logits and policy kernels evolve with width scaling and seed variation.

## M.1. Consistency with Width Scaling

Figure 27 shows the comparison of action logits between models of different widths for NTK and CompleteP agents across multiple environments. For each setting, the largest width ($N = 2048$) is used as a reference; logit mean squared error is measured between each smaller width model and the largest over the course of learning.

Both NTK and CompleteP agents demonstrate increasing logit alignment across widths as models get wider, indicating consistency in policy learning, especially in the high-capacity regime. Averaged logit kernels are consistent across widths for both parameterizations after learning, which is expected since the NTK parameterization places no restriction on output evolution (see desideratum 2). In smaller networks, divergence is higher, aligning with their reduced performance. CompleteP models not only match NTK but often show even faster width-dependent convergence of logits, likely a result of their more robust feature learning.

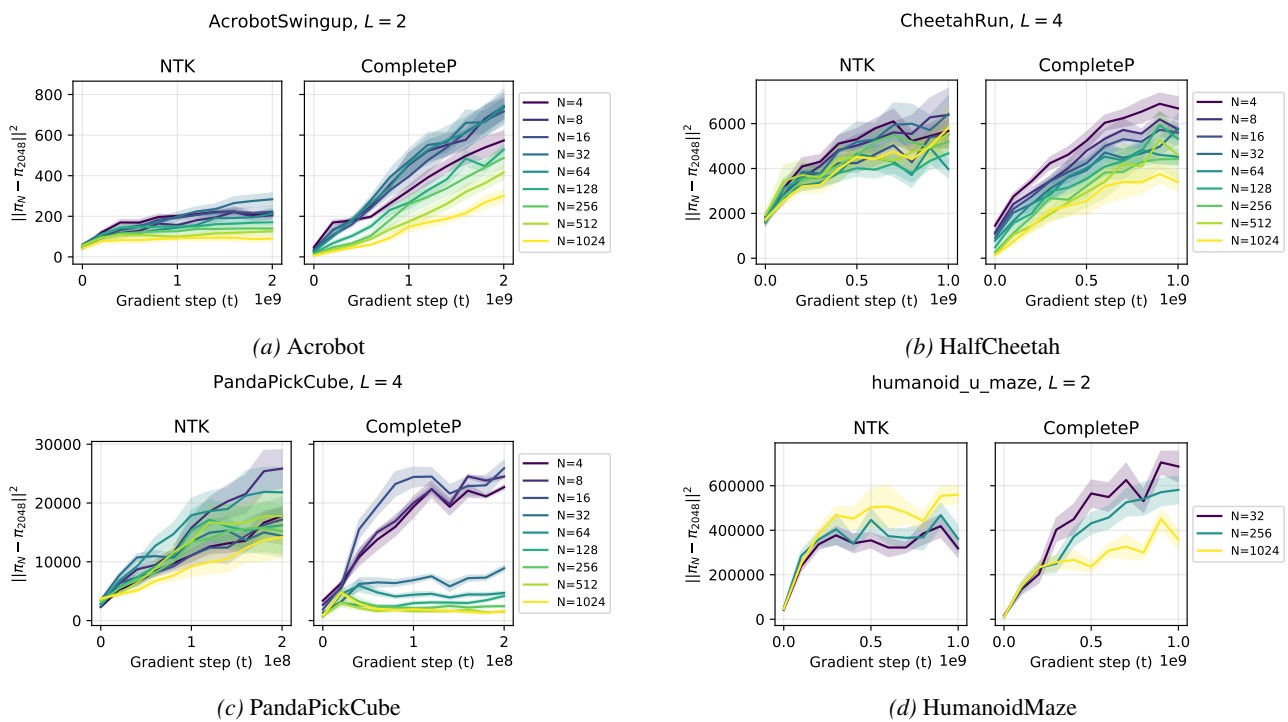

*(a)* Acrobot

*(b)* HalfCheetah

*(c)* PandaPickCube

*(d)* HumanoidMaze

*Figure 27.* **Consistent logit evolution across width.** NTK and CompleteP agents both show increasing policy consistency across width, with logits converging toward those of the largest agent. Policy outputs for narrow networks are more variable due to limited capacity, while wide networks converge toward similar control strategies.

## M.2. Inconsistency Across Seeds

Despite improved consistency with width scaling, stochasticity inherent to RL (e.g., exploration noise, environment transitions) can introduce variability in learned policies across seeds. **Figure 27** examines the kernel structure of action logits for NTK and muP agents trained with different random seeds.

While both parameterizations show some variation across seeds, CompleteP demonstrates notably higher CKA (centered kernel alignment) similarity compared to NTK, suggesting improved reproducibility of learned policies even in the presence of RL stochasticity. In contrast, NTK policies are more susceptible to seed-dependent variations, consistent with reduced feature plasticity at high width.

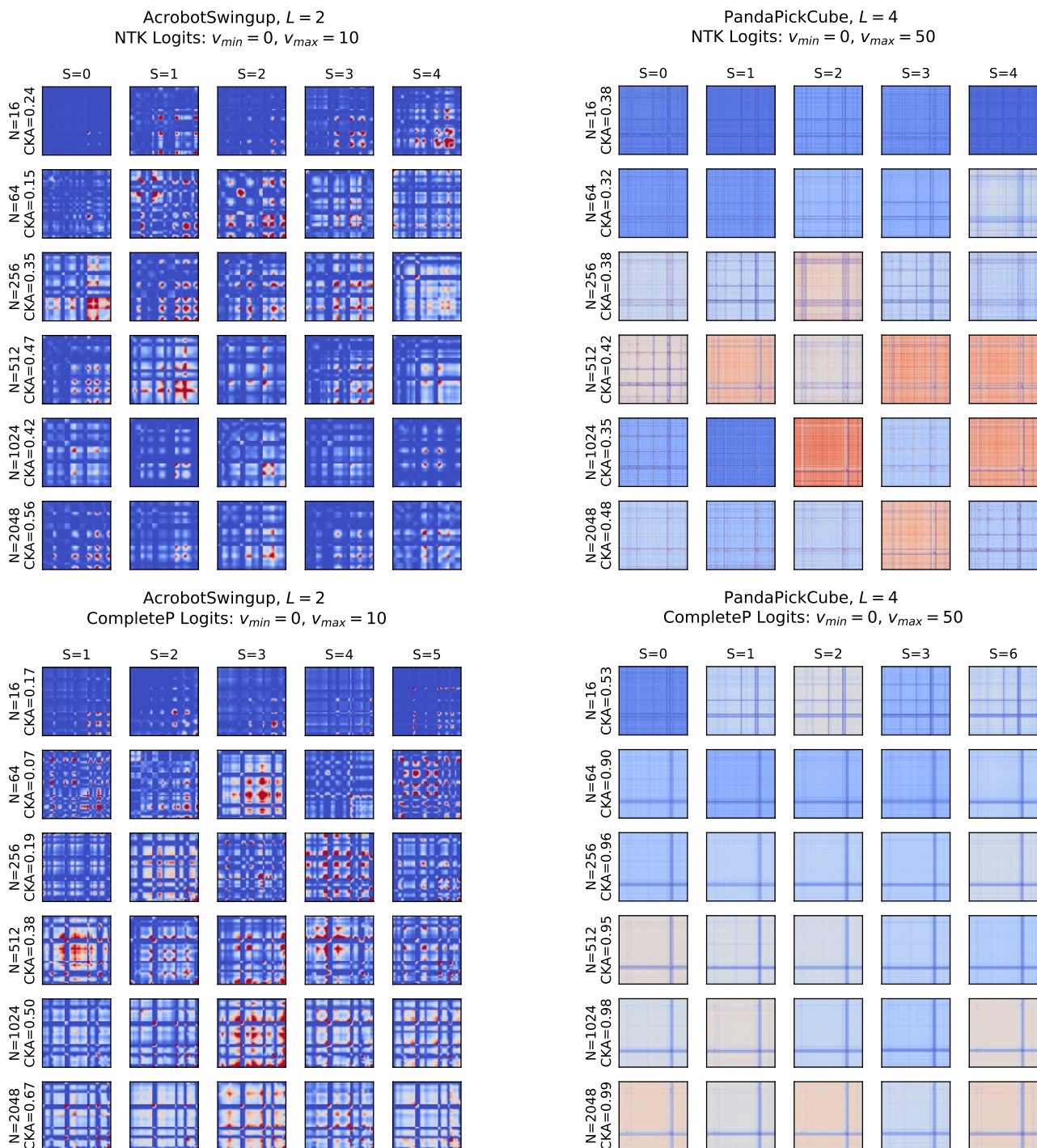

*Figure 28.* **CompleteP achieves higher logit consistency acros seeds with width scaling.** Both NTK and muP agents show some inconsistency in action logits across seeds, but muP exhibits notably higher kernel alignment (CKA), supporting improved reproducibility of the learned policy as width increases.

# N. Features and Policies Converge at Different Rates

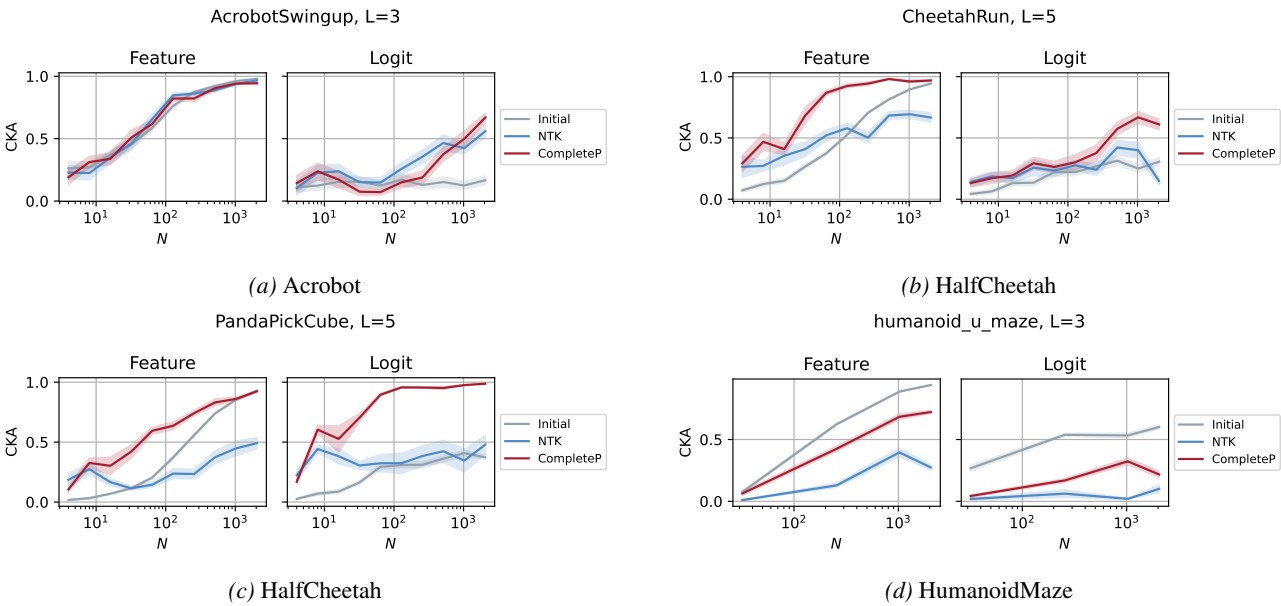

*(a)* Acrobot

*(b)* HalfCheetah

*(c)* HalfCheetah

*(d)* HumanoidMaze

*Figure 29.* **Logit kernels for individual seeds converge slower than feature kernels with width scaling suggesting possible divergence between feature and policy learning.**

# O. Additional Learning Curves

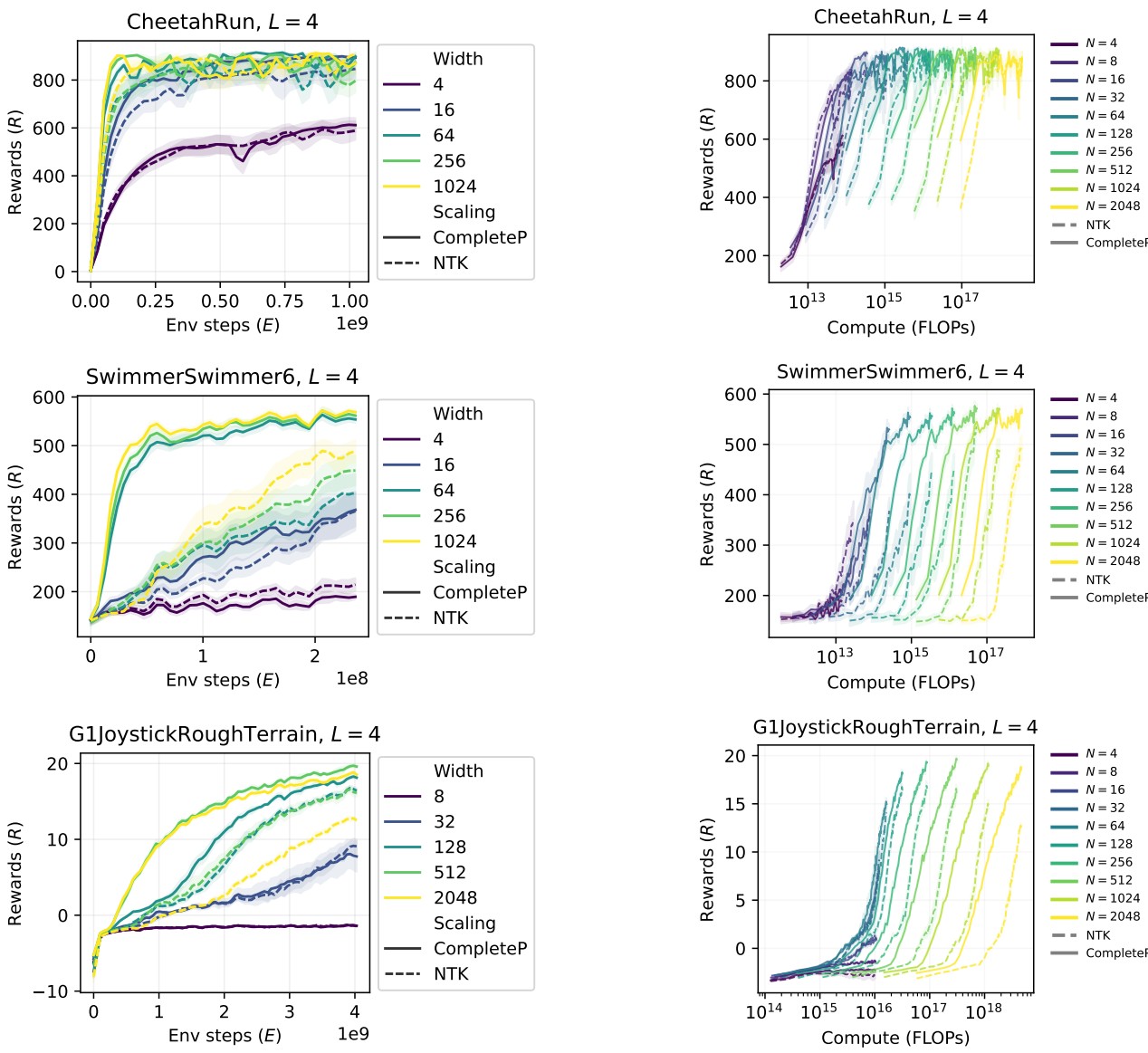

*Figure 30.* **Summary of reward learning curves, compute usage, and Pareto frontiers for three hard control tasks.**

## O.1. Depth Scaling

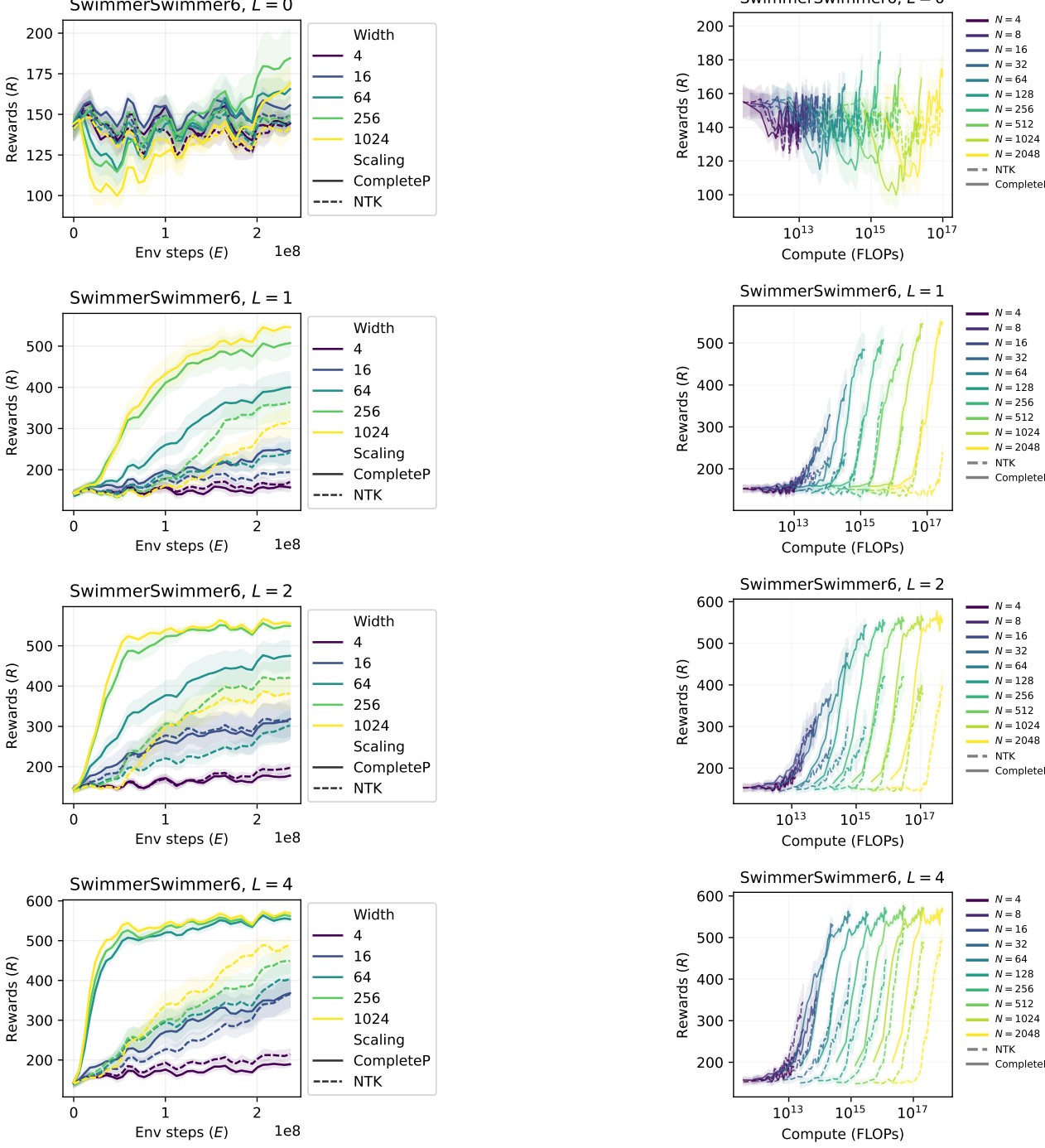

*Figure 31.* **Depth scaling improves learning performance in both CompleteP and NTK agents on Swimmer environment.**

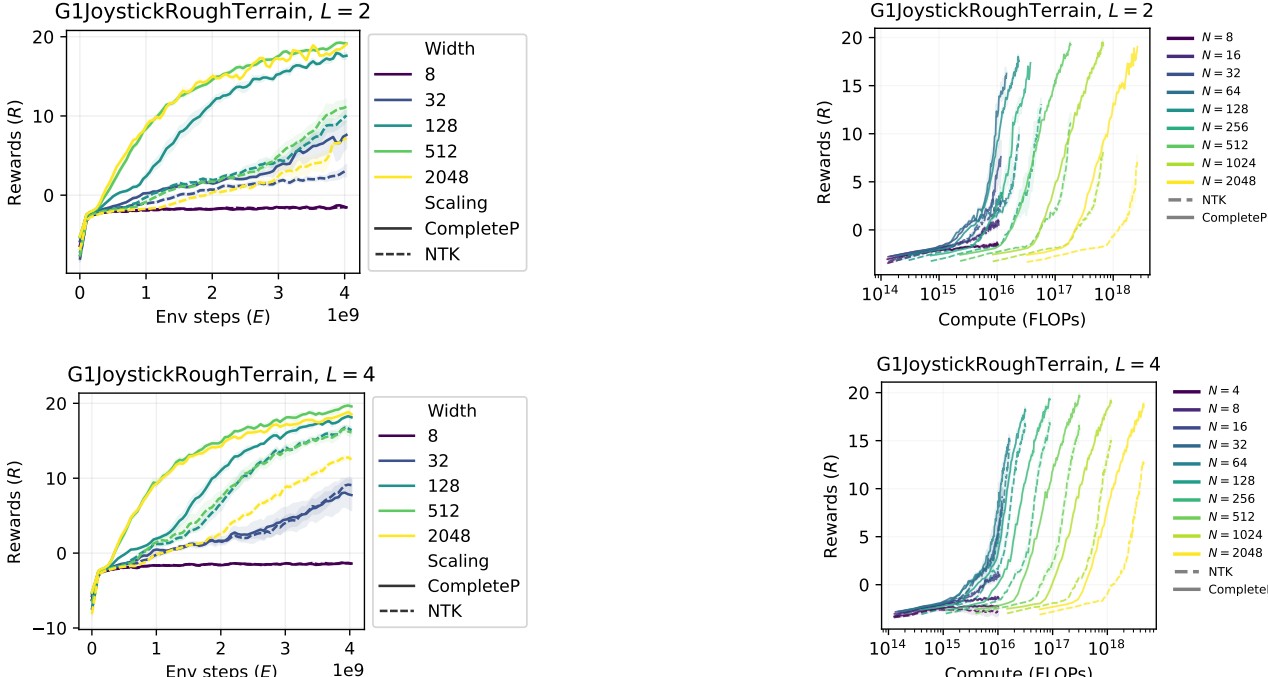

*Figure 32.* **Depth scaling improves learning performance in both CompleteP and NTK agents on G1Rough environment.**

## O.2. Orthogonal Weight Initialization

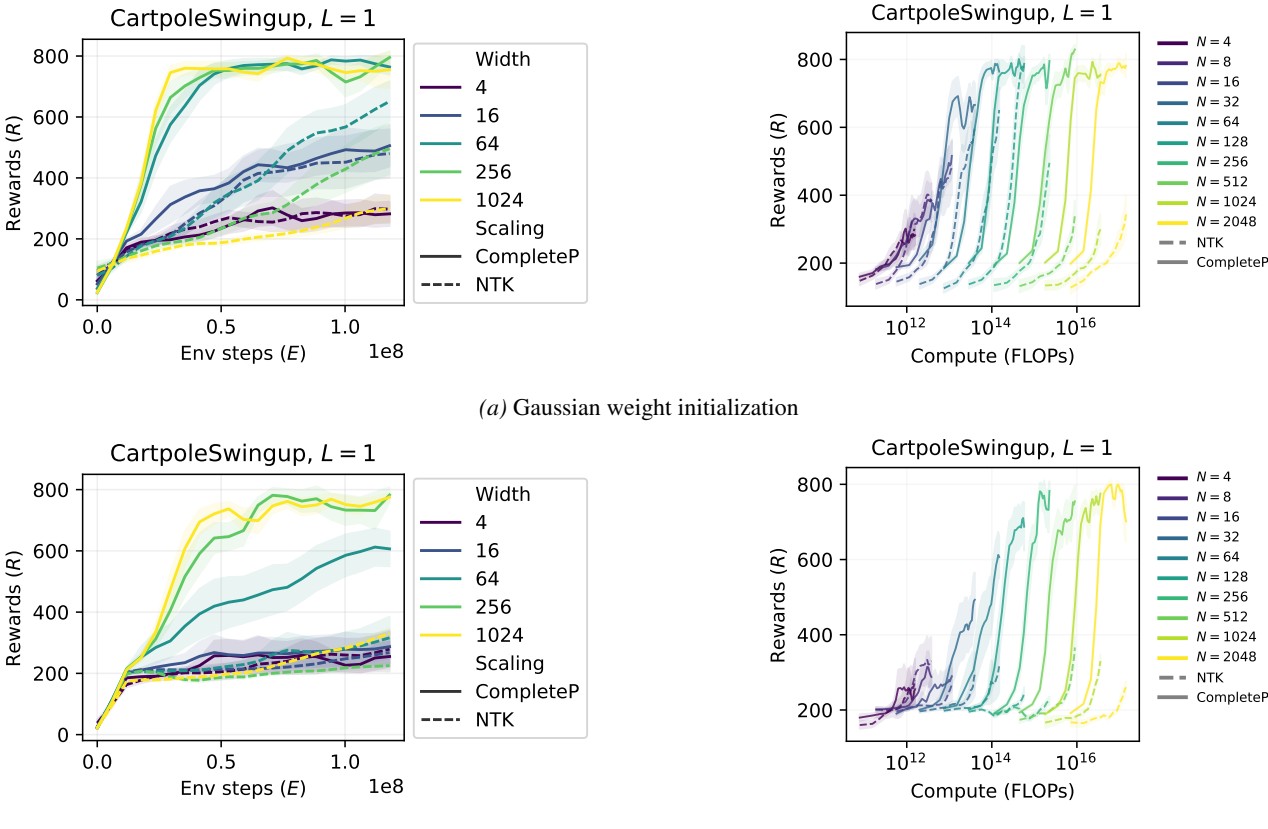

*(a)* Gaussian weight initialization

*(b)* Orthogonal weight initialization

*Figure 33.* **Cartpole and orthogonal initialization.** Each row shows reward curves, compute, and Pareto frontier for (top) Gaussian and (bottom) Orthogonal initialization.

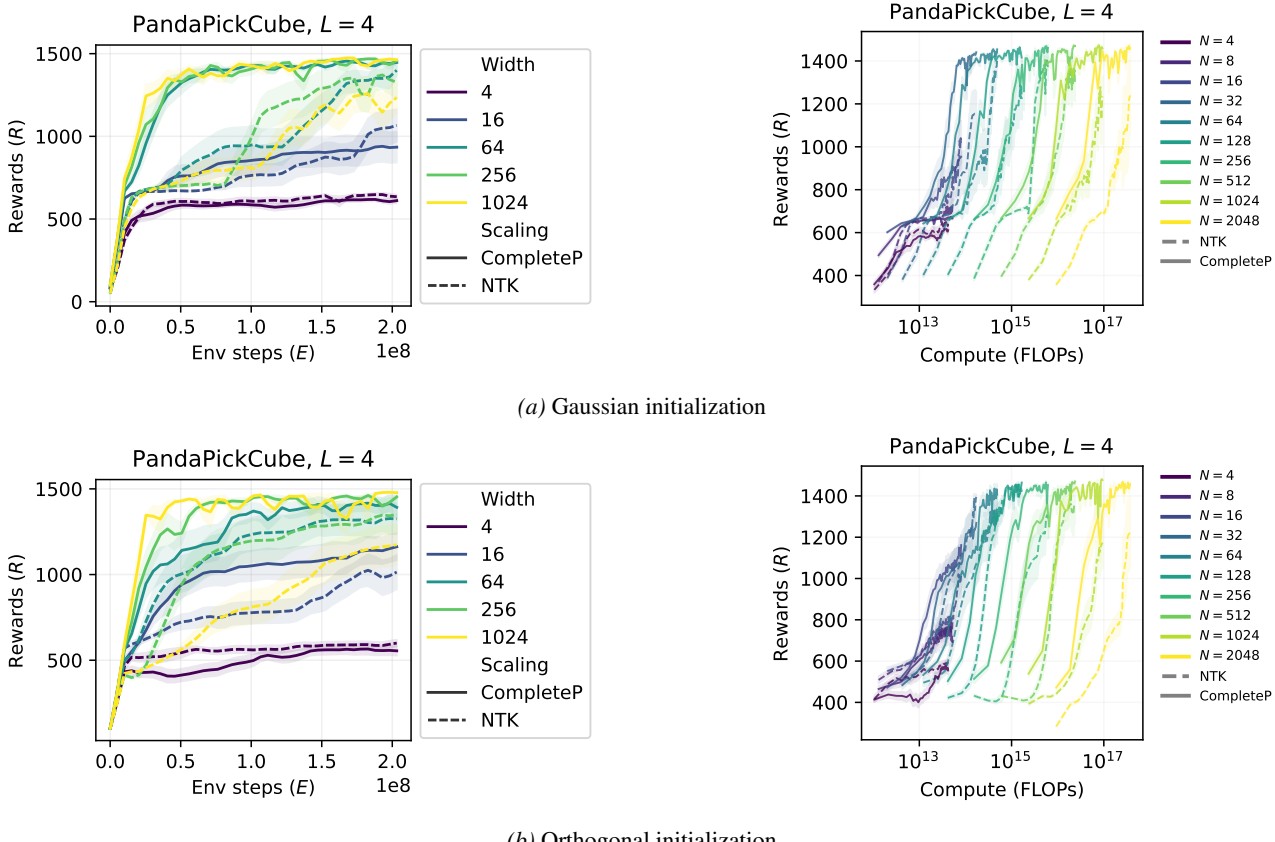

*(a)* Gaussian initialization

*(b)* Orthogonal initialization

*Figure 34.* **PandaPickCube and orthogonal initialization.** Each row: reward-vs-steps, reward-vs-FLOPs, Pareto frontier for (top) Gaussian and (bottom) Orthogonal weight initialization.

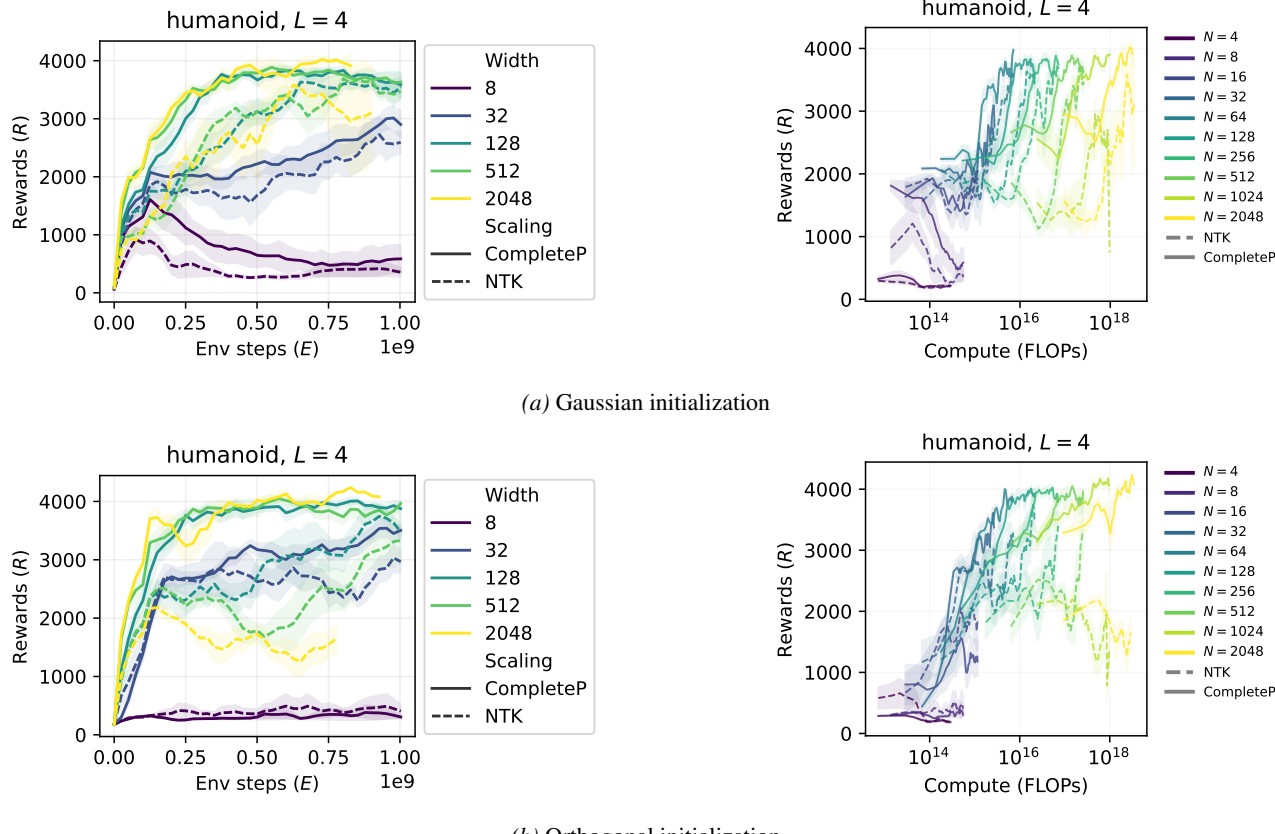

*(a)* Gaussian initialization

*(b)* Orthogonal initialization

*Figure 35.* **Humanoid and orthogonal initialization.** Each row: reward-vs-steps, reward-vs-FLOPs, Pareto frontier for (top) Gaussian and (bottom) Orthogonal weight initialization.

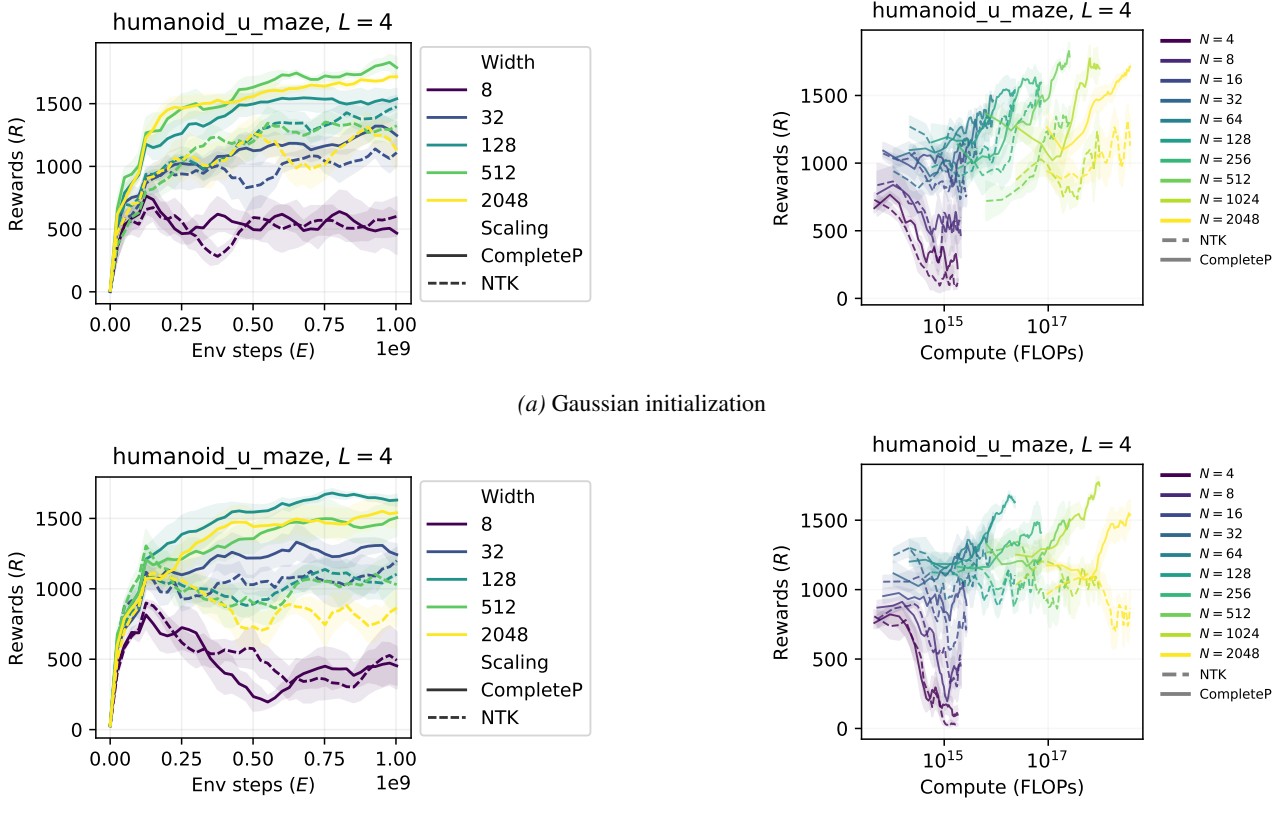

*(a)* Gaussian initialization

*(b)* Orthogonal initialization

*Figure 36.* **HumanoidMaze and Orthogonal initialization.** Each row: reward-vs-steps, reward-vs-FLOPs, Pareto frontier for (top) Gaussian and (bottom) Orthogonal weight initialization.

## O.3. Layer Normalization

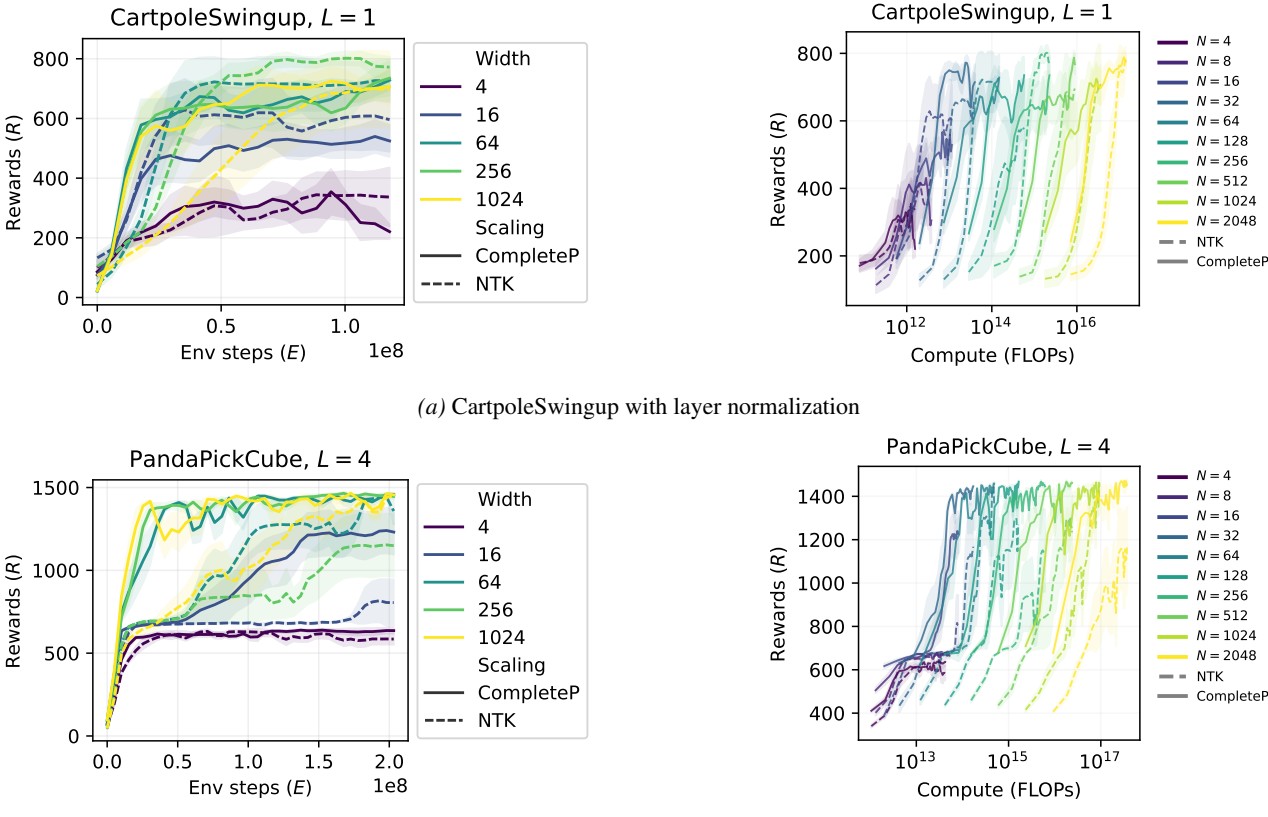

*(a)* CartpoleSwingup with layer normalization

*(b)* PandaPickCube with layer normalization

*Figure 37.* **Adding Layer Normalization into each residual layer.** Each row: reward-vs-steps, reward-vs-FLOPs, Pareto frontier for (top) Cartpole and (bottom) PandaPickCube.

## O.4. Sparse Rewards with Scaling

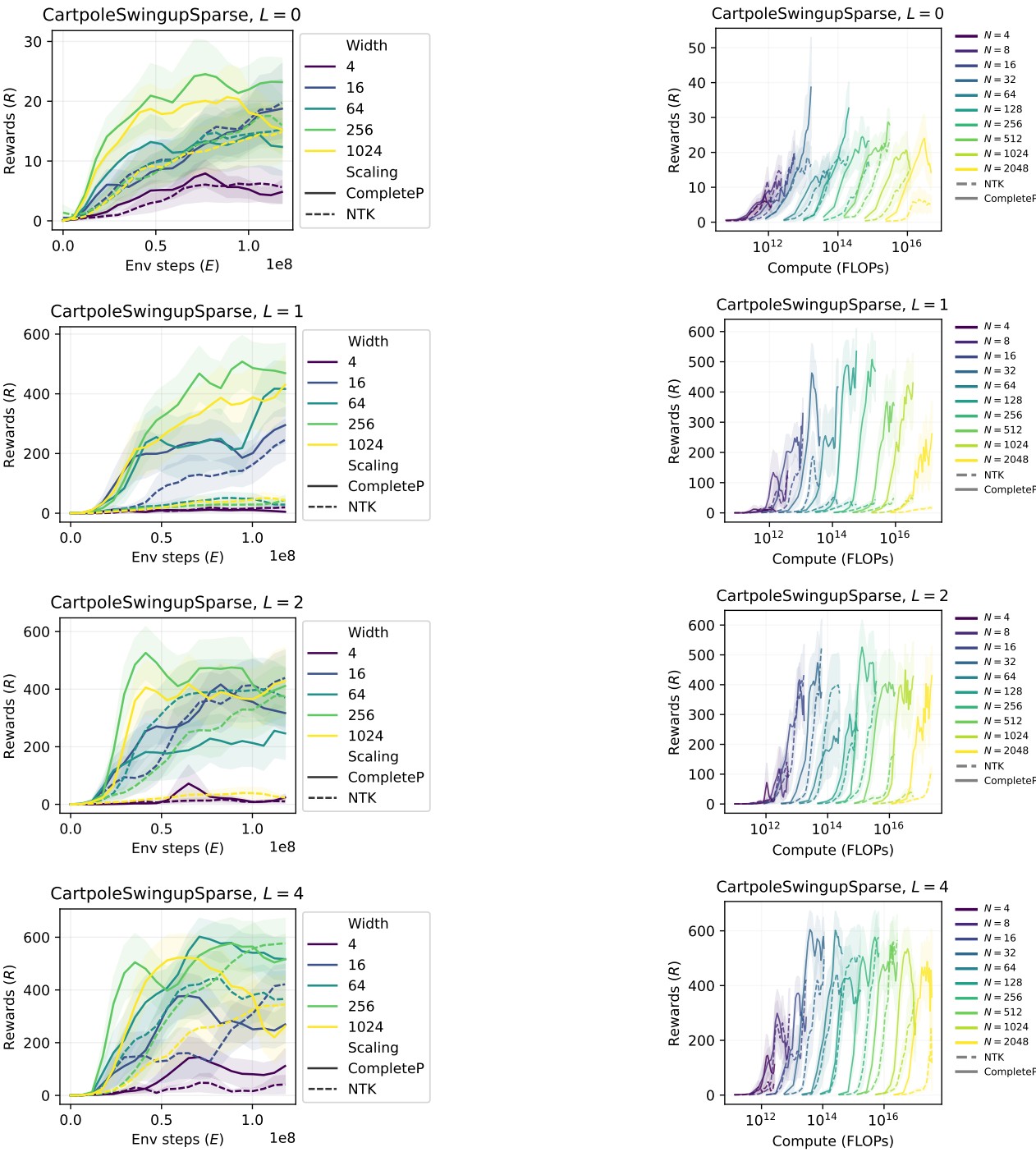

*Figure 38.* **CompleteP demonstrates improved learning with width and depth scaling in CartpoleSwingup with sparse rewards.** Each row indicates models with 0,1,2 or 4 residual layers.

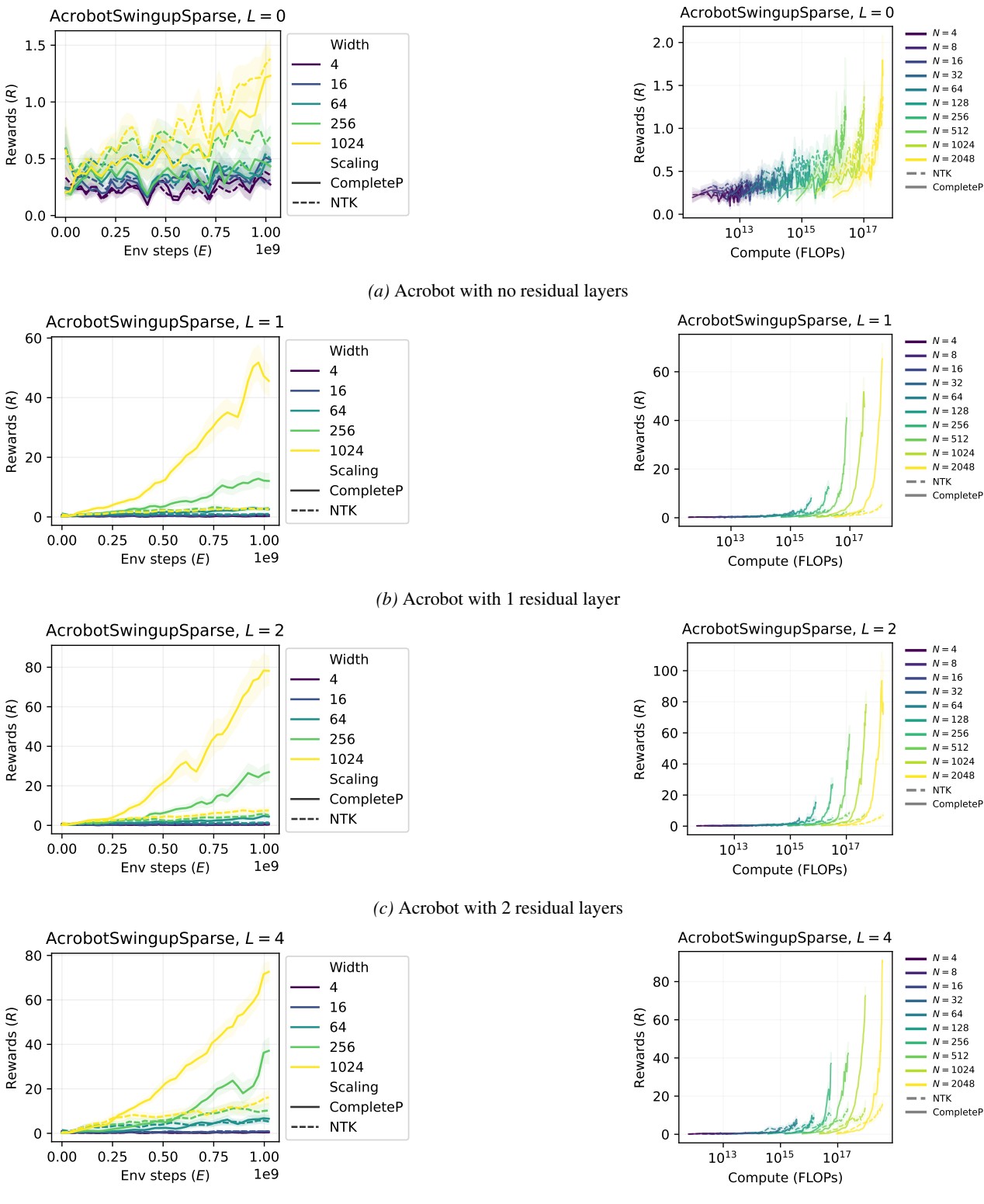

*Figure 39.* **CompleteP demonstrates improved learning with width and depth scaling in AcrobotSwingup with sparse rewards.** Each row indicates models with 0,1,2 or 4 residual layers.

# P. Feature and Logit Dynamics

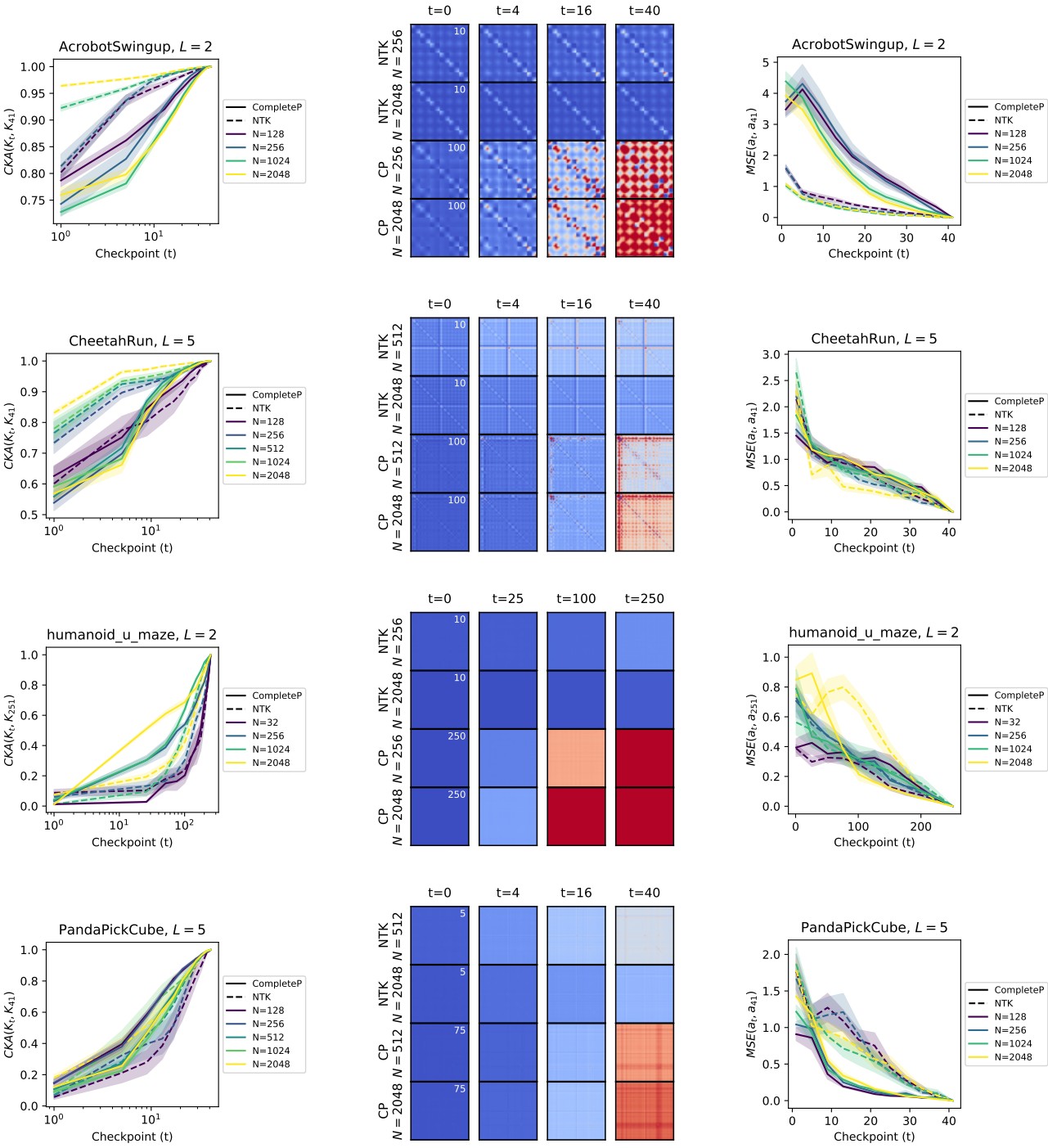

*Figure 40.* Temporal feature and logit evolution dynamics across environments. Each row corresponds to a task (AcrobotSwingup, CheetahRun, humanoid_u_maze, PandaPickCube); columns show feature CKA, kernel similarity, and logit MSE, respectively.

# Q. Width Scaling in Environments with Longer Training Time

We provide more plots comparing the performances of CompleteP and NTK agents across different widths. In some of these plots we train agents around 5-10x times longer than before in order to observe convergence in reward. CompleteP tends to perform better than NTK agents, and does not suffer from reward collapse at larger widths.

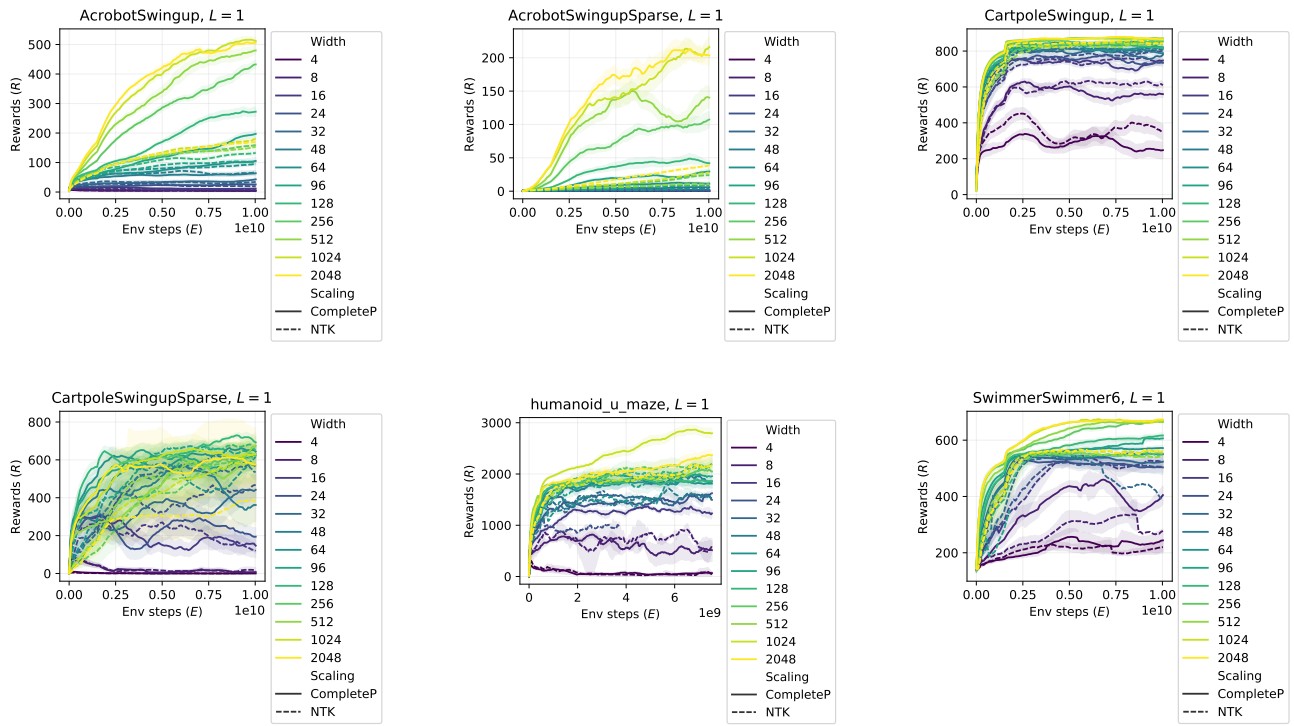

*Figure 41.* **Environment reward vs timesteps for** $L = 1$ **across tasks.**

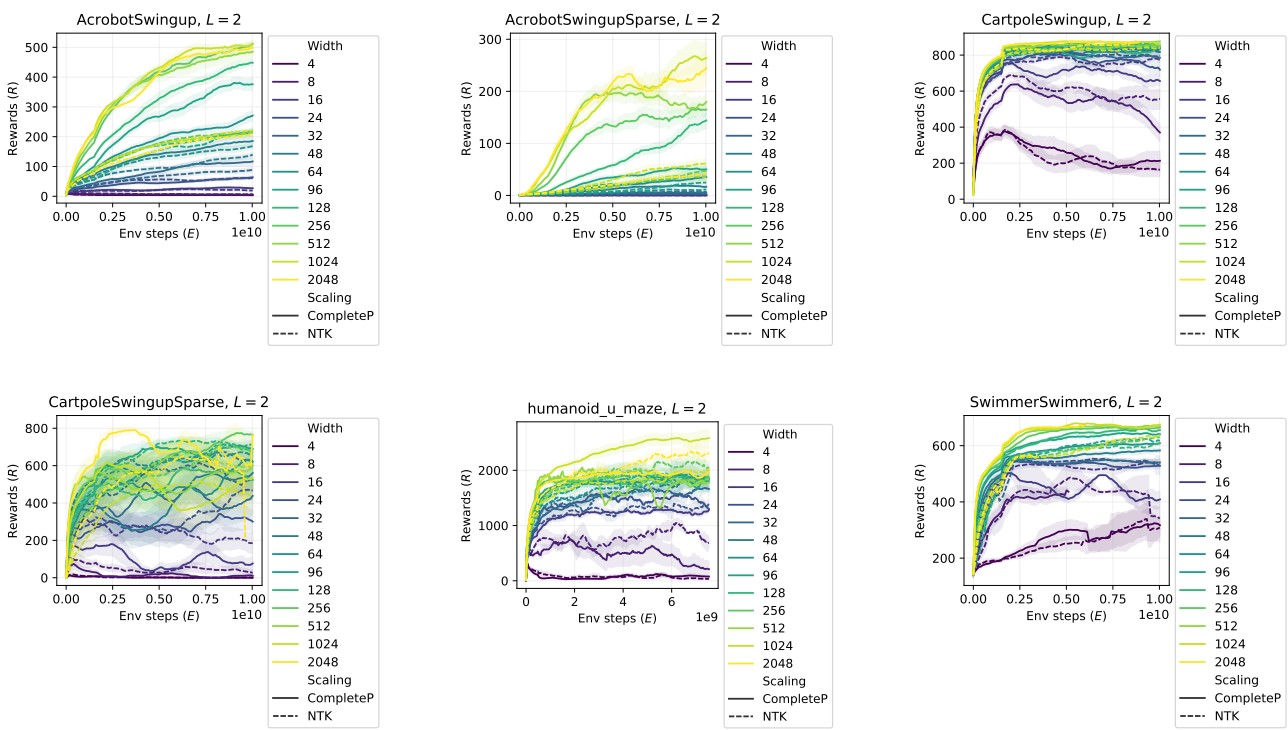

*Figure 42.* **Environment reward vs timesteps for $L = 2$ across tasks.**

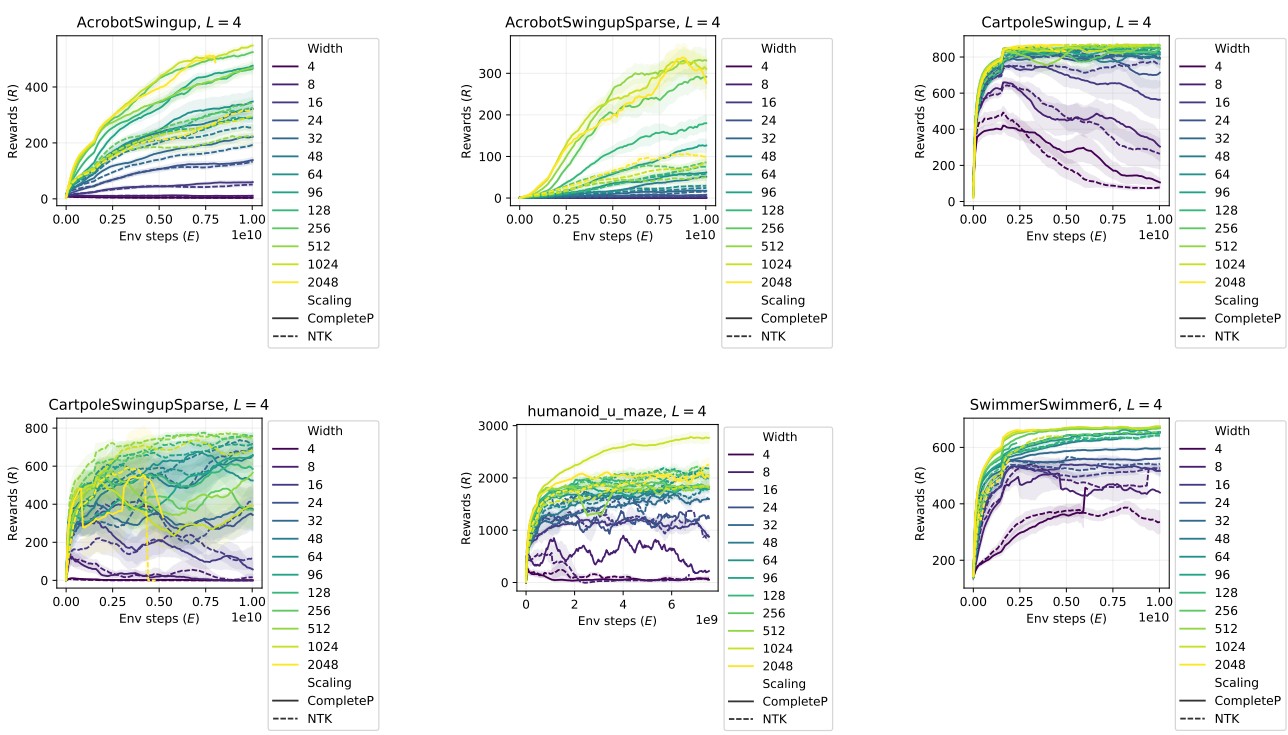

*Figure 43.* **Environment reward vs timesteps for $L = 4$ across tasks.**

# R. FLOP Curves from Environments with Longer Training Time

What follows are plots describing the relationship of compute usage to reward in the same set of environments as Appendix Q. This corroborates our findings from before: CompleteP agents are better or at parity with NTK agents at smaller widths, and are much more efficient than NTK agents at larger widths.

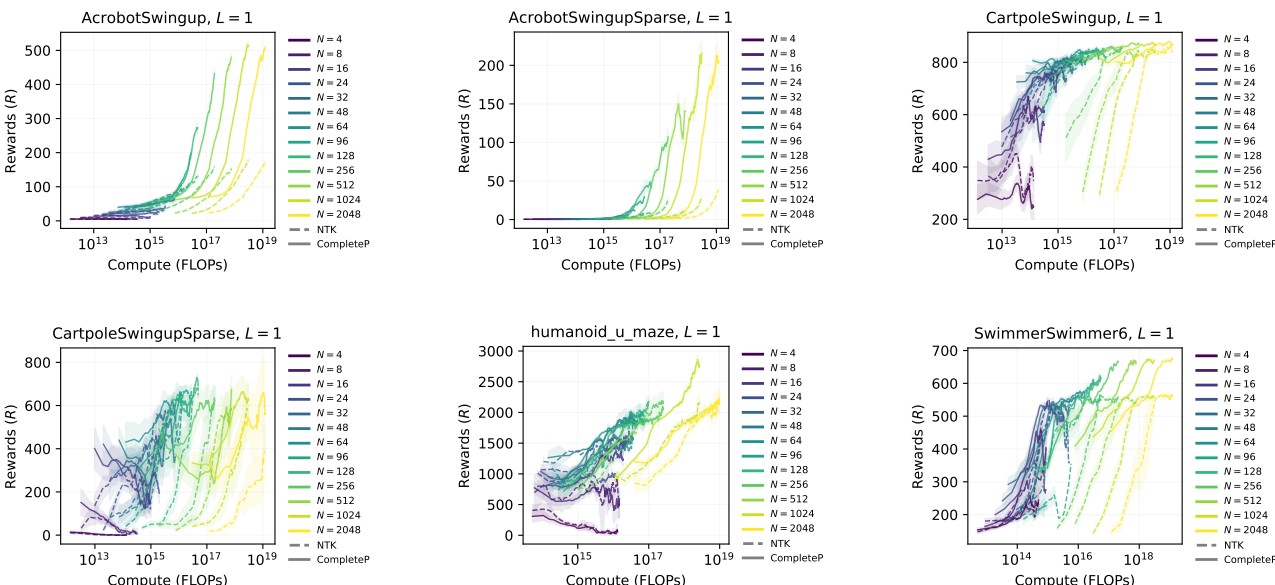

*Figure 44.* **Reward–compute tradeoffs for $L = 1$ as performance v.s. FLOPs.**

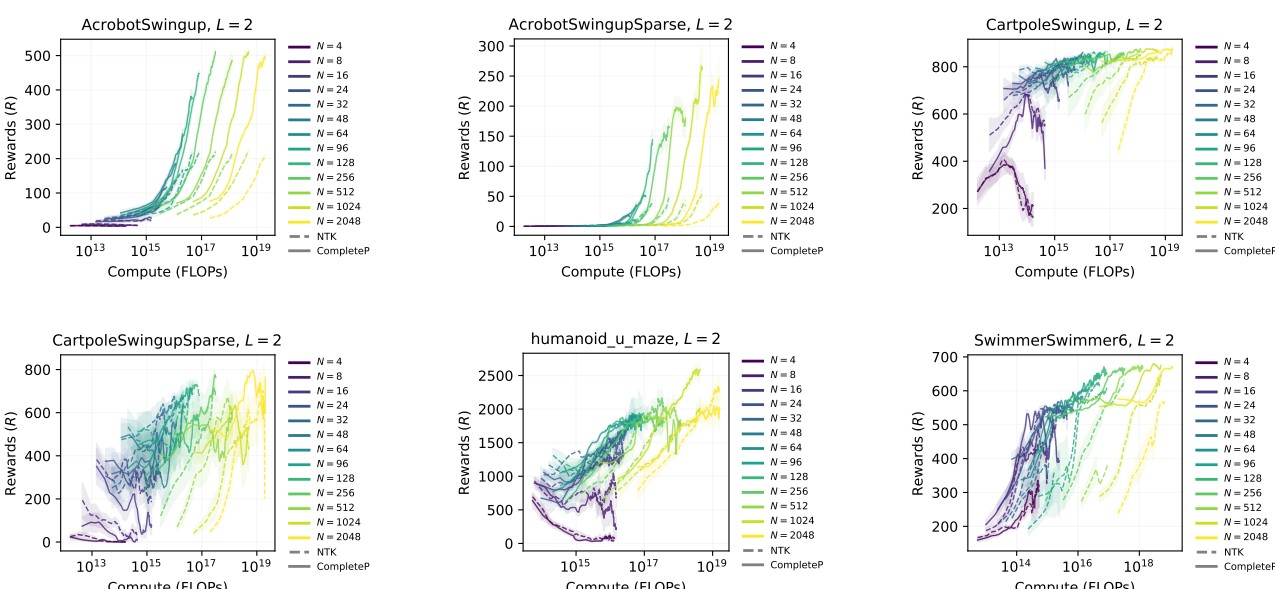

*Figure 45.* **Reward–compute tradeoffs for $L = 2$ as performance v.s. FLOPs.**

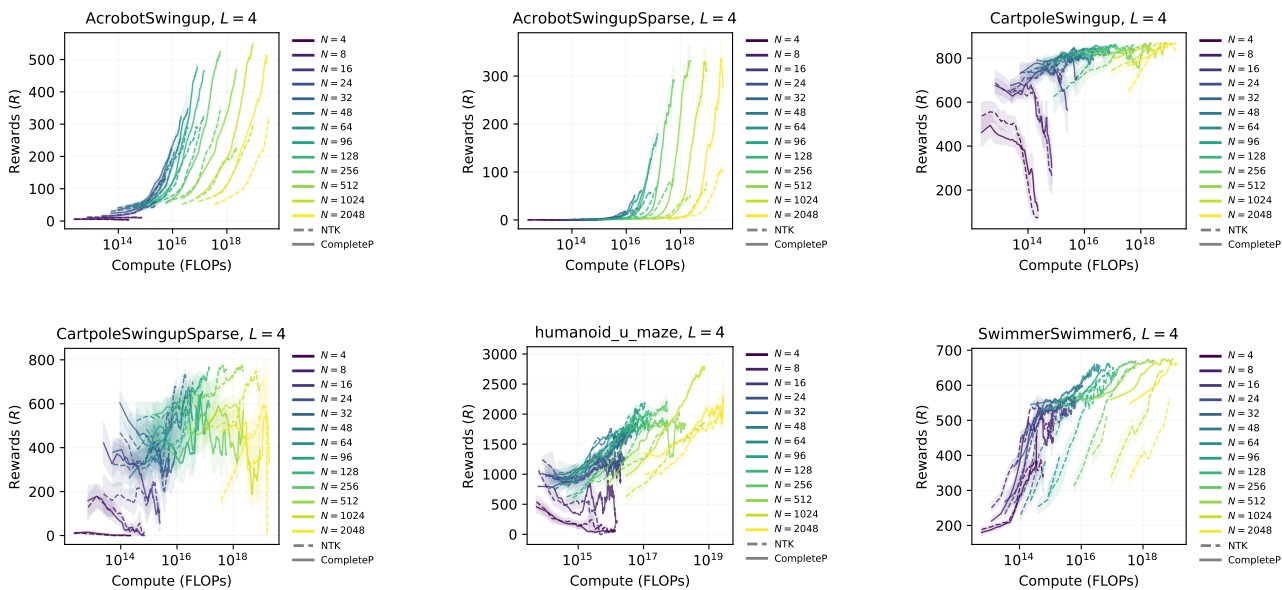

*Figure 46.* **Reward–compute tradeoffs for** $L = 4$ **as performance v.s. FLOPs.**

# S. Pareto Curves from Environments with Longer Training Time

What follows are plots describing isotonic regression fits on the relationship of compute usage to reward from the data in Appendix R in the same set of environments as Appendix Q. We find that CompleteP agents perform better or at parity than NTK agents in terms of their pareto frontier. We note that in more complex environments, where more parameters and hence larger widths are required to perform well, CompleteP agents improve upon the pareto frontier. However, in a simpler task such as Cartpole, both CompleteP and NTK agents share roughly the same pareto frontier.

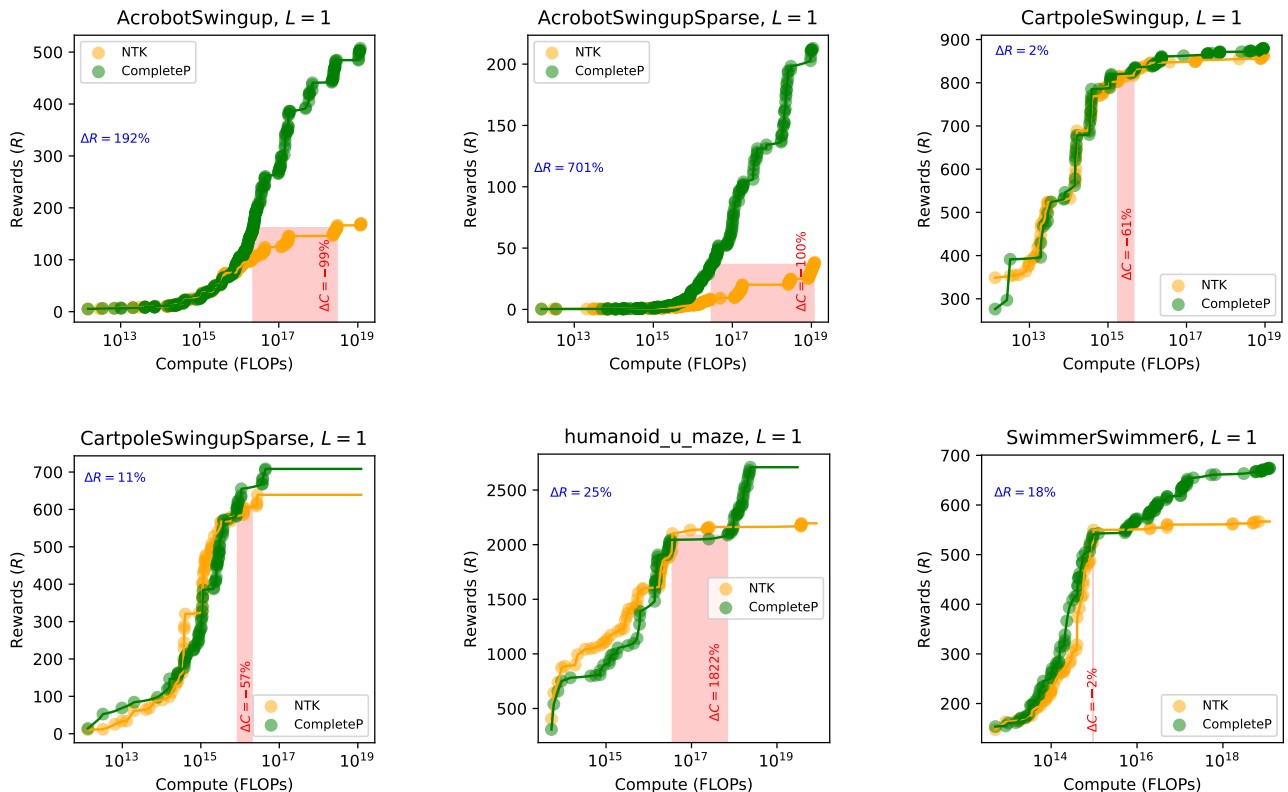

*Figure 47.* **Pareto frontiers for reward vs compute at $L = 1$.**

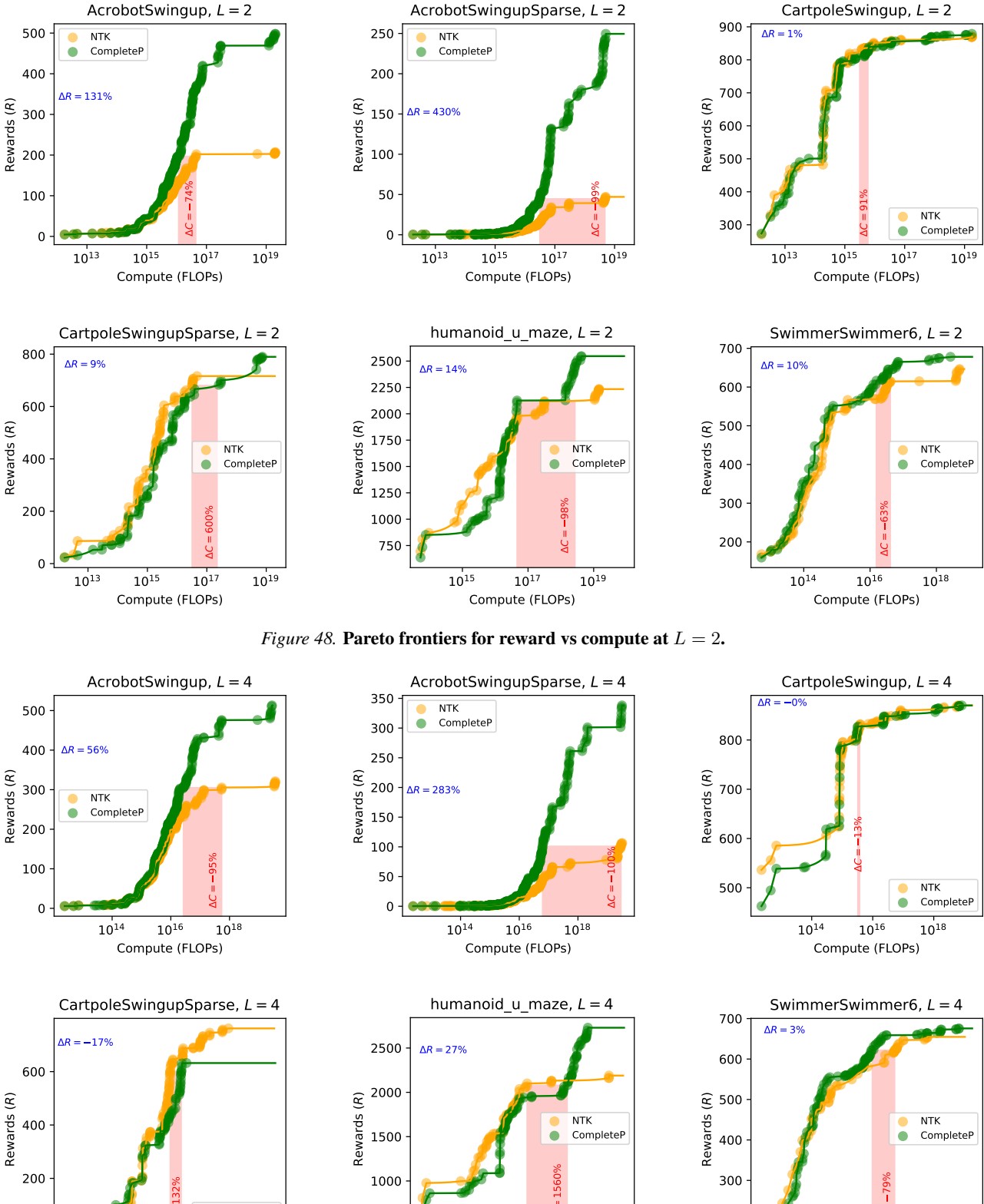

*Figure 48.* **Pareto frontiers for reward vs compute at** $L = 2$**.**

*Figure 49.* **Pareto frontiers for reward vs compute at** $L = 4$**.**

# T. Walltime Dependent Learning Performance

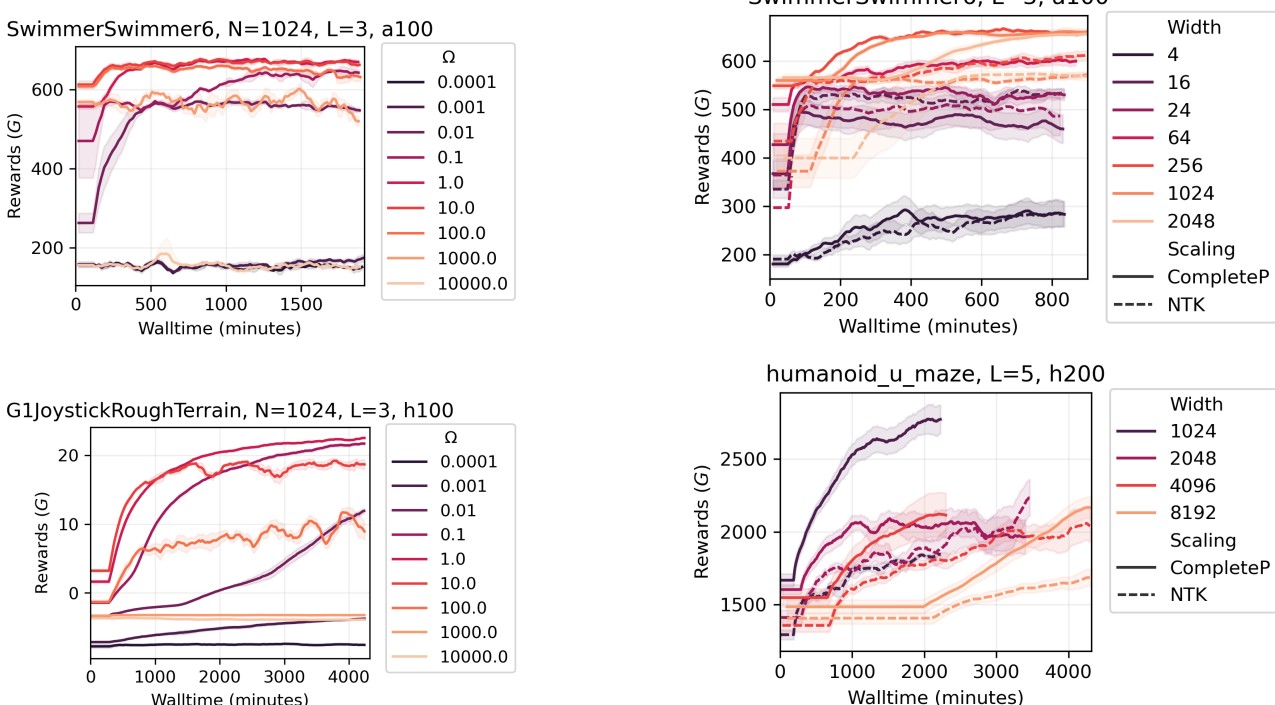

*Figure 50.* Reward versus wall clock. CompleteP demonstrates improved learning performance over NTK parameterization over different widths and sweeps on $\Omega$. These environments were trained with a single GPU type (A100, H100, or H200) for direct comparison. Note that different widths vary in runtime i.e. larger widths require longer walltime.

## U. Craftax Hyperparameter Transfer and Learning Curves

To verify that the width scaling behaviors observed on continuous control tasks also hold for a more challenging, discrete-action, partially observable environment, we additionally evaluate CompleteP, NTK, and Standard Parameterization (SP) on the Craftax benchmark (Matthews et al., 2024) with both PPO and SAC. Figures 51 and 52 show reward at a fixed timestep budget as a function of base learning rate $\eta_0$ for three widths $N$. Figure 53 shows PPO learning curves on Craftax at $L = 4$ with a proxy width of $W = 256$ to pick learning rate $\eta_0$ while scaling width.

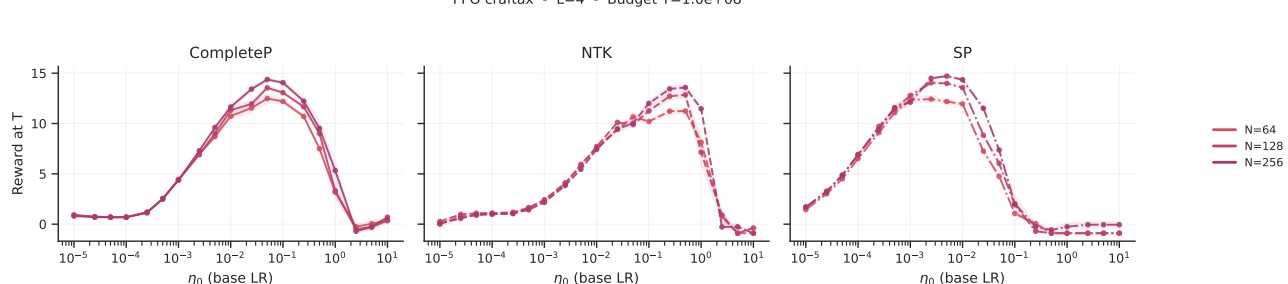

*Figure 51.* **Craftax PPO learning rate transfer across widths.** Final reward at timestep budget $T = 10^8$ as a function of base learning rate $\eta_0$ for CompleteP, NTK, and SP at $L = 4$ across widths $N \in \{64, 128, 256\}$.

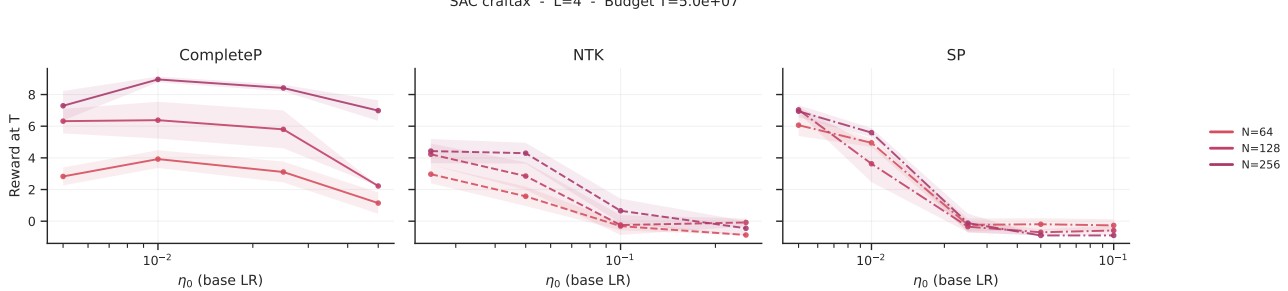

*Figure 52.* **Craftax SAC learning rate transfer across widths.** Final reward at timestep budget $T = 5 \times 10^7$ as a function of base learning rate $\eta_0$ for CompleteP, NTK, and SP at $L = 4$ across widths $N$.

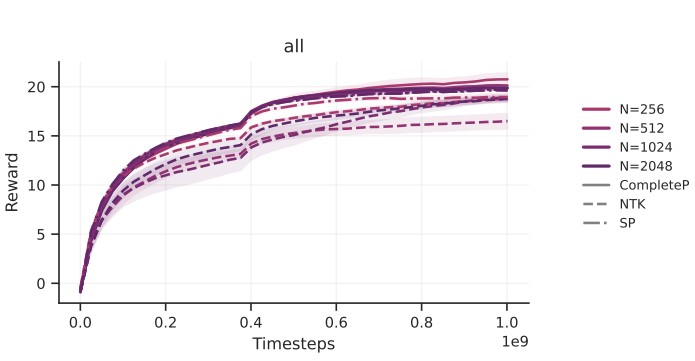

*Figure 53.* **Craftax PPO learning curves across widths and parameterizations.** Reward versus environment steps for PPO on Craftax at $L = 4$ with proxy width $W = 256$, sweeping width $N$ for CompleteP, NTK, and SP.

# V. Comparing CompleteP $\alpha = 1$ and $\alpha = 1/2$

We provide a brief comparison of the depth scaling coefficient as mentioned in (Dey et al., 2026; Mlodozeniec et al., 2025) via an experiment on learning rate transfer on the CheetahRun environment with PPO at a fixed width, varying depth.

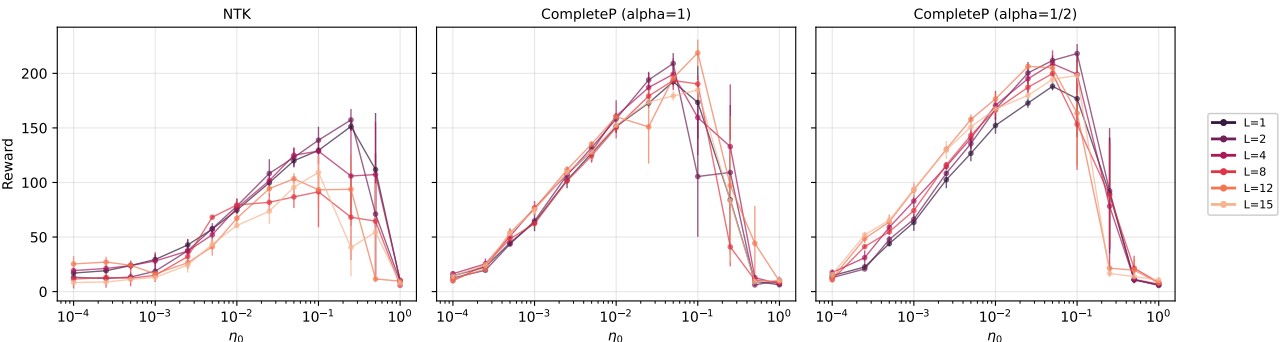

*Figure 54.* Reward on CheetahRun as a function of the base learning rate $\eta_0$, at width $N = 128$ and $1.1 \times 10^7$ environment steps. Each panel corresponds to a parameterization: NTK, CompleteP ($\alpha = 1$), and CompleteP ($\alpha = 1/2$). Within each panel, one curve is shown per network depth $L \in \{1, 2, 4, 8, 12, 15\}$. Points are shown with error bars.

## W. Batch-Size Scaling

To probe learning-rate transfer under large-batch training, we ran a batch-size sweep with PPO on CheetahRun. We held the total batch size fixed at $B \times \text{NMB} = 32{,}768$ environment steps and the training budget at $10^7$ environment steps, and varied the minibatch size $B \in \{128, 512, 2048, 8192, 32768\}$ while reducing the number of minibatches $\text{NMB} \in \{256, 64, 16, 4, 1\}$ accordingly, so that the same amount of data is ingested in every configuration. For each minibatch size we swept the base learning rate $\eta_0$ from $10^{-4}$ to $10^0$ across widths $N \in \{32, 64, 512\}$ and depths $L \in \{1, 4\}$ for both the CompleteP and NTK parameterizations. Figure 55 shows the resulting learning-rate basins.

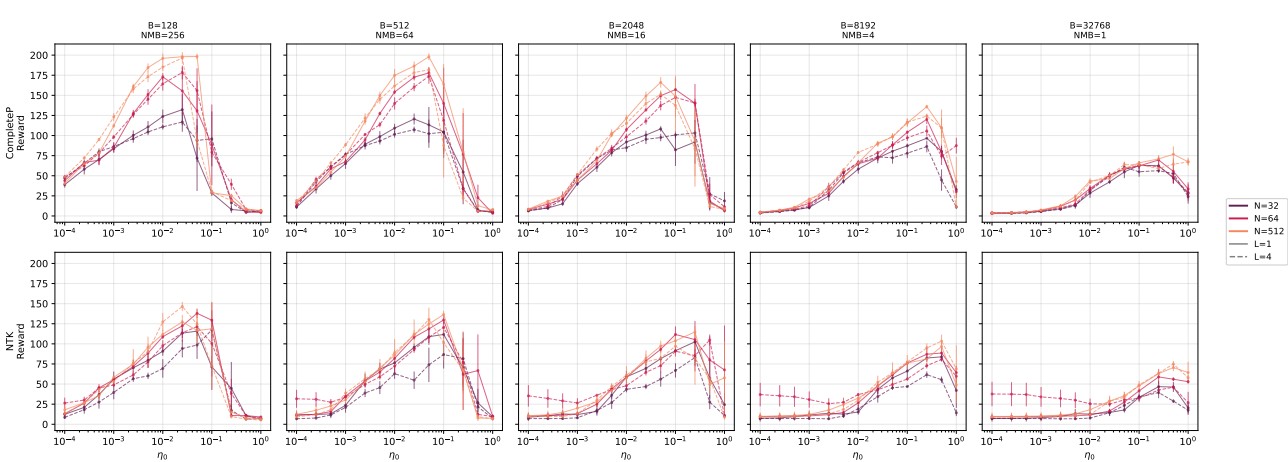

*Figure 55.* Reward on CheetahRun as a function of the base learning rate $\eta_0$ (log scale) for PPO across different minibatch sizes. The total batch size is fixed at 32,768 and the training budget at $10^7$ environment steps. From left to right, columns increase the minibatch size $B$ and decrease the number of minibatches NMB so that $B \times \text{NMB} = 32{,}768$: $B$=128 (NMB=256), $B$=512 (NMB=64), $B$=2048 (NMB=16), $B$=8192 (NMB=4), and $B$=32768 (NMB=1). Top row: CompleteP; bottom row: NTK. Color denotes width $N \in \{32, 64, 512\}$ and line style denotes depth ($L$=1 solid, $L$=4 dashed).

