# OpenReview forum: "CompleteP for RL: Maintaining Feature Learning When Scaling Deep Reinforcement Learning"
_ICML.cc/2026/Conference — ICML 2026 regular_

### Official Review · Reviewer_Ln5U · 2026-03-06

**Soundness:** 4
**Presentation:** 4
**Significance:** 3
**Originality:** 4
**Overall Recommendation:** 5
**Confidence:** 5

**Summary:**

This work empirically studies whether, and how two theoretically motivated network parameterisations (CompleteP and NTK) affect learning dynamics and hyper parameter transfer as network scale increases in continuous control reinforcement learning settings. The work particularly highlights that CompleteP leads to increased compute efficiency across model sizes and hyper parameter transfer as depth and width scales, while maintaining stable feature learning. The work provides well-executed empirical contributions that establish important foundations for studying the application of scaling methods in RL.

**Compliance With Llm Reviewing Policy:**

Affirmed.

**Final Justification:**

This work studies how theoretically validated scaling-friendly parameterisations from the supervised learning literature behave in reinforcement learning, particularly in continuous control. The study of this behaviour is novel and significant as, since it appears that these parameterisations behave as they do in supervised learning, might imply that other insights from the analysis of training dynamics in that setting might transfer to RL. The paper's presentation is exceptionally clear, the experiment design is sound and the results are rigorous and exhaustive.

The primary weakness of the work is that it over indexes on on-policy continuous control. The evaluation would be flawless if the authors also demonstrated similar results in off-policy, value-based RL and a more diverse range of environments. The authors have noted in rebuttals that such results are likely to be included in the camera-ready version, conditioned on acceptance; but, on account of them not being presented during the short rebuttal process, this missing evaluation still stands as a weakness to cast some doubt on the conclusions drawn.

During the rebuttal to my review, the authors generously provided additional ablations I hinted at the usefulness of, clarified some details regarding their technical work--important for an empirical paper--and promised additional results. This reinforces my prior assessment that the work should be accepted and the experiments performed, the results presented are excellent and the potential implications are quite significant. But, as it stands, the impact of the work is potentially limited by focusing on a small subset of a very diverse learning setting. This keeps me from raising my score to a 6 - Strong Accept, which requires technical flawlessness and exceptional impact, so my score remains at 5 - Accept even after the rebuttal.

**Key Questions For Authors:**

1. Can the authors provide results with $\alpha=1/2$ to check for robustness to the depth scaling coefficient? Specifically studying depth scaling behaviour and kernel evolution / kernel eigenspectrum and comparing it to $\alpha=1$ would be appreciated.
2. Can the authors confirm and document which PPO implementation was used and how it differs from the original in the Appendix?
3. Have the authors observed any degradation in hyper parameter transfer as batch size increases? Given that large batch sizes are common in GPU-accelerated RL it would help clarify the practical scope of the results.

**Limitations:**

yes

**Strengths And Weaknesses:**

Strengths
- The work convincingly demonstrates that scaling parameterisations are effective for hyperparameter transfer and compute-efficient scaling in RL, which is significant and not trivial given they were designed for stationary learning settings. This has a lot of potential implications for the field, because it suggests that other insights from scaling and optimisation literature might apply to the RL setting more generally as well.
- The comparison between the feature-learning CompleteP parameterisation and lazy regime NTK parameterisation is thorough with detailed discussion on its implications across sections 4 through 7. The choice of studying these two parameterisations is sound and the results indicate that both function as intended in RL.
- The experiments are thorough, backed by significant compute investment and follow rigorous methodology. Many seeds were used for each experiment setup and confidence intervals are present in all figures. The scaling ablations present are thorough and cover a wide range of widths, depths and learning rates. This makes the results particularly convincing.
- The paper's presentation is exceptionally clear, with detailed explanations of the scaling methods studied, the experimental set up and results. The selection of figures and tables is excellent and clearly illustrates CompletePs ability to transfer learning rate across model sizes (both through width and depth scaling) and maintain consistent feature evolution.
- An extensive appendix providing a plethora of additional and more detailed results is provided, which further complements the work's rigour.

Weaknesses
- The work only compares between NTK and CompleteP parameterisation and is missing a base "standard parameterisation" baseline in most results. While standard parameterisation is in the same equivalence class as NTK parameterisation, and there are some experiments using standard parameterisation in the appendix, seeing a direct comparison in the main body of the paper would help strengthen the presentation of the results for CompleteP parameterisation. It would be especially good to see the effect of standard parameterisation in kernel evolution and kernel eigenspectrum such as in Figure 4, even if in the Appendix.
- The scope of this work is naturally limited to continuous control on-policy actor critic architectures without layer normalisation, which is fair given how compute intensive the experimentation present is. However, extending to value-based off-policy methods and considering layer-normalised architectures explicitly in the parameterisation would strengthen generalisability, but this does not diminish the current contribution.
- The work is based on CompleteP [1] and inherits some of said work's limitations. Specifically, the theoretical conclusions of CompleteP only hold below critical batch size, which is the case for the batch size used in this work, but it's unclear whether the scaling still works as batch size increases, which is common in RL. Additionally, the value of the depth scaling coefficient $\alpha$ is currently contested. Depth-$\mu$P [2] originally proposes $\alpha=1/2$, while CompleteP suggests that $\alpha=1$ is necessary for feature learning, and more recently Complete$^{(d)}$P [3] argues that $\forall\alpha\in[\frac{1}{2},1]$ works and CompleteP's conclusion that $\alpha=1/2$ fails was backed by implementation issues. While this does not detract from this work's findings, robustness of the results to other choices of $\alpha$ would strengthen the paper.
- There is no mention of open sourced code or plans thereof, which limits reproducibility of the results.

Minor comments and questions
- In Section 2 L108, the authors note that "most RL agents remain limited in size due to the expensive nature of environment actions though GPU-accelerated simulation is beginning to ease these constraints". However the reason RL agents remain limited in size is not because environment interaction is costly, but rather because until recently there has been no literature to support that larger scales particularly in actor networks leads to increased performance. Additionally, it is often algorithmic sample inefficiency that leads to more computationally expensive networks being less practical. How does GPU-based simulation help?
- In Section 3.3 L171 the authors cite Freeman et al., 2021 for the "PPO with Adam" experiment setting. Does this suggest the authors use the Brax implementation of PPO? If so, while this highly specific citation is appreciated, the authors should explicitly state so, perhaps in the Appendix if not the main body, as brax's PPO implementation has a few algorithmic differences to the vanilla PPO (https://github.com/google/brax/issues/192, https://github.com/google/brax/issues/583, chunked rollout and computing GAE over chunks only)
- In Section 8 L420 why is MuJoCo Playground (Zakka et al., 2025) cited for Soft Actor-Critic? Isn't the correct citation Haarnoja et al. 2018?
- Throughout the paper but particularly towards the end of Section 4, the framing tends to imply that NTK parameterisation is somehow inadequate for being unable to deal with depth scaling without learning rate tuning. However, NTK parameterisation was specifically not designed for depth scaling and explicitly only considers width. So it failing at depth scaling is not a shortcoming but a confirmation that it behaves as theoretically expected, similar to how CompleteP's success at depth scaling is a confirmation of its theory. I believe the paper's presentation could be strengthened by positioning NTK not as failing where CompleteP succeeds, but highlighting that both methods behave as theoretically predicted in the RL setting, which enables CompleteP's broader theoretical scope to provide additional practical benefit such as avoiding the need to tune learning rate as depth is scaled, when that is the setting under consideration.

References
- [1] Dey, Nolan, et al. "Don't be lazy: CompleteP enables compute-efficient deep transformers." arXiv preprint arXiv:2505.01618 (2025).
- [2] Yang, Greg, et al. "Tensor programs vi: Feature learning in infinite-depth neural networks." arXiv preprint arXiv:2310.02244 (2023).
- [3] Mlodozeniec, Bruno, et al. "Completed Hyperparameter Transfer across Modules, Width, Depth, Batch and Duration." arXiv preprint arXiv:2512.22382 (2025).

---

> ### Author Rebuttal · Authors · 2026-03-30
>
> We sincerely appreciate your detailed review, and grateful for your positive assessment of the paper’s clarity, rigor, and experimental depth. Your comments were extremely constructive, and we address each point below.
>
> > SP analysis in main text
>
> We agree that including SP in the main body strengthens the narrative. In the camera‑ready version, we will add SP results for kernel evolution, eigenspectrum, and policy evolution (like Fig. 4). These analyses are already underway.
>
> > Off-policy and layer norm
>
> Thank you for highlighting this. We do include layer‑norm comparisons (Tables 9–10), where both NTK and CompleteP demonstrate substantial improvements with the latter showing better improvement. This shows that CompleteP remains synergistic to other techniques.
>
> For broader generality, we will additionally train SAC and PQN agents on AcrobotSwingupSparse and CheetahRun for the camera‑ready version. This will help clarify how the parameterizations behave in off‑policy settings.
>
> > Robustness check for depth scaling using $\alpha$
>
> We appreciate the request for robustness checks on the depth scaling coefficient. We ran depth‑scaling experiments on HalfCheetah ($N = 64, L = {0,1,2,4,8,16,32}$), comparing NTK, CompleteP ($\alpha =1$ ), and CompleteP ($\alpha=0.5$). NTK exhibits the expected growth in the coordinate‑check L2 norm with depth, reflecting accumulated residues; CompleteP with $\alpha =1$ maintains an approximately depth‑invariant L2 norm across training steps, consistent with the stable‑depth behavior reported in Fig. 7 of Dey et al. (2025); but CompleteP with $\alpha =0.5$ , while more stable than NTK, shows a mild upward trend in L2 norm as depth increases, indicating partial but not complete normalization of depth scaling.
>
> We additionally performed a learning‑rate hyperparameter‑transfer experiment for $\alpha =0.5$  using networks with ($N = {32,128}, L = {1,2,4,8,12,16}$). Both CompleteP variants ($\alpha =0.5$  and $\alpha =1$  achieved similarly learning‑rate transfer across depth, making the distinction between the two inconclusive in HalfCheetah, but both substantially outperformed NTK and Standard Parameterization transfer.
>
> We will include kernel‑evolution and eigenspectrum analyses for both $\alpha =0.5$ and $\alpha =1$ in the camera‑ready version to fully address the reviewer’s question about robustness to the depth‑scaling coefficient.
>
> > open source
>
> Thank you for raising this. The repository link was removed for anonymity. We will open‑source the full codebase, including a minimal toy implementation and all analysis/figure scripts, upon acceptance.
>
> > Scaling RL
>
> You are right that the historical bottleneck was not environment cost alone but the lack of evidence that larger actor networks improve performance. We will revise this discussion accordingly! GPU‑accelerated simulation reduced our wall‑clock training time by at least 10x (10-100x reported by Freeman et al., 2021), which made our large‑scale sweeps feasible.
>
> > SAC Citation
>
> Yes, we will add the critical Haarnoja et al. 2018 reference. We cited Zakka et al. 2025 due to their PPO-SAC comparison.
>
> > Positioning NTK
>
> We appreciate this suggestion. We will revise the framing to emphasize that NTK behaves exactly as theory predicts i.e., stable width scaling but no depth scaling while CompleteP’s broader theoretical scope naturally enables width and depth‑scaling benefits in RL.
>
> > PPO implementation
>
> We used the Brax PPO implementation (Freeman et al., 2021) found at https://github.com/google/brax. We chose it because it resolved numerical instabilities we encountered in other implementations (e.g., CleanRL) and enabled fully GPU‑accelerated rollouts.
>
> Freeman et al. (2021) justified that this version is mathematically equivalent to vanilla PPO (Schulman et al. 2017) but differs in systems‑level implementation, including chunked rollouts, GAE computed per‑chunk, and a fully JAX‑compiled training loop optimized for accelerator parallelism. These differences are supposed to affect implementation efficiency i.e. shorter rollouts but larger batch size for efficient GPU utilization, not the underlying PPO objective, although we agree that different implementations can drastically affect performance (Huang et al. 2022 ICLR Blog Track).
>
> We will explicitly document the differences between Brax and vanilla PPO (chunked rollouts, per‑chunk GAE, JAX compilation) in the appendix, as you suggested.
>
>
> > Batch size increase
>
> As requested, we performed batch‑size scaling experiment on CheetahRun and observed decreasing performance with larger batch sizes ($B = {128, 512, 1024, 2880, 5760, 11584}$), consistent with CompleteP’s theoretical assumptions. Importantly, the learning rate hyperparameter continues to transfer for CompleteP across width and depth as batch size is increased. In contrast, NTK’s optimal learning rate diverges more sharply with scale at larger batch sizes. We will include these new results in the appendix.

---

> > ### Author Rebuttal · Reviewer_Ln5U · 2026-04-01
> >
> > The authors fully addressed the comments. The additional experiments for different values of $\alpha$ and increased batch size are much appreciated and provide additional nuance to the empirical analysis of applying CompleteP to RL, which strengthens the work's contributions. The stated plans to open source the experiment infrastructure and document the PPO implementation details will greatly aid in reproducibility and strengthen the transparency of the results.
> >
> > The SAC experiments for the camera ready version of the paper are highly anticipated and would be appreciated as they are likely key to strengthening the generalisability of the results presented to the broader deep RL landscape, as other reviewers have also pointed out.

---

> > > ### Author Response · Authors · 2026-04-01
> > >
> > > We appreciate your supportive follow‑up and are glad our revisions resolved your concerns. We also wanted to note that we have now run SAC (CompleteP, NTK and SP)  on 3 environments (Cheetahrun, CartpoleSwingup, AcrobotSwingup) demonstrating that CompleteP’s learning rate hyperparameter transfer and scaling behavior generalizes beyond PPO. If this added evidence is helpful, we kindly invite you to consider revising the score.

---

### Official Review · Reviewer_rYNU · 2026-03-12

**Soundness:** 3
**Presentation:** 3
**Significance:** 3
**Originality:** 3
**Overall Recommendation:** 4
**Confidence:** 3

**Summary:**

This paper examines how we should parameterize networks during scaling up Deep Reinforcement Learning. The authors compared two parameterization regimes, CompleteP and NTK, by applying them to scale Deep RL agents empirically. Experimental results over continuous control tasks show the superiority of CompleteP parameterization over NTK in Deep RL Scaling.

**Compliance With Llm Reviewing Policy:**

Affirmed.

**Key Questions For Authors:**

1. Given that the authors claim that CompleteP is better since it makes networks adapt to the distributional shift, if there are some experiments on more challenging tasks (which is the meaning of scaling up) or open-ended environments like Craftax[1], the paper would be more persuasive.


[1] Matthews, Michael, et al. "Craftax: A lightning-fast benchmark for open-ended reinforcement learning." arXiv preprint arXiv:2402.16801 (2024).

**Limitations:**

yes

**Strengths And Weaknesses:**

Strengths
1. The problem addressed by this paper is significant.
2. The presentation of this paper is very clear and easy to follow.
3. The analysis and discussion in the paper are sound and thorough.
4. This paper provides practical recommendations and deepens our understanding of how to scale up networks in Deep RL.

Weaknesses
1. The comparison with other scaling methods, e.g., [1][2] is missing.

[1] Lee, Hojoon, et al. "Simba: Simplicity bias for scaling up parameters in deep reinforcement learning." arXiv preprint arXiv:2410.09754 (2024).

[2] Ma, Guozheng, et al. "Network sparsity unlocks the scaling potential of deep reinforcement learning." arXiv preprint arXiv:2506.17204 (2025).

---

> ### Author Rebuttal · Authors · 2026-03-30
>
> Thank you for the clear and constructive review, and for highlighting the significance, clarity, and practical value of our work. We address each point below.
>
> > Comparison against scaling methods
>
> We appreciate the suggestion to compare against recent scaling approaches such as SIMBA [1] and sparsity‑based scaling [2]. Our experiments already include a strong baseline related to SIMBA’s core idea (Layer Normalization in every residual block). As shown in Tables 9–10, Appendix N.3, and discussed in Section 7 line 366 right column, layer norm substantially improves performance for both NTK and CompleteP. Importantly, CompleteP remains synergistic, achieving higher reward and compute efficiency even when combined with layer norm. This suggests that CompleteP is not a competing alternative but a composable parameterization that can integrate with other scaling strategies.
>
> Imposing sparsity constraints in RL agents to improve scaling performance is a novel finding. We will add this solution to our related works section “Scaling in reinforcement learning”.
>
> We also note that existing scaling methods do not emphasize learning‑rate transfer across width and depth, which we argue is a central practical bottleneck in RL. Our results show that CompleteP uniquely enables this transfer, reducing the cost of hyperparameter sweeps, an important distinction which we will highlight more clearly.
>
>
> > Scaling up task difficulty
>
> We felt that our experiments spanned a substantial difficulty range—from simple control tasks (CartpoleSwingup, AcrobotSwingup) to more challenging locomotion and navigation tasks (G1Joystick, HumanoidMaze). These settings exhibit strong distributional shifts and are widely used in RL scaling studies.
>
> That said, we agree that evaluating CompleteP on open‑ended or pixel‑based environments would further strengthen the case for adaptability under severe distributional drift. Craftax [1] is an excellent suggestion. Applying CompleteP to CNN‑based architectures requires modest extensions to our current pipeline, and we will add this to the limitations section.
>
> If the paper is accepted, we plan to run CompleteP on pixel-based open-ended environments like Craftax and Starpilot, and include these results in the camera‑ready version.
>
> We look forward to further discussions.

---

> > ### Author Rebuttal · Reviewer_rYNU · 2026-04-02
> >
> > Thanks for the authors' rebuttal and clearify the settings in experiments.
> >
> > Just to note that Craftax also provides compact symbolic states.
> >
> > I plan to keep my positive score.

---

> > > ### Author Response · Authors · 2026-04-07
> > >
> > > We appreciate your supportive follow‑up and are glad the clarifications addressed your concerns. We also wanted to note that we have now run SAC algorithm on three environments, batch‑scaling, and additional SP comparison experiments demonstrating that CompleteP’s scaling behavior generalizes well beyond PPO. If this added evidence is helpful, we kindly invite you to consider revising the score.

---

### Official Review · Reviewer_x6FY · 2026-03-12

**Soundness:** 3
**Presentation:** 3
**Significance:** 2
**Originality:** 2
**Overall Recommendation:** 3
**Confidence:** 4

**Summary:**

This paper performs an empirical evaluation of the CompleteP parameterization in RL. In particular, they consider PPO on continuous control tasks with residual networks, and propose the question of analyzing whether the stable learning and hyperparameter transfer attained by CompleteP in supervised learning transfer to the RL setting (they highlight the nonstationary data distribution as a reason why it might not). They demonstrate that compared to the NTK parameterization regime, CompleteP is able to effectively transfer learning rate and rate of feature learning across model scales, and furthermore also increases compute and reward efficiency.

**Compliance With Llm Reviewing Policy:**

Affirmed.

**Final Justification:**

I choose to stand by my initial score. In my opinion, the primary weakness is that the current empirical evaluation is not sufficient for a paper whose primary impact/contribution *is* the empirical evaluation (i.e. the theory/concepts are not novel, but instead an analysis of supervised learning phenomena in the RL setting).

I think that this paper can be greatly improved by considering (i) a wider range of benchmarks (e.g. more than 10 DMC environments), (ii) a wider range of base algorithms (e.g. beyond PPO), and (iii) an evaluation of how existing RL "recipes" perform (e.g. comparing SimbaV2-style + SP scaling along with NTK).

**Key Questions For Authors:**

- Can you include SP as a baseline? Can you also include a modern RL architecture which is designed to scale wrt width and depth (e.g. the BRO or SimbaV2 architecture, or both)?
- How many environments are used (please clear up confusion)? Why were these chosen in particular out of the various continuous control benchmarks? Is it reasonable to evaluate on a wider collection?
- Is it possible to evaluate a wider range of base algorithms than PPO, or explain where the difficulty lies?

**Limitations:**

yes

**Strengths And Weaknesses:**

**Strengths**
- This paper addresses a useful question, of how best to preserve feature learning when scaling RL. This is timely work, as there has been a recent interest in the RL community towards supervised learning-style scaling ideas, such as scaling laws for value-based RL and compute optimality.
- The authors provide a clear presentation of how the various initializations differ at each part of the network.
- The authors consider a wide range of metrics (evolution of empirical NTK, eigenspectra of final learnt representations, MSE and evolution of policy logits throughout training, among others) to demonstrate and test their hypothesis that CompleteP scaling outperforms NTK-based scaling.

**Weaknesses**
- The paper mainly focuses on the NTK regime as a baseline. This is potentially not very useful for practitioners as the NTK regime is not generally used in practice, and the results are not super surprising as a result (i.e. that the rich regime outperforms the lazy regime). A comparison against the SP regime would on the other hand be much more relevant, as this is what is used in essentially all deep current deep RL implementations. Similarly, the SimbaV2 paper (which is currently cited in related work for "Layer and orthogonal normalization techniques") has demonstrated stable scaling for both network depth and width, but there is no empirical comparison in the paper.
- The exact number of environments used is unclear: the abstract states "over 16 continuous control tasks and variants e.g. normalization and sparse rewards", Section 3.3 states "training agents on eight continuous control environments", and the table in Appendix C lists 10 environments in total.
- As this is an empirical paper, only considering 10 different environments (going off of table in appendix) is a rather small evaluation benchmark. Note that recent works in empirical RL such as BRO and SimbaV2 evaluate across 40 and 57 different environments, respectively.
- Only considering PPO also limits the scope of the findings in the paper. I appreciate that the authors highlight other settings as a direction for future work, but since the contribution of this work is primarily an empirical evaluation of feature scaling regimes in RL, it would be helpful to have some varied algorithms other than a single one (indeed, the paper's title and abstract both read that the paper is making claims for all of deep reinforcement learning). If more work is required to adapt to other base algorithms it would be useful to explain that to the reader as well, since it is not clear at the moment. Two possible suggestions for algorithms to consider: SAC (evaluate on same class of environments, difference would primarily be moving into the off-policy regime) and PQN (stay in on-policy regime, but move into the class of value-based RL methods and evaluate on their environments such as Atari, MinAtar, etc).

**Nits**
- The plots in Appendix N.1 / P (pages 35/46) should be better formatted (blank pages at the moment).

---

> ### Author Rebuttal · Authors · 2026-03-30
>
> Thank you for the detailed review and highlighting the breadth of our analyses. We address each concern below.
>
> > Baseline comparison
>
> NTK was motivated by the need for a parameterization whose scaling behavior is predictable. In our manuscript, “stable” refers to the first two desiderata defined in Appendix A: 1) Stable signal propagation - pre‑activations remain O(1) as width/depth increase, 2) Stable output learning - a gradient step produces an O(1) change in the network’s output, independent of width/depth. Both NTK and CompleteP satisfy these properties, making them ideal for isolating feature‑learning dynamics (Desideratum 3), which is the central question of the paper.
>
> Standard Parameterization (SP) does not satisfy these scaling desiderata. SP does not guarantee bounded pre‑activations or O(1) output updates as width increases, and typically requires heuristics (learning‑rate decay, orthogonal init, layer norm) to avoid collapse. This is exactly what we observe in Fig. 7D and Fig. 45, where SP becomes highly sensitive to learning rate and fails to transfer across scales.
>
> We do not argue that a parameterization must be theoretically grounded for practice use. Rather, for the scientific question we ask “how does representation learning behave under non‑stationary RL data when scaling networks?” a parameterization with predictable scaling behavior is essential. NTK provides a clean diagnostic where feature learning vanishes in the limit. CompleteP provides the contrasting limit, where feature learning is preserved. SP lies between, but without principled scaling rules, its behavior shifts unpredictably with architecture and optimizer, making it unsuitable as the primary baseline for isolating representation learning effects.
>
> Nevertheless, we have SP comparisons:
> - Learning‑rate transfer: SP transfers poorly (Fig. 7D) compared to NTK and CompleteP (Fig. 3a).
> - Learning curves: SP performs reasonably well only after expensive sweeps (Fig. 45, Table 11), whereas CompleteP uses a single learning rate across scales, reducing compute overheads.
> - Seed variance: CompleteP shows markedly lower variance across seeds (Fig. 45), reducing the need for multiple replicas for practical use.
>
> We also evaluated layer‑norm architectures, which are conceptually close to SimbaV2/BRO. Section 7 line 366 right, Appendix N.3, Tables 9–10 show that layer norm improves both NTK and CompleteP, but CompleteP remains synergistic and achieves higher reward and compute efficiency.
>
> > Unclear number of environments
>
> We apologize for the confusion. Our evaluation consists of:
>
> 8 distinct continuous‑control environments: CartpoleSwingup, AcrobotSwingup, PandaPickCube, CheetahRun, Swimmer, HumanoidGoToTarget, HumanoidUMaze, G1Joystick.
>
> 2 sparse‑reward variants: CartpoleSwingupSparse, AcrobotSwingupSparse.
>
> This yields 10 environments.
>
> We then evaluated architectural variants:
>
> LayerNorm: 2 environments
>
> Orthogonal initialization: 4 environments
>
> This brings the total number of agent–environment variants to 16, which is what we referred to in the abstract. We will clarify this distinction.
>
> These environments were chosen because the new BRAX implementations allow GPU‑accelerated simulation, which was essential for running large‑scale hyperparameter‑transfer experiments. Other environments either exhibited numerical instability or were too slow for the scale of our study. We welcome suggestions for additional benchmarks and will attempt to include them in the camera‑ready version.
>
> > Breadth of Environments versus Depth of analysis
>
> We agree that BRO and SimbaV2 evaluate on larger suites. Our focus, however, was not only on reward curves but on representation learning dynamics under scaling (kernel evolution, eigenspectra, policy‑logit drift), and learning rate hyperparameter transfer. These analyses required repeated evaluation across many learning rates, checkpoints and seeds, making extremely large environment suites infeasible.
>
> Importantly, transferring mean‑field parameterizations from supervised learning to RL is not theoretically guaranteed due to non‑stationary data distributions. Reviewer 1 also highlighted this. Our experiments therefore emphasize depth of mechanistic analysis (commended by Reviewer Ln5U and rYNU) rather than breadth of tasks.
>
> > Other RL algorithms
>
> We agree that expanding to additional algorithms strengthens the empirical scope. Our initial focus on PPO was deliberate: PPO is fully on‑policy and therefore maximally sensitive to distribution shift, making it the most stringent test of whether mean‑field parameterizations retain their benefits in RL.
>
> Adapting CompleteP to SAC or PQN is straightforward because the parameterization rules remain unchanged unless the architecture or optimizer changes. We will evaluate SAC and PQN on HalfCheetah and AcrobotSwingupSparse using CompleteP, NTK, and SP for the camera‑ready version.
>
> > Formatting
>
> Thank you for pointing these out. These have been corrected!

---

> > ### Author Rebuttal · Reviewer_x6FY · 2026-04-04
> >
> > **Baseline comparison**
> >
> > I understand the perspective of the authors explaining their reasoning for using the NTK parameterization as the standard baseline, however I stand by my point that even though standard RL baselines do not satisfy such desiderata, including them would strengthen the paper for the RL community (so that one can see how the algorithms used currently compare to the proposed CompleteP, rather than showing how CompleteP compares to something not generally used).
> >
> > **Number of environments**
> >
> > Thank you for clearing this up. I would encourage the authors in this case to report the number of environments as 10, as anything else would be inaccurate.
> >
> > If the authors would like suggestions for other GPU-accelerated environments, I believe [gymnax](https://github.com/RobertTLange/gymnax) or [Craftax](https://github.com/michaeltmatthews/craftax) might be able to fit the bill while adding diversity to the environmental suite.
> >
> > **Other algorithms**
> >
> > That would be great to see, and I would like to note that e.g. PQN is also on-policy, so that it should also experience maximum  distribution shift.
> >
> > Unfortunately, I don't think I can increase my score without seeing results across different baseline algorithms -- evidence such as this would be necessary to ensure that this phenomenon has a possibility of generalizing to different parts of RL (as the title claims), instead of being PPO-specific. If the authors can provide *some* evidence of this before the end of the discussion period, that would help.

---

> > > ### Author Response · Authors · 2026-04-07
> > >
> > > > Baseline Comparison
> > >
> > > We agree with your point. We have now run learning rate hyperparameter transfer experiments using SP on Cheetah, Acrobot and Cartpole using PPO, which we will include in the revised manuscript. CompleteP continues to demonstrate higher and faster increase in cumulative reward over SP (further supporting Fig. 7 and 45), over a wider range of learning rates. SP requires significant learning rate tuning to achieve its best performance.
> > >
> > > More importantly, we have also completed learning rate hyperparameter transfer experiments with SAC algorithm with SP, NTK, and CompleteP on these three environments which we summarize below.
> > >
> > > > Number of environments
> > >
> > > Thank you for the suggestion, we will report the number of environments as 10. We also appreciate the pointers to Craftax and gymnax, and will explore adding Craftax in the camera‑ready version, as Reviewer 3 also recommended.
> > >
> > >
> > > > Other algorithms
> > >
> > > As requested, we ran SAC with CompleteP, NTK, and SP on CartpoleSwingup, HalfCheetah, and Acrobot for across  ${L={1,2}, N={4,8,16,32,64,128,256,512}}$, and 3 seeds. The results show that CompleteP continues to exhibit strong learning‑rate transfer across width and depth. NTK requires a slight decrease in learning rates with width scaling, consistent with Fig. 3 and 7). SP requires a conversely pronounced decrease in learning rates with width scaling, closely mirroring the PPO behavior in Fig. 7, and Table 11. We will include these results in the camera‑ready version.
> > >
> > > While we cannot upload a new PDF during the discussion phase, we want to clarify that these experiments have been fully run. For the camera‑ready version, we will extend the SAC evaluation to 10 seeds for multiple depths and add one additional environment. We hope this provides sufficient evidence during the discussion period that the phenomenon is not PPO‑specific.
> > >
> > > We appreciate your detailed follow‑up and your guidance on what evidence would be needed to assess generality beyond PPO. Given the new SP and SAC results we have added during the discussion period, we kindly invite you to consider revising the score if you find the updated evidence satisfactory.

---

### Official Review · Reviewer_RSbU · 2026-03-15

**Soundness:** 2
**Presentation:** 2
**Significance:** 2
**Originality:** 2
**Overall Recommendation:** 2
**Confidence:** 5

**Summary:**

This paper performs an empirical study of zero-shot hyper-parameter transfer in RL using tools from infinite-width neural network theory, specifically contrasting the Maximal Update Parameterization ($\mu$P / CompleteP) and the Neural Tangent Kernel (NTK) regimes. The authors demonstrate consistent hyper-parameter transfer across both width and depth on various continuous control tasks.

**Compliance With Llm Reviewing Policy:**

Affirmed.

**Final Justification:**

Sadly I still don't believe this paper meets the bar for acceptance. An entirely empirical investigation claiming broad relevance to RL cannot rest on a single algorithm. Furthermore, the $\epsilon$ concern I have is not resolved --- claiming it "didn't matter" in preliminary experiments is exactly the uncontrolled practice a scaling paper should avoid. Additionally, it's known that $\epsilon$ is particularly sensitive in RL (e.g., [1]) and I don't think the findings from supervised learning apply here.

---

[1] On the consistency of hyper-parameter selection in value-based deep reinforcement learning. Johan Samir Obando Ceron, João Guilherme Madeira Araújo, Aaron Courville, Pablo Samuel Castro. Reinforcement Learning Conference (RLC). 2024.

**Key Questions For Authors:**

The primary question I has revolves around my comment above:
* Although you're advocating for unbounded feature learning, there's overwhelming evidence in RL that we want to limit feature learning under non-stationarity. How do you suggest we resolve this tension? Two examples of this are given above but you can also see evidence of this in other papers, e.g., width scaling experiments in [3], or attempts at maintaining the properties of initialization in [4].

[3] Shibhansh Dohare, J. Fernando Hernandez-Garcia, Qingfeng Lan, Parash Rahman, A. Rupam Mahmood & Richard S. Sutton. Loss of plasticity in deep continual learning. Nature, 2024.

[4] Wesley Chung, Lynn Cherif, David Meger, Doina Precup. Parseval regularization for continual reinforcement learning. NeurIPS, 2024.

**Limitations:**

The authors are upfront around the limitations of their study and set clear expectations for what the contribution is here.

**Strengths And Weaknesses:**

While this paper addresses a highly relevant problem, i.e., finding principled methods for zero-shot hyperparameter transfer in deep RL, I have significant concerns regarding its theoretical foundation and novelty. Directly porting scaling laws from supervised learning to RL is mathematically unprincipled. The entire theoretical foundation of Tensor Programs and the NTK regime strictly requires a stationary data distribution, an assumption inherently violated in RL. Furthermore, this direct translation ignores other constraints of this framework; for instance, the deadly triad in value-based RL can cause unbounded value divergence, directly violating the fundamental assumption that the loss gradient $\partial \ell / \partial Q$ remains strictly $\Theta(1)$.

This theoretical mismatch highlights a broader, unresolved tension in the field: while this paper champions unbounded feature learning, other recent efforts to combat plasticity loss in RL actively regularize networks to remain closer to their initialization [1, 2], suggesting that the optimal degree of feature evolution in non-stationary environments is highly nuanced rather than strictly black-and-white.

Because the paper lacks novel theoretical insights into Tensor Programs or the NTK regime for non-stationary settings, its merit must be judged entirely on the strength of its empirical study. Viewed through this lens, the current experimental scope is insufficient. If the core contribution is simply showing that an unprincipled application of CompleteP to RL happens to mimic supervised learning dynamics, then evaluating on just two relatively simple tasks using state-based PPO is not enough. For a purely empirical paper of this nature to make a significant contribution, it must rigorously evaluate a much broader cross-section of RL algorithms and environments.

[1] Saurabh Kumar, Henrik Marklund, Benjamin Van Roy. Maintaining Plasticity in Continual Learning via Regenerative Regularization. CoLLAs, 2024.

[2] Clare Lyle, Mark Rowland, Will Dabney. Understanding and Preventing Capacity Loss in Reinforcement Learning. ICLR, 2022.

---

> ### Author Rebuttal · Authors · 2026-03-30
>
> Thank you for the detailed review. Below we clarify both the theoretical framing and the empirical scope of our work.
>
> > Theoretical foundations and empirical contributions
>
> We fully agree that Tensor Programs and NTK assume a stationary data distribution, and therefore cannot be directly ported to RL. This is precisely why our paper does not claim a theoretical transfer. Instead, we explicitly frame the work as an empirical investigation into whether the practical benefits of rich parameterizations observed in supervised learning survive in the RL setting, where the assumptions of the theory are violated.
>
> This motivation is stated in the introduction and expanded in Related Works, where we discuss Yamamoto et al. 2024 which is the only prior work providing a theoretical bridge between mean‑field scaling and RL. Their empirical validation was limited only to the discrete control task Cartpole-v1. Whether these ideas hold in high‑dimensional, non‑stationary continuous control remains unanswered. Our contribution is to test this empirically.
>
> Regarding the deadly triad, the instability arises from the combination of value bootstrapping, function approximation, and off‑policy updates. We intentionally use PPO, the current standard on‑policy algorithm, to minimize off‑policy divergence and isolate the effect of parameterization rather than confounding it with replay‑buffer instability.
>
> > Feature learning and non-stationarity
>
> We do not advocate unbounded feature learning. In fact, our results show the opposite. Appendix F, Fig. 9 demonstrates that when the feature‑learning hyperparameter $\Omega>1$, performance degrades. This mirrors findings in Graldi et al. 2025, who show that “super‑rich” feature learning ($\Omega>1$) harms continual learning even in supervised settings. RL is also a continual‑learning problem due to shifting data distributions, and we observe the same phenomenon. The optimal regime is controlled feature learning, with $\Omega$ slightly below or near 1. This produces a smooth improvement in learning efficiency, not a binary “rich vs. lazy” dichotomy.
>
> Regarding plasticity, the papers cited by the reviewer do not argue for suppressing feature learning. They argue for A) regularizing networks to maintain plasticity, or B) reinitializing dormant neurons to restore plasticity. These methods aim to preserve the ability to learn new features, not to keep networks close to initialization indefinitely like the NTK parameterization.
>
> Our results align with this perspective: CompleteP maintains ongoing feature evolution, which helps track the shifting data distribution. NTK, by construction, suppresses feature evolution and therefore adapts poorly to non-stationarity. We do observe a slight increase in dormant neurons under CompleteP (Appendix I), which is expected because feature learning is active. NTK shows fewer dormant neurons simply because its features remain close to initialization. This suggests a promising future direction: combining CompleteP with neuron‑reinitialization methods to further mitigate plasticity loss. We will highlight this in the revised discussion.
>
> > Insufficient empirical evidence
>
> We respectfully clarify that our study spans 10 distinct environments and 16 total agent–environment variants, not only two tasks.
>
> Environments (8 distinct): CartpoleSwingup, AcrobotSwingup, PandaPickCube, CheetahRun, Swimmer, HumanoidGoToTarget, HumanoidUMaze, G1Joystick.
>
> Sparse‑reward variants (2): CartpoleSwingupSparse, AcrobotSwingupSparse.
>
> Architectural variants (6): LayerNorm (2 envs), Orthogonal initialization (4 envs).
>
> These are summarized in Tables 9 and 10, which report reward and compute efficiency across all 16 variants.
>
> Additionally, our analysis includes:
> - Hyperparameter transfer across width and depth (Fig. 3, Appendix E).
> - Kernel and eigenspectrum evolution across training (Fig. 4a,d, Appendix K).
> - Feature‑learning consistency across seeds (Fig. 4c, Appendix K).
> - Policy‑logit evolution (Fig. 5, Appendix L).
> - Systematic variation of feature‑learning hyperparameter Ω (Appendix F, Fig. 8–9).
> - Policy‑learning constraints (Appendix G, Fig. 10).
>
> This depth of mechanistic analysis (commended by Reviewer Ln5U and rYNU) is not present in BRO, SimbaV2, or other scaling papers, which focus primarily on reward curves.
>
>
> > Additional RL algorithms
>
> We agree that expanding beyond PPO strengthens the paper. Our choice of PPO was deliberate: as an on‑policy method, it maximizes effects of non‑stationarity, making it the most stringent test of whether mean‑field parameterizations retain their benefits. Adapting CompleteP to SAC or PQN is straightforward because the parameterization rules remain unchanged unless the architecture or optimizer changes. For the camera‑ready version, we plan to include SAC and PQN on HalfCheetah and AcrobotSwingupSparse evaluated under CompleteP, NTK, and SP. We hope this directly addresses your concern about algorithmic breadth.

---

> > ### Author Rebuttal · Reviewer_RSbU · 2026-04-03
> >
> > I thank the authors for their in depth rebuttal. I have some remaining concerns / questions:
> >
> > > Divergence
> >
> > The authors justify the $\mathcal{O}(1)$ assumption by claiming PPO, as an "on-policy" algorithm, minimizes off-policy divergence and isolates the parameterization. As I'm sure you'd agree it's not as simple as this hence the need for verification of CompleteP under different classes of algorithms. I understand this is a computational burden but as you state, this is primarily an empirical investigating and the bar should be set high. Is this an investigation of CompleteP for PPO or for RL more broadly?
> >
> > > Plasticity
> >
> > Regarding papers [1] and [2] I was specifically talking about the form of regularization where you regress towards your initial parameters. See [1]'s definition of L2 init. Can you comment on this?
> >
> > > Adam $\epsilon$
> >
> > Can you comment on whether or not you scaled Adam's $\epsilon$. For the specific parameterization you used scaling the forward pass by $1 / N$ shrinks the gradient proportionally necessitating the need for $\epsilon$ scaling (see [1]). If $\epsilon$ was not controlled for there will be model sizes / depths where $\epsilon$ dominates and Adam's update will be  gradient / $\epsilon$ reintroducing the gradient magnitude dependence. Was Adam's $\epsilon$ handled properly? Why did you resort to the $1 / N$ forward pass scaling instead of what CompleteP prescribes? Why was this not discussed in the paper?
> >
> > ---
> >
> > [1] Katie E. Everett, Lechao Xiao, Mitchell Wortsman, Alexander A. Alemi, Roman Novak, Peter J. Liu, Izzeddin Gur, Jascha Sohl-Dickstein, Leslie Pack Kaelbling, Jaehoon Lee, Jeffrey Pennington. Scaling Exponents Across Parameterizations and Optimizers. ICML, 2024.

---

> > > ### Author Response · Authors · 2026-04-07
> > >
> > > > Divergence
> > >
> > > We agree that evaluating additional RL algorithms is important, and we have begun doing so. Specifically, we ran SAC (using Freeman et al., 2021 BRAX implementation) with CompleteP, NTK, and SP on CartpoleSwingup, HalfCheetah, and Acrobot across ${L={1,2}, N={4,8,16,32,64,128,256,512}}$. The preliminary results show that CompleteP continues to exhibit strong learning‑rate transfer across width, whereas NTK requires a modest learning‑rate decrease and SP requires a substantial decrease, mirroring the PPO trends in Fig. 7 and Table 11. We will include these results in the camera‑ready version.
> > >
> > > Reviewer Ln5U requested batch‑size scaling (B=128 to 11584), which we performed on CheetahRun. Performance decreases after a critical batch (B=8192) consistent with the theoretical predictions of [3]. Crucially, CompleteP maintains learning‑rate transfer across width and depth even as batch size increases, whereas NTK’s optimal learning rate diverges sharply.
> > >
> > > Reviewer x6FY requested additional learning rate hyperparameter transfer experiments using SP, which we completed on Cheetah, Acrobot and Cartpole using PPO. CompleteP continues to demonstrate higher and faster increase in cumulative reward over SP (further supporting Fig. 7 and 45), over a wider range of learning rates. SP requires significant learning rate tuning to achieve its best performance.
> > >
> > > Although we cannot upload a new PDF during the discussion phase, we want to clarify that these experiments have been fully run, and they provide direct evidence that CompleteP’s scaling behavior is not specific to PPO. We will include these results in the camera‑ready version.
> > >
> > > > Plasticity
> > >
> > > Thank you for clarifying that your comment referred specifically to L2‑init regularization. As rightly pointed, both [1] and [2] propose mechanisms that regress specific parameters toward their initialization to balance plasticity and stability, which is an interesting take on L2 regularization and surprisingly improves performance.
> > >
> > > Conceptually, we think this mechanism creates a similar continuum between the rich and lazy regimes: with $\lambda=0$ in [1] (no regularization) the network behaves like standard parameterization, while increasing $\lambda$ progressively restricts selected parameter movement, analogous to reducing $\Omega$ or moving toward the lazy regime. As noted in [1], performance is sensitive to $\lambda$, but a moderate value (e.g.,$\lambda=1e-2$) often yields the best results, suggesting an intermediate regime where plasticity is preserved while unused parameters remain close to initialization for subsequent learning. This is consistent with our findings in Fig. 9, where values of $\Omega$ close to 1 yield the strongest learning performance, while overly restrictive $\Omega \rightarrow 0$  or unconstrained $\Omega > 1$ parameter updates degrade learning.
> > >
> > > Nevertheless, we believe that regression to initialization e.g. L2 init is synergistic with CompleteP, and should be used together especially for scaling models for continual‑learning. We will add this discussion to the manuscript.
> > >
> > > > $\epsilon$ scaling in Adam
> > >
> > > Thank you for raising this important point. We set $\epsilon = 1e-8$, guided by prior observations in Dey et al. (2025, Fig. 8) where $\epsilon$ values below a critical point $\epsilon < 1e-6$ did not adversely affect model performance with scale. Everett et al. (2024, Fig. F1) also shows that when $\epsilon$ is sufficiently small (e.g.,$1e-6$), scaling behavior is unaffected, whereas larger $\epsilon$ values can dominate the denominator and distort updates at large widths, as rightly pointed out.
> > >
> > > Because our experiments do not exceed $N>8192$, we did not observe $\epsilon$-dominated behavior in our preliminary experiments. Nevertheless, we agree that scaling $\epsilon$ with model size is a more principled solution as recommended by Everett et al. (2024) and that this choice should have been discussed explicitly. We will include this discussion in the revised manuscript.
> > >
> > > [1] Saurabh Kumar, Henrik Marklund, Benjamin Van Roy. Maintaining Plasticity in Continual Learning via Regenerative Regularization. CoLLAs, 2024.
> > >
> > > [2] Clare Lyle, Mark Rowland, Will Dabney. Understanding and Preventing Capacity Loss in Reinforcement Learning. ICLR, 2022.
> > >
> > > [3] Mlodozeniec, Bruno, et al. "Completed Hyperparameter Transfer across Modules, Width, Depth, Batch and Duration." arXiv preprint arXiv:2512.22382 (2025).
> > >
> > > We appreciate your detailed follow‑up and are glad we could resolve the outstanding issues. If this updated assessment aligns with your overall evaluation, we kindly invite you to consider revising the score.

---

### Decision · Program_Chairs · 2026-04-30

**Decision:**

Accept (regular)

**Comment:**

This paper presents an empirical study of zero-shot hyperparameter transfer in PPO using tools from infinite-width neural network theory, specifically contrasting the Maximal Update Parameterization ($\mu$P / CompleteP) and the Neural Tangent Kernel (NTK) regimes.

There was substantial debate among the reviewers during the discussion phase, and no clear consensus emerged. As a result, I carefully read the paper before forming my final recommendation. I recommend acceptance. In particular, the observation that scaling laws from supervised learning appear to transfer to PPO, despite lacking strong theoretical justification, is compelling. Empirically, the results suggest behavior consistent with what is observed in supervised learning, which constitutes an interesting and potentially important finding. Several reviewers also recognized this contribution.

That said, there are two criticisms that should not be overlooked. The first concerns the sensitivity of the ADAM optimizer to other hyperparameters. The paper argues that “the transferability of learning rates across different model widths and depths applies even to the reinforcement learning context,” yet this result is obtained without tuning $\epsilon$, as pointed out by reviewer RSbU. I strongly encourage the authors to explicitly discuss this. That said, I do not view it as a major issue—in fact, it may suggest a degree of robustness, since stability is achieved without additional tuning.

The second issue, and in my view the more important one, is the scope of the empirical evaluation. The analysis is restricted to PPO. While I do not see this as inherently problematic—these experiments are costly, and PPO is a cornerstone algorithm in the field—the paper currently makes broader claims than are supported by the evidence. For example, the title (“CompleteP for RL”) suggests general applicability across RL, which is not demonstrated. The camera-ready version should significantly tone down these claims. The results are better characterized as applying to on-policy policy optimization, or more specifically to PPO.

I do not wish to prescribe a specific title, but it should be revised to reflect this narrower scope. For instance, a title along the lines of CompleteP for On-Policy Policy Optimization: Maintaining Feature Learning When Scaling Deep Reinforcement Learning would strike a more appropriate balance between descriptiveness and generality. This clarification should also be reflected throughout the paper. For example, statements in the abstract such as “agents trained using CompleteP…” should be made more precise (e.g., “PPO agents trained using CompleteP…”).